# SPATIALAB: CAN VISION–LANGUAGE MODELS PERFORM SPATIAL REASONING IN THE WILD?

**Azmine Toushik Wasi[1,2], Wahid Faisal[1,2], Abdur Rahman[1,2],**
**Mahfuz Ahmed Anik[1,2], Munem Shahriar[1,3], Mohsin Mahmud Topu[1,2],**
**Sadia Tasnim Meem[1,2], Rahatun Nesa Priti[1,2], Sabrina Afroz Mitu[1,2],**
**Md. Iqramul Hoque[1,2], Shahriyar Zaman Ridoy[1,4], Mohammed Eunus Ali[5],**
**Majd Hawasly[6], Mohammad Raza[6], Md Rizwan Parvez[6]**
[1]Computational Intelligence and Operations Laboratory (CIOL)
[2]Shahjalal University of Science and Technology (SUST) [3]BRAC University
[4]North South University (NSU) [5]Monash University
[6]Qatar Computing Research Institute (QCRI)
Correspondence to: `azminetoushik.wasi@gmail.com`, `mparvez@hbku.edu.qa`

## ABSTRACT

Spatial reasoning is a fundamental aspect of human cognition, yet it remains a major challenge for contemporary vision–language models (VLMs). Prior work largely relied on synthetic or LLM-generated environments with limited task designs and puzzle-like setups, failing to capture the real-world complexity, visual noise, and diverse spatial relationships that VLMs encounter. To address this, we introduce **SPATIALAB**, a comprehensive benchmark for evaluating VLMs' spatial reasoning in realistic, unconstrained contexts. SPATIALAB comprises 1,400 visual question–answer pairs across six major categories: *Relative Positioning, Depth & Occlusion, Orientation, Size & Scale, Spatial Navigation, and 3D Geometry*, each with five subcategories, yielding 30 distinct task types. Each subcategory contains at least 25 questions, and each main category includes at least 200 questions, supporting both multiple-choice and open-ended evaluation. Experiments across diverse state-of-the-art VLMs, including open- and closed-source models, reasoning-focused, and specialized spatial reasoning models, reveal a substantial gap in spatial reasoning capabilities compared with humans. In the multiple-choice setup, InternVL3.5-72B achieves 54.93% accuracy versus 87.57% for humans. In the open-ended setting, all models show a performance drop of around 10–25%, with GPT-5-mini scoring highest at 40.93% versus 64.93% for humans. These results highlight key limitations in handling complex spatial relationships, depth perception, navigation, and 3D geometry. By providing a diverse, real-world evaluation framework, SPATIALAB exposes critical challenges and opportunities for advancing VLMs' spatial reasoning, offering a benchmark to guide future research toward robust, human-aligned spatial understanding. SPATIALAB is available at: `https://spatialab-reasoning.github.io/`.

## 1 INTRODUCTION

Spatial reasoning is a fundamental aspect of human cognition that involves understanding spatial layouts, imagining and manipulating objects, and navigating environments. It supports applications ranging from robotics, autonomous driving and augmented/virtual reality to geospatial analytics, computer graphics, human–computer interaction, and education (Huang et al., 2024b; Qi et al., 2024; 2025). For aritificial agents, autonomous systems that perceive and act in dynamic environments, reliable spatial competence requires interpreting geometric relations, depth cues, occlusion patterns, and agent-centric perspectives in scenes that are visually noisy and structurally complex (Bar-Anan et al., 2006; Cai et al., 2025; Trope & Liberman, 2010; Ramalho et al., 2018). Recent vision–language models have advanced multimodal representation and language grounding, but their spatial judgments remain fragile in realistic environments (Chen et al., 2025; Kosoy et al., 2025; Pothiraj et al., 2025).

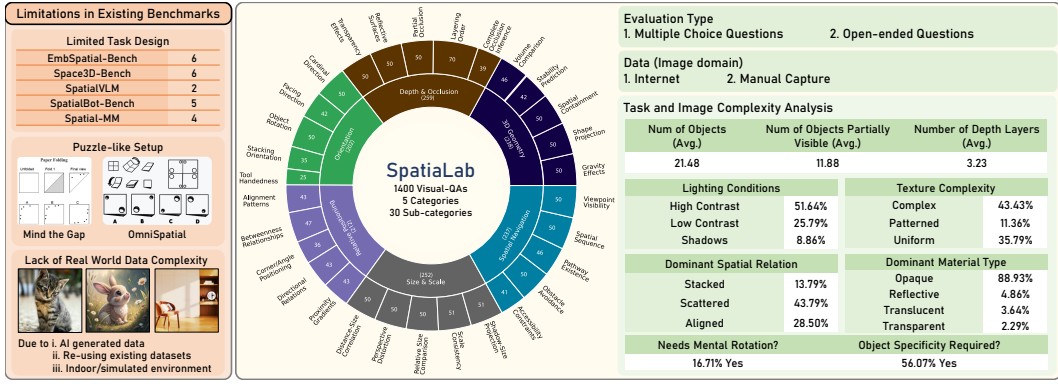

Figure 1: **Overview of SPATIALAB.** The benchmark addresses limitations of prior datasets (left), introduces 1,400 visual QA pairs spanning 5 categories and 30 subcategories (center), and enables systematic evaluation through multiple-choice and open-ended tasks. It features diverse task and image complexity, with varied object counts, layers, lighting, textures, relations, and materials (right).

Existing spatial benchmarks, however, are narrow and simplified. Most focus on elementary tasks such as binary spatial relations, low-resolution depth categories, or schematic navigation cues, often in synthetic or simplified settings that reduce visual clutter and homogenize objects (Du et al., 2024; Jia et al., 2025; Szymanska et al., 2024; Shiri et al., 2024; Kamath et al., 2023; Fu et al., 2024; Song et al., 2024; Yang et al., 2024; Qi et al., 2025; Cai et al., 2024). These controlled setups lessen perceptual and reasoning demands, leading to apparent saturation while masking failures under distribution shift. Critical challenges such as robust occlusion inference, scale-consistent judgments across viewpoints, and planning under partial observability remain under-sampled (Kosoy et al., 2025; Pothiraj et al., 2025; Chen et al., 2025; Cai et al., 2025). As a result, models that perform well on existing benchmarks often fail on integrated, multi-step inference tasks, limiting safe deployment in real-world environments where spatial errors carry practical consequences.

A meaningful evaluation must therefore span the all core axes of spatial reasoning: relative position, depth and occlusion, orientation, size and scale, navigation, and 3D geometry. Unlike current task-isolated setups, human cognition integrates these dimensions seamlessly under noise and ambiguity (Driess et al., 2023; Brohan et al., 2023; Zitkovich et al., 2023; Collaboration et al., 2023; Yuan et al., 2024; Qi et al., 2025). Robust benchmarks should therefore test both choice and generation, benchmark against human performance, and draw from real-world scenes that reveal failures masked by simulator datasets-principles that motivate our benchmark.

To address the above needs, we introduce **SPATIALAB**, a benchmark designed to assess spatial reasoning of vision–language models in realistic, unconstrained visual contexts, as outlined in Figure 1. **SPATIALAB** provides a balanced suite of visual question–answer items covering a broad taxonomy of spatial tasks and supports both multiple-choice and open-ended evaluation formats. The benchmark emphasizes photographic diversity, complex spatial relational queries, and tasks that require integrating depth, perspective, and 3D structure. We evaluate a wide collection of state-of-the-art VLMs, including open- and closed-source models and reasoning-focused systems, compare them to human performance, and perform extensive error analysis. Our key contributions are as follows:

1. We introduce **SPATIALAB**, a large-scale benchmark of 1,400 visual question–answer pairs spanning six principal categories and thirty task types in realistic, unconstrained contexts. With at least 200 questions per category and 25 per subcategory, it ensures balanced coverage and supports both multiple-choice (SPATIALAB-MCQ) and open-ended (SPATIALAB-OPEN) evaluation.

2. We perform a **comprehensive evaluation** of 25+ state-of-the-art vision–language models: open- and closed-source, reasoning-oriented, and spatially specialized reasoning models against human baselines, exposing a consistent and significant gap in spatial reasoning.

3. We deliver **in-depth error analyses and diagnostic studies** that uncover systematic failure modes in depth perception, navigation, occlusion, and 3D geometry, yielding concrete insights and actionable directions for advancing robust, human-aligned spatial understanding.

Table 1: Comparison of spatial reasoning benchmarks.

| Dataset | E. | T.C. | Data | | | | QAs | | | Venue/ | Eval | | A.L. | Md. | Best Baseline | B.B.P. |
|---|---|---|---|---|---|---|---|---|---|---|---|---|---|---|---|---|
| | | | Domain | Source | A. | Size | Count | Df. | Cmpl. | Year | Type | Depth | | | | |
| ScanQA | Y | 7 | Indoor | E.D. | M | 800 | 41K | H | E | CVPR'22C | Close-ended | L | L | I | - | - |
| Visual Spatial | Y | 7 | R.W. | E.D. | T | 10K | 10K | E | E | TACL'23 | MCQ,Bin. | L | M | I | LXMERT | 70.10 |
| What's up | N | 6 | Indoor | E.D. | T | 5K | 5K | E | E | EMNLP'23C | MCQ | S | Hi | I | XVLM-COCO | 60.40 |
| EmbSpatial-Bench | Y | 6 | Indoor | E.D. | T | 2.2K | 3.6K | H | E | ACL'24C | MCQ,% | M | S | I | Qwen-VL-Max | 49.11 |
| Space3D-Bench | N | 6 | Indoor | E.D. | M | 211 | 1K | Mm | E | ECCV'24W | Mix | L | S | I | RAG3D-Chat | 66.80 |
| SpatialRGPT-Bench | N | 12 | Mix | E.D. | T | 1.5K | 1.5K | Mm | E | NeurIPS'24C | MCQ,Bin. | M | L | I | GPT-4o | 57.83 |
| BLINK-Spatial | Y | 14 | Mix | E.D. | M | 7.3K | 3.8K | H | E | ECCV'24C | MCQ,Bin. | M | M | I | GPT-4o | 59.03 |
| Spatial-MM | N | 4 | Internet | Internet | T | 2.3K | 2.3K | E | Mm | EMNLP'24C | Open | L | Hi | I | GPT-4 Vision | 63.82 |
| RoboSpatial | Y | 4 | Indoor | E.D. | T | 1M | 3M | H | E | CVPR'25C | Open,Bin. | L | S | I | LEO | 71.90 |
| SpatialVLM | N | 2 | Web | E.D. | T | 10M | 2B | H | E | CVPR'24C | Open | L | S | I | GPT-4V | 68.00 |
| VSI-Bench | Y | 8 | Indoor | M,E.D. | T | 288 | 5K | Mm | Mm | CVPR'25C | MCQ,Nm. | M | M | V | Gemini-1.5 Pro | 45.40 |
| OmniSpatial | Y | 50 | Internet | Internet | M,P | 1387 | 1.5K | H | Mm | 2025 | MCQ,Bin. | M | S | I | o3-2025-04-16 | 56.33 |
| Mind the Gap | N | 6 | Mix | AI-Gen | M,P | 1.8K | 1.8K | E | Mm | 2025 | Mix | M | S | I | InternVL2.5-26B | 48.83 |
| VLM4D | N | 4 | Mix | Mix | M | 1K | 1.8K | E | H | ICCV'25 | MCQ | Hi | M | Mix | Gemini-2.5 Pro | 62.00 |
| **SPATIALAB-MCQ** | Y | 30 | Mix | Mix,M | M | 1.2K+ | 1.5K | H | H | 2025 | MCQ | Hi | H | Mix | InternVL3.5-72B | 54.93 |
| **SPATIALAB-OPEN** | Y | 30 | Mix | Mix,M | M | 1.2K+ | 1.5K | H | VH | 2025 | Open | Hi | H | Mix | GPT-5-mini | 40.93 |

4. We explore multiple strategies to enhance spatial reasoning in VLMs, including **supervised fine-tuning (SFT), chain-of-thought (CoT) prompting, CoT with self-reflection, and multi-agent architectures (SPATIOXOLVER) with multi-step reasoning**, analyzing their benefits and limitations across multiple-choice and open-ended tasks.

In the multiple-choice setting, leading models achieve roughly 50–55% accuracy, while human annotators exceed 85%. In open-ended generation, model performance drops by another 10–25 percentage points, with the best systems near 40% versus about 65% for humans. Error analysis reveals concentrated failures in occlusion inference, scale-consistency, and multi-step navigational planning, indicating limits in compositional spatial representations and reasoning. Our comprehensive study highlights which interventions provide robust gains, which amplify existing biases, and the critical role of perceptual grounding for generalizable spatial reasoning. By providing tasks, baseline results, and diagnostic analyses, SPATIALAB establishes a foundation for progress and reproducible comparison in spatial reasoning evaluation.

## 2 PRELIMINARIES: SPATIAL REASONING IN VLMS

Visual–spatial reasoning, the ability to perceive, represent, and manipulate spatial relations among objects, plays a central role in human cognition and in enabling embodied agents to navigate and act in the world (Palmer, 1999; Chabris et al., 2006). Formally, we can describe a scene as a set of objects $O = \{o_1, \ldots, o_n\}$ with attributes (e.g., position, orientation, size) in a world coordinate system $W$. Visual–spatial reasoning then seeks to learn a mapping $f : (I, Q) \mapsto R$, where $I$ is an image (or image sequence), $Q$ is a linguistic query, and $R$ is a structured representation of spatial relations $R = \{r(o_i, o_j) \mid o_i, o_j \in O\}$ capturing predicates such as *left-of*, *occludes*, or *supports*. For vision–language models (VLMs), learning $f$ is highly challenging because it requires integrating noisy visual perception with compositional linguistic abstraction under embodied constraints such as perspective, gravity, and physical interaction (Dai et al., 2017; Alayrac et al., 2022).

Early approaches to spatial reasoning in VLMs have largely relied on basic spatial relations or narrow task designs (Cheng et al., 2024; Du et al., 2024; Szymanska et al., 2024), often situated in synthetic datasets like CLEVR (Johnson et al., 2016) or structured question–answering benchmarks such as GQA (Hudson & Manning, 2019) or puzzle-like simulated setups such as MGT (Stogiannidis et al., 2025) and OmniSpatial (Stogiannidis et al., 2025), as we can notice in Table 1[1]. While useful for controlled evaluation, these settings oversimplify spatial structure, omit environmental noise, and neglect dynamic factors such as temporal changes or viewpoint shifts. To overcome these limitations, we argue for the need to benchmark spatial reasoning in rich, real-world contexts that capture scene complexity, temporal variation, and multiple perspectives. Such settings expose VLMs to challenges they would encounter in embodied or interactive applications, including robotics, AR/VR, and autonomous navigation (Mu et al., 2023; Collaboration et al., 2023).

In SPATIALAB, we organize spatial reasoning into six principal categories, *Relative Positioning, Depth & Occlusion, Orientation, Size & Scale, Spatial Navigation, and 3D Geometry*, each linked to a distinct cognitive faculty studied in psychology and spatial cognition (Baddeley, 1998; Previc, 1998).

---

[1]Here, E. → Embodied, R.W. → Real World, T.C. → Task Categories, E.D. → Existing Dataset, A. → Annotation, T → Template, M → Manual, P → Puzzles, Md. → Modality (I → Image,V → Video), Bin. → Bin., Nm. → Nm., Df. → Difficulty, Cmpl. → Complexity (E → Easy,Mm → Medium, H → Hard, S → Shallow, L → Low, Hi → Hi, VH/VHi → Very Hard/Very High), A.L. → Analysis Level, B.B.P. → Best Baseline Performance.

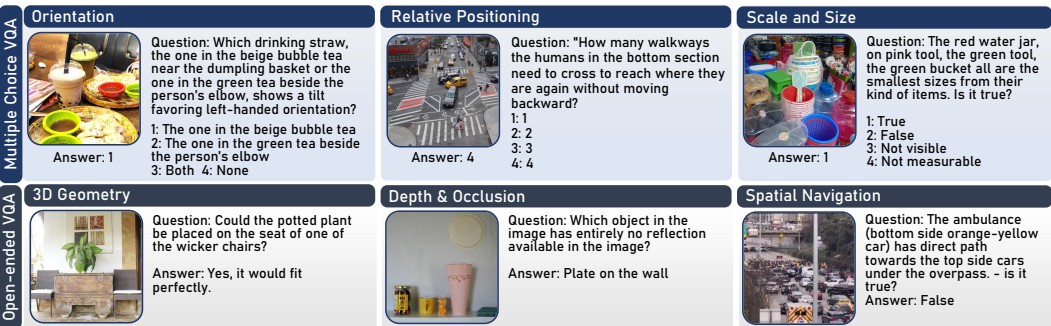

Figure 2: Representative examples from six categories in open-ended and MCQ Tasks.

As shown in Figure 1 and Appendix B, each category is further decomposed into five subcategories, producing a taxonomy of thirty task types that isolate fine-grained reasoning skills, such as directional alignment, occlusion inference, or stability prediction. This classification scheme is motivated by the need to capture the full spectrum of visual–spatial reasoning challenges. By systematically breaking down complex reasoning into measurable and testable units, we can evaluate not only whether models "solve" spatial tasks but also which specific dimensions remain fragile or underdeveloped (Gardner, 2011; Huang et al., 2022; Gupta & Kembhavi, 2023; Yang et al., 2023).

Moreover, we adopt two complementary evaluation formats: multiple-choice and open-ended question answering. The multiple-choice format probes recognition and fine discrimination among distractors, while the open-ended format measures generative reasoning and compositional description (Driess et al., 2023; Liu et al., 2025; Zheng et al., 2022). Together, these formats reflect real-world reasoning demands and expose different strengths and weaknesses in model capabilities. By combining a principled taxonomy, diverse real-world imagery, and dual evaluation formats, SPATIALAB provides a structured framework for systematic benchmarking. This setup enables fine-grained model comparison, alignment with human-level performance, and deeper insights into the development of robust spatial cognition in artificial agents.

## 3 SPATIALAB BENCHMARK

Spatial reasoning remains a persistent challenge for vision–language models (VLMs), especially in naturalistic conditions where occlusion, perspective distortion, and noisy backgrounds co-occur. To address this, we present **SPATIALAB**, a benchmark designed to systematically evaluate VLMs' capacity for complex spatial cognition under realistic visual conditions. The benchmark comprises 1,400 visual question–answer pairs organized into six major categories: *Relative Positioning, Depth & Occlusion, Orientation, Size & Scale, Spatial Navigation, and 3D Geometry*, each further divided into five subcategories, yielding 30 distinct task types. Every subcategory includes at least 25 QA pairs, while each top-level category contributes a minimum of 200, ensuring balanced coverage across the taxonomy. **SPATIALAB** supports both multiple-choice and open-ended evaluation, enabling assessment of discriminative accuracy as well as generative reasoning and explanatory competence, thus aligning with broader multimodal evaluation practices while emphasizing spatial-specific reasoning. High visual diversity, fine-grained annotation, and rigorous quality control make **SPATIALAB** a reliable and scalable testbed, and Figure 2 illustrates representative examples from the six categories, showcasing both open-ended and multiple-choice formats that capture the richness and difficulty of real-world spatial reasoning tasks.

### 3.1 BENCHMARK CONSTRUCTION

**Image Collection.** The image corpus was curated using three complementary strategies: (i) automated large-scale web crawling, (ii) targeted online retrieval of scene-specific images, and (iii) manual snapshots collected in diverse indoor and outdoor environments (Figure 3). This multi-source pipeline ensures diversity across object categories, environmental conditions, and spatial configurations. To explicitly capture natural variability, images were profiled along six meta-dimensions: *lighting condition* (high contrast, low contrast, shadows, and reflective settings), *texture complexity* (uniform, patterned, and complex), *edge complexity* (sharp and smooth boundaries), *dominant spatial relation*

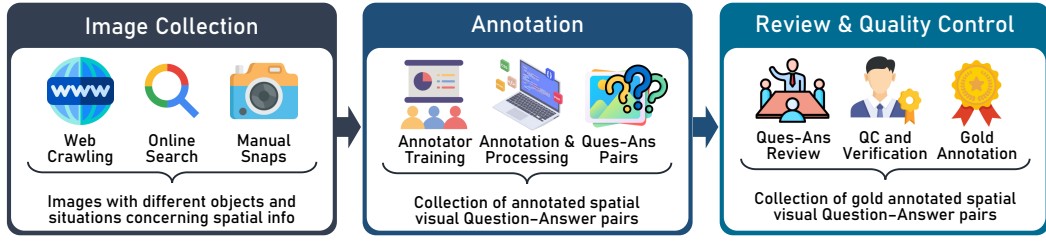

Figure 3: **Data creation pipeline for SPATIALAB.** Images are collected via web crawling, targeted search, and manual snapshots, followed by structured annotation of spatial question–answer pairs.

(stacked, scattered, aligned), *material type* (transparent, translucent, opaque, reflective), and *gravity constraints* (normal, floating, and unconstrained). Each dimension was systematically represented in the final corpus, ensuring that the benchmark reflects a broad spectrum of real-world, in-the-wild scenarios rather than simplified or synthetic setups. Complexity analysis of the corpus reveals that on average images contain 21.48 objects, of which 11.88 are partially visible, distributed across 3.23 distinct depth layers. Spatial reference chains (e.g., "object A is left of B, which is behind C") contain 2.07 links, indicating the presence of multi-step relational reasoning. Also, 16.71% of the questions require mental rotation; 56.07% require object specificity (specific color, pattern, or other details). Such statistics posits that SPATIALAB captures cluttered, layered, and relationally rich scenes, which are characteristic of real-world spatial reasoning, underrepresented in synthetic datasets.

**Annotation.** Annotation proceeded in three phases. First, annotators underwent targeted training (derails are available in Appendix C.2) to ensure consistency in interpreting spatial categories and subcategories. Second, each image was paired with one or more spatial QA tasks covering a range of reasoning types such as proximity gradients, alignment patterns, occlusion inference, and stability prediction. Questions were designed to balance perceptual grounding with higher-order inference (e.g., predicting stability under gravity constraints). Third, QA pairs were encoded in both multiple-choice (04 options) and open-ended formats, ensuring comparability across evaluation paradigms. In total, it produces around 1,500 spatial visual QA pairs, across six categories and 30 subcategories.

**Review and Quality Control.** To ensure reliability, annotation outputs underwent a three-tier review and validation pipeline. In the first stage, QA items were checked for semantic validity and alignment with the designated subcategory. In the second stage, independent annotators verified correctness of both the question phrasing and the designated answer, focusing on eliminating ambiguity and enforcing task clarity. In the third stage, a gold-standard annotation round was conducted, establishing a set of high-confidence QA pairs for evaluation. All metadata was removed from the images, and the dataset was carefully curated to avoid copyright concerns, though minor oversights may still occur. This procedure yielded a final benchmark of 1,400 validated QA items.

## 4 EXPERIMENTS AND EVALUATION

**Models.** We categorize the evaluated models into four groups. **Proprietary models** include GPT-4o-mini (OpenAI, 2024), GPT-5-nano, Gemini-2-Flash and Gemini-2.5-Flash (Team, 2025a), and Claude 3.5 Haiku (Anthropic, 2024). **Open-source models** comprise Intern-VL3 (1B, 2B, 4B) (Zhu et al., 2025), Qwen-VL2.5 (3B, 7B, 32B, 72B Instruct) (Bai et al., 2025), GLM 4.5V-3.06B and GLM-4.1V-9B-Thinking (Team et al., 2025), Gemma-3 (4B, 27B)-it (Gemma Team, Google DeepMind, 2025), Llama-3.2 (11B, 90B) Vision-Instruct (Grattafiori et al., 2024), and Step-3-321B-MoE (StepFun, 2025). **Reasoning models** include o4-mini (OpenAI, 2025), Gemini-2-Flash and Gemini-2.5-Flash thinking variants (Team, 2025a), and Kimi-VL-A3B-Thinking-2506 (Team, 2025b). **Spatial reasoning specialists** comprise SpaceOm, SpaceThinker-Qwen2.5VL-3B, and SpaceQwen2.5-VL-3B-Instruct (Chen et al., 2024a). For GLM-4.5V-106B-MoE, GLM-4.1V-9B-Thinking, and Step-3-321B-MoE we only performed open-ended evaluation, as these models are not instruction-tuned and were trained primarily on open-ended tasks, resulting in excessive refusals during MCQ evaluation. For ease of comparison in our results tables, we use color coding: proprietary , open-source up to 7B , larger than 7B , proprietary reasoning , open-source reasoning , spatial specialists , and human baseline .

Table 2: Multiple Choice Evaluation Accuracy (%)(↑) on SPATIALAB-MCQ by Question Categories.

| Model | 3D Geom. (#238) | Dep. & Occu. (#259) | Orientation (#202) | Relat. Posit. (#212) | Size & Scale (#252) | Spati. Navig. (#237) | Overall (#1400) |
|---|---|---|---|---|---|---|---|
| *Random Choice* | 25.00 | 25.00 | 25.00 | 25.00 | 25.00 | 25.00 | 25.00 |
| *Proprietary Models* | | | | | | | |
| GPT-4o-mini | 47.06 | 39.00 | 47.03 | 47.17 | 49.60 | 49.79 | 46.50 |
| GPT-5-mini | **48.74** | 54.83 | **60.40** | **62.74** | 44.84 | 56.54 | **54.29** |
| Gemini-2.0-Flash | 47.06 | **55.21** | 53.96 | 58.02 | **54.37** | 46.84 | 52.50 |
| Gemini-2.5-Flash | 44.96 | 48.26 | 48.02 | 56.13 | 42.46 | 51.05 | 48.29 |
| Gemini-2.5-Pro | 47.48 | 50.19 | 49.50 | 58.49 | 43.65 | 52.32 | 50.07 |
| Claude 3.5 Haiku | 42.44 | 42.08 | 46.53 | 46.23 | 35.71 | 45.99 | 42.93 |
| Mistral Medium 3.1 | 46.64 | 49.81 | 47.52 | 61.79 | 41.67 | 41.77 | 47.93 |
| *Open-Source Models* | | | | | | | |
| InternVL3.5-1B | 33.61 | 32.43 | 23.27 | 37.26 | 31.75 | 30.80 | 31.64 |
| InternVL3.5-2B | 34.03 | 31.66 | 31.68 | 40.57 | 32.54 | 32.49 | 33.71 |
| Qwen-VL2.5-3B-Instruct | 41.18 | 35.52 | 46.04 | 40.09 | **47.22** | 39.24 | 41.43 |
| InternVL3.5-4B | 42.86 | **42.86** | 42.08 | **54.72** | 36.51 | **42.19** | **43.29** |
| Gemma-3-4B-it | **43.70** | 34.36 | **46.53** | 45.75 | 37.30 | 37.97 | 40.57 |
| Qwen-VL2.5-7B-Instruct | 42.86 | 37.84 | 42.57 | 46.23 | 42.06 | 35.44 | 41.00 |
| Llama-3.2-11B-Vision-Instruct | 26.47 | 30.50 | 20.30 | 42.92 | 30.56 | 32.07 | 30.50 |
| Gemma-3-27B-it | 43.28 | 40.15 | 48.02 | 54.25 | 48.02 | 47.26 | 46.57 |
| Qwen-VL2.5-32B-Instruct | 41.18 | 40.15 | 46.53 | 45.28 | 45.24 | 41.77 | 43.21 |
| InternVL3.5-72B | **50.00** | **57.14** | **53.47** | **66.04** | **49.21** | **54.85** | **54.93** |
| Qwen-VL2.5-72B-Instruct | 47.06 | 48.65 | 51.98 | 54.25 | 43.65 | 48.95 | 48.86 |
| Llama-3.2-90B-Vision-Instruct | 46.22 | 52.12 | 50.50 | 58.96 | 46.83 | 48.52 | 50.36 |
| *Reasoning Models* | | | | | | | |
| o3-mini | 50.00 | 54.83 | 51.98 | 61.32 | 38.89 | 50.21 | 50.93 |
| o4-mini-medium | **51.26** | **58.30** | **54.95** | **64.15** | 40.87 | 51.48 | **53.21** |
| Gemini-2-Flash-Thinking | 37.82 | 41.31 | 41.58 | 45.75 | 50.40 | 43.04 | 43.36 |
| Gemini-2.5-Flash-Thinking | 45.80 | 53.67 | 52.97 | 56.60 | **55.16** | **53.59** | 52.93 |
| Kimi-VL-A3B-Thinking-2506 | 42.86 | 41.31 | 40.59 | 51.42 | 39.68 | 41.35 | 42.71 |
| *Spatial Reasoning Models* | | | | | | | |
| SpaceOm | **42.44** | **38.61** | **48.02** | 37.74 | 42.86 | 39.24 | **41.36** |
| SpaceThinker-Qwen2.5VL-3B | 40.34 | 37.84 | 47.03 | **38.21** | 43.25 | 37.97 | 40.64 |
| SpaceQwen2.5-VL-3B-Instruct | 31.51 | 35.14 | 37.62 | 37.74 | **50.79** | **47.26** | 40.14 |
| *Human Baseline* | 93.70 | 74.13 | 91.58 | 91.51 | 88.89 | 87.76 | 87.57 |

**Evaluation Process and Metrics.** We measure accuracy as the primary metric for both MCQ and open-ended tasks. For **MCQ evaluation**, we use direct prompting, where models return the option number corresponding to their selected answer. Responses are automatically checked against the correct answer. Detailed prompts and details are provided in Appendix D.1. For **open-ended evaluation**, models are prompted to generate free-form answers, which are then assessed using a large language model judge. Prompts and details are available Appendix D.2. To evaluate the performance of LLM Judge with humans, we performed LLM-human agreement analysis too, as detailed in Appendix D.2. Table 5 demonstrates that the LLM judge (Gemini-2.5-Flash) achieves substantial agreement with human annotators, with a Cohen's kappa of 0.738 against the majority vote and 0.681–0.795 against individual annotators. Raw accuracy against the majority vote is 0.880, while Fleiss' kappa among human annotators is 0.774, indicating strong reliability. For additional details on experiment setup, prompts, and evaluation procedures, see Appendix D.

**Improving Visual Reasoning Capabilities.** To enhance spatial reasoning on our diverse benchmark, we explore reasoning strategies including inherent reasoning in the VLMs (Appendix H.1), Chain-of-Thoughts (CoT) (Appendix H.2), CoT with self-reflection (Appendix H.3), supervised fine-tuning (SFT) (Appendix H.4), and multi-agent systems (Appendix H.5). For SFT, specifically, Qwen-VL2.5-3B-Instruct was fine-tuned on 40% of the dataset with stratified sampling, improving generalization across relational and geometric tasks. Evaluation on the remaining 60% showed reduced errors in navigation and orientation, with additive gains when integrating CoT and SFT. However, agentic reasoning revealed that while orientation benefits substantially, other categories stagnate or degrade, underscoring that deeper perceptual and spatial grounding is required beyond multi-step reasoning.

## 5 FINDINGS AND ANALYSIS

### 5.1 OVERALL RESULTS

**SPATIALAB-MCQ.** Table 2 demonstrates model performances in multiple choice evaluation on SPATIALAB-MCQ. We observe substantial variation in performance across models, tasks, and families: overall model accuracies span roughly 30–55% (random choice = 25%), while the human baseline is 87.57%, indicating large headroom. InternVL3.5-72B, an open source model, is the best

Table 3: Open-ended Evaluation Accuracy (%)(↑) on SPATIALAB-OPEN by Question Categories.

| Model | 3D Geom. (#238) | Dep. & Occu. (#259) | Orientation (#202) | Relat. Posit. (#212) | Size & Scale (#252) | Spati. Navig. (#237) | Overall (#1400) |
|---|---|---|---|---|---|---|---|
| *Proprietary Models* | | | | | | | |
| GPT-4o-mini | 23.53 | 16.60 | 23.27 | 30.66 | 17.86 | 21.94 | 26.00 |
| GPT-5-mini | **45.38** | **34.75** | **37.13** | **49.53** | **42.46** | **37.13** | **40.93** |
| Gemini-2.0-Flash | 31.93 | 24.32 | 27.23 | 31.13 | 26.19 | 24.47 | 27.43 |
| Gemini-2.5-Flash | 34.03 | 26.64 | 31.68 | 38.68 | 26.59 | 29.54 | 30.93 |
| Gemini-2.5-Pro | 37.14 | 45.45 | 36.36 | 37.14 | 23.91 | 24.44 | 33.61 |
| Claude 3.5 Haiku | 26.05 | 18.92 | 24.75 | 25.94 | 20.24 | 21.10 | 22.64 |
| Mistral Medium 3.1 | 25.21 | 19.31 | 21.78 | 29.25 | 15.08 | 16.88 | 21.00 |
| *Open-Source Models* | | | | | | | |
| InternVL3.5-1B | 05.88 | 09.65 | 9.90 | 13.68 | 09.13 | 10.13 | 09.64 |
| InternVL3.5-2B | 12.18 | 11.20 | 10.89 | 23.58 | 11.90 | 18.14 | 14.50 |
| Qwen-VL2.5-3B-Instruct | 15.55 | 8.49 | 15.35 | 10.85 | **18.25** | 9.28 | 12.93 |
| InternVL3.5-4B | 19.33 | **17.76** | 15.84 | 19.81 | 16.27 | 18.99 | 18.00 |
| Gemma-3-4B-it | **20.17** | 13.13 | 14.85 | 23.58 | 15.08 | **19.83** | 17.64 |
| Qwen-VL2.5-7B-Instruct | 15.13 | 15.83 | **20.30** | 27.83 | 15.87 | **19.83** | **18.86** |
| Llama-3.2-11B-Vision-Instruct | 16.81 | 16.99 | 22.28 | 25.00 | 13.49 | 18.57 | 18.57 |
| Gemma-3-27B-it | 22.69 | 16.22 | 24.75 | **34.43** | 22.62 | 21.94 | 23.43 |
| Qwen-VL2.5-32B-Instruct | 16.39 | 09.65 | 14.85 | 16.98 | 09.92 | 13.50 | 13.36 |
| InternVL3.5-72B | 22.69 | 20.46 | 20.30 | 31.60 | 19.84 | 26.16 | 23.36 |
| Qwen-VL2.5-72B-Instruct | 26.89 | 20.85 | **25.25** | 30.66 | **24.60** | 20.68 | 24.64 |
| Llama-3.2-90B-Vision-Instruct | 22.69 | **23.17** | 21.29 | 28.30 | 21.83 | **27.00** | 24.00 |
| GLM-4.5V-106B-MoE | **31.09** | 20.46 | 25.25 | 26.42 | 24.21 | 24.47 | **25.21** |
| *Reasoning Models* | | | | | | | |
| o3-mini | 39.08 | 30.12 | 29.70 | 39.15 | 41.27 | 30.38 | 35.00 |
| o4-mini-medium | **40.76** | 32.82 | 32.18 | **42.92** | **44.05** | **34.18** | **37.86** |
| Gemini-2-Flash-Thinking | 31.09 | 27.41 | 31.19 | 34.43 | 29.37 | 29.54 | 30.36 |
| Gemini-2.5-Flash-Thinking | 37.14 | **45.45** | **36.36** | 37.14 | 21.74 | 22.22 | 32.77 |
| Kimi-VL-A3B-Thinking-2506 | 13.45 | 14.29 | 11.88 | 17.92 | 12.70 | 18.57 | 14.79 |
| GLM-4.1V-9B-Thinking | 21.43 | **19.31** | 26.73 | 24.53 | 19.84 | 22.78 | 22.21 |
| Step-3-321B-MoE | **32.35** | 18.53 | **27.72** | **27.36** | **29.76** | **27.85** | **27.14** |
| *Spatial Reasoning Models* | | | | | | | |
| SpaceOm | 12.61 | 06.95 | 15.84 | **11.79** | 18.65 | **12.24** | 12.93 |
| SpaceThinker-Qwen2.5VL-3B | **13.45** | **09.27** | **17.82** | 10.38 | **19.44** | 10.13 | **13.36** |
| SpaceQwen2.5-VL-3B-Instruct | 12.61 | 03.86 | 13.86 | 09.43 | 11.90 | 11.39 | 10.36 |
| *Human Baseline* | 73.53 | 50.19 | 70.30 | 69.81 | 65.48 | 62.87 | 64.93 |

performing model, with 54.93% accuracy. Reasoning- and instruction-tuned models (e.g., o4-mini: 53.21%, Gemini-2.5-Flash-Thinking: 52.93%, GPT-5-mini: 54.29%) and some very large open-source models (e.g., InternVL3.5-72B: 54.93%, Llama-3.2-90B: 50.36%) rank near the top, but model scale alone is not determinative: for example, Llama-3.2-11B attains only 30.50% despite its substantial size. Category-wise, Orientation and 3D Geometry are where many strong models excel (several models exceed 60% on orientation or 3D geometry), whereas Spatial Navigation, Depth & Occlusion, and Size & Scale are more variable and generally harder across the board. Notably, the spatially specialized models (SpaceOm, SpaceThinker, SpaceQwen) achieve only mid-40s to low-40s overall, so specialization does not automatically translate to higher MCQ accuracy across our diverse question set. Depth & Occlusion and Size & Scale show mixed results, some architectures handle occlusion relatively well while others fail, pointing to differing inductive biases or training data. These patterns imply that instruction-tuning and reasoning capability particularly help geometric and orientation reasoning, while multi-step grounding tasks like navigation remain an open weakness.

**SPATIALAB-OPEN**. Open-ended evaluation scores presented in Table 3 are substantially lower than MCQ results and show a wide spread: overall accuracy ranges from about 9.6% (InternVL3.5-1B) up to 40.9% (GPT-5-mini), while the human baseline sits at 64.9%, indicating a large gap to close. Proprietary models generally lead the leaderboard on open-ended tasks (e.g., GPT-5-mini 40.93%, GPT-4o-mini 26.00%, Gemini-2.5-Flash 30.93%), but reasoning-tuned variants stand out in particular, the o4-mini (37.86%) and Gemini-2.5-Flash-Thinking (32.77%) perform noticeably better than many baseline open-source counterparts, suggesting instruction/CoT-style tuning substantially improves generative outputs. Most open-source small models cluster at the bottom (many in the teens; InternVL3.5-1B = 9.64%, Qwen-VL2.5-3B = 12.93%), whereas some very large open models improve to the low-20s (e.g., InternVL3.5-72B = 23.36%, Qwen-VL2.5-72B = 24.64%), showing that scale alone yields only modest gains. Spatial-specialist models (SpaceOm, SpaceThinker, SpaceQwen) attain only 10–13% overall, implying that architecture or task specialization does not automatically translate to robust open-ended generation across our diverse question set. Across subtasks we observe relatively better performance on orientation and certain relative-position items for top models (e.g., GPT-5-mini Rel.Pos = 49.53%, Ori = 37.13%), while depth & occlusion, size

& scale, and especially spatial navigation remain consistently hard (many models score <30% on these subtasks). This pattern reinforces the earlier finding that multi-step grounding and perceptual reasoning are the bottlenecks for open generative answers.

## 5.2 PERFORMANCE DROP IN OPEN-ENDED EVALUATION COMPARED TO MCQ

Across 25 models, the average MCQ→open-ended performance gap in SPATIALAB is 23.0% ($\sigma = 5.5\%$), with subtask means ranging from 22.89% (spatial navigation) to 24.57% (3D geometry). Specialist spatial reasoning models exhibit the largest gaps (about 27%), particularly in spatial navigation (up to 36.68%) and orientation (34.44%), while reasoning-oriented models achieve smaller gaps (around 19%) and lower subtask variance, reflecting the stabilizing effect of instruction-tuning and CoT decoding. Correlations confirm that spatial navigation dominates overall disparity (Pearson $r = 0.99$), with orientation ($r = 0.83$) and 3D geometry ($r = 0.79$) also contributing. Negative or near-zero gaps (e.g., Llama-3.2-11B: -1.98 in depth & occlusion; o4-mini: -3.18 in relative position) indicate that poorly designed MCQ distractors can misrepresent true competence. We hypothesize that format sensitivity arises from MCQ's structural advantage, specialization bias toward categorical tasks, sequential reasoning challenges in spatio-navigation, and calibration differences in generative output. These findings suggest that MCQ alone can overestimate practical spatial reasoning ability. We recommend complementing MCQ with open-ended evaluations, auditing distractors, employing stepwise generation or instruction-tuning, and reporting per-subtask diagnostics to capture true model performance and reduce format-dependent artifacts. More details are in Appendix G and Table 18.

## 5.3 ERROR ANALYSIS

**SPATIALAB-MCQ**. Our analysis (Appendix E) shows that VLMs achieve selective peaks but lack holistic spatial competence. Closed-source models reach the highest scores (e.g., 85.71% in *Stacking Orientation*), yet collapse to near-chance in *Relative Size Comparison*. Open-source scaling improves ceilings (80.56% in *Corner/Angle Positioning*) but does not prevent catastrophic failures (2.0% in *Object Rotation*). Reasoning-augmented and spatially specialized models add localized gains, such as in occlusion inference or navigation, but remain capped below 55% in many physical abstraction tasks (*Gravity Effects*, *Stability Prediction*). Error distributions further reveal systematic weaknesses in embodied reasoning (*Tool Handedness*), recursive relational chaining (*Pathway Existence*), and non-local cues (*Reflective Surfaces*), while stronger results in *Obstacle Avoidance* suggest shortcut exploitation rather than robust planning. Taken together, these patterns confirm that current VLMs rely heavily on surface-level correlations and lack stable encodings of orientation, physics, and compositional logic. Progress will likely require both richer training distributions (e.g., embodied and physics-driven data) and architectural mechanisms for geometric and reference-frame grounding.

**SPATIALAB-OPEN.** Error analysis of SPATIALAB-OPEN (Appendix F) reveals that even top-performing closed-source models (e.g., GPT-5-mini: 58.14% in *Directional Relations*) collapse on tasks like *Proximity Gradients*, indicating that dominance remains task-specific. Large open-source models achieve strong results in *Depth & Occlusion* (e.g., 60.0%) but still fall to 9.3% on *Proximity*, showing that scale alone cannot ensure robustness. Smaller models exhibit isolated strengths (e.g., 26.0% in *Relative Size Comparison*) but often break down completely (e.g., 0.0% in *Betweenness*), reflecting the limitations of compact architectures. Reasoning-tuned systems show promise, Gemini-2.5-Flash-Thinking reaches 75.0% in *Tool Handedness*, yet similar models fall to single digits, revealing fragile integration. Specialized spatial models perform worst overall, rarely exceeding 20% and frequently failing on core relational tasks. Failure patterns highlight systemic weaknesses in occlusion handling, orientation, and multi-step relational chaining, while partial success in size–scale cues suggests reliance on superficial correlations. These results point to three root causes: missing representations for spatial analogies, architectural limits tied to scale and training diversity, and brittle reasoning pipelines underscoring the requirement for a unified geometric encodings, physics-aware reasoning, and embodied data to move from isolated peaks toward consistent spatial competence.

**Qualitative Error Analysis**. To better understand model weaknesses, we conducted a qualitative error analysis (see Appendix I) and found that failures cluster into a small number of recurring classes rather than being random noise. Common errors include spatial mislocalization, where models confuse referents in crowded scenes; perspective and scale mistakes, where reliance on object-size priors overrides image-based cues; and occlusion or ordering failures, especially with thin or

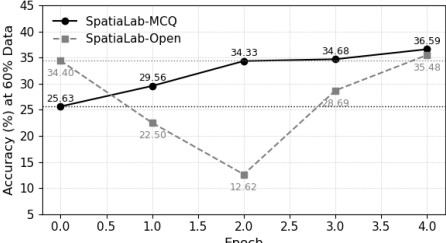

(a) SFT performance trends over epochs.

| Category | MCQ | | | Open-ended | | |
|---|---|---|---|---|---|---|
| | Before | After | Gain | Before | After | Gain |
| 3D Geometry | 30.07 | 37.76 | 7.69 | 20.28 | 23.78 | 3.50 |
| Depth & Occlusion | 23.87 | 32.90 | 9.03 | 43.23 | 45.81 | 2.58 |
| Orientation | 23.33 | 36.67 | 13.33 | 39.67 | 42.15 | 2.48 |
| Relative Positioning | 26.19 | 27.78 | 1.59 | 51.18 | 50.39 | −0.79 |
| Size & Scale | 27.63 | 44.74 | 17.11 | 18.42 | 14.47 | −3.95 |
| Spatial Navigation | 22.38 | 38.46 | 16.08 | 36.62 | 39.44 | 2.82 |
| **Grand Total** | 25.63 | 36.59 | 10.97 | 34.40 | 35.48 | 1.07 |

(b) Performance comparison (Before vs After) across categories for MCQ and Open-ended formats.

Figure 4: SFT training results: (a) learning trends over epochs, and (b) final accuracy values.

partially hidden structures. We also observe attribute confusion, such as mixing perceptual properties with functional ones, and open-ended rationalizations that generate fluent but visually ungrounded narratives. These errors are consistent across architectures and prompting styles, suggesting structural biases in current VLMs rather than dataset-specific noise. Importantly, failure rates spike when multiple cues must be fused, such as combining depth ordering with relative size, and confidence calibration is especially poor in open-ended settings. Our diagnostic protocols confirm that models often ignore minimal but decisive visual features, rely on brittle heuristics, and fail to update when counterfactual edits are applied. The root causes point to insufficient object-centric binding, lack of geometric supervision, and training objectives that reward plausibility over grounding. Together, these findings highlight that current VLMs achieve strong coarse perception but struggle with multi-cue integration and grounded reasoning, underscoring the need for geometry-aware supervision, multi-scale feature retention, and verification pipelines.

## 5.4 PERFORMANCE IMPROVEMENT APPROACHES

**Inherent Reasoning Mechanisms.** We notice that reasoning-enabled models consistently outperform their baselines in both MCQ and open-ended formats, with the largest gains in relational and orientation tasks (e.g., +13.1% in Relative Positioning for MCQ). While reasoning helps stabilize open-ended performance, improvements remain uneven across categories (notably declining in Size & Scale), underscoring that reasoning modules boost logical consistency but do not fully solve grounding or scale sensitivity. Detailed per-category comparisons are provided in Appendix H.1.

**Chain-of-Thought (CoT) Prompting.** Across models, CoT prompting provides little benefit and often reduces accuracy (Table 19), with orientation being the only category showing consistent gains. This indicates that while step-by-step reasoning helps with directional alignment, it fails in tasks requiring robust perceptual grounding such as depth, scale, and spatial navigation. Unlike in textual reasoning, CoT here tends to amplify flawed priors rather than correct them, underscoring that solving SPATIALAB requires perceptual understanding beyond logical chaining (details in Appendix H.2).

**Chain-of-Thought (CoT) with Self-Reflection.** Adding self-reflection to CoT prompting, where models are explicitly prompted to review their previous reasoning step-by-step and correct mistakes (see Figure 8 in Appendix for prompt), yields modest benefits in multiple-choice settings, especially for geometry and depth. However, this fails to generalize to open-ended tasks (Appendix H.3). This confirms that reflective reasoning cannot compensate for missing perceptual grounding, highlighting a core limitation of current VLMs.

**Supervised Fine-Tuning (SFT).** As shown in Figure 4, supervised fine-tuning (SFT), conducted on `Qwen2.5-VL-3B-Instruct` using a stratified 40% split of SPATIALAB (∼560 samples), consistently improves MCQ accuracy across all spatial reasoning categories. However, it offers little, and sometimes negative, transfer to open-ended tasks. This divergence suggests overfitting to task-specific answer distributions, reinforcing concerns that SFT often *teaches to the test* rather than fostering generalizable reasoning (Wang et al., 2025). The sharp drop in open-ended performance points to biased internal representations, tuned for categorical discrimination but unstable in generative settings. A likely cause is catastrophic forgetting of linguistic priors during fine-tuning (Huang et al., 2024a), compounded by objectives that favor discrete-choice alignment over spatial reasoning in language. Overall, while SFT enhances performance in structured formats, it risks creating an algorithmic straightjacket, boosting accuracy in constrained tasks while stalling progress in naturalistic spatial reasoning. More details are available in Appendix H.4.

**AI Agents for Spatial Reasoning (SPATIOXOLVER).** To explore agentic capabilities, we employ SPATIOXOLVER, a multi-agent system adapted and extended from Xolver (Hosain et al., 2025) designed to perform structured spatial reasoning on images. This framework decomposes complex visual reasoning into specialized sub-tasks managed by dedicated agents, including modules for object segmentation, attribute extraction, spatial relation mapping, and transformation tracking. By consolidating these outputs into a unified structured representation, the system attempts to propagate low-level perceptual cues through higher-order relational inferences. We implemented the pipeline using `Gemini-2.5-Flash` with low temperature to ensure deterministic outputs during the multi-stage refinement process. Experimental results demonstrate substantial gains in the Orientation category, improving by +8.00% in MCQ and an impressive +36.00% in open-ended evaluation. However, this success is not universal; categories such as Depth & Occlusion and Spatial Navigation suffered severe declines of -24.00% and -12.00% in open evaluation, respectively. These drops indicate that when inherent perceptual priors—such as depth or occlusion cues—are weak, multi-step reasoning tends to reinforce misconceptions rather than correct them. Ultimately, these findings suggest that while agentic workflows enhance tasks reducible to sequential alignment, they cannot compensate for a lack of fundamental perceptual grounding in complex 3D environments. More details are available in Appendix H.5.

## 6 DISCUSSION

Spatial reasoning is central to embodied intelligence, yet most benchmarks rely on synthetic scenes, templated questions, or narrow task scopes that overlook real-world visual variability. We introduce SPATIALAB, a benchmark of 1,400 visual question–answer pairs covering six major spatial categories and 30 subcategories, including 3D geometry, occlusion, and navigation. The benchmark uses both Multiple-Choice and Open-Ended formats to separate option-based pattern use from genuine generative reasoning. Its quality is supported by a multi-stage review process and complexity checks covering lighting, texture, and spatial diversity. Evaluations of more than 25 contemporary models and human baselines show that SPATIALAB functions as a precise diagnostic tool for identifying weaknesses in current vision–language systems.

Detailed error analysis uncovers systematic failures in handling transparency, multi-step navigation, and occlusion, indicating that models often rely on surface-level texture cues rather than internalized 3D geometric representations. While interventions such as Supervised Fine-Tuning (SFT) yielded clear gains in structured MCQ tasks, they simultaneously triggered instability in open-ended reasoning, suggesting a "catastrophic forgetting" of linguistic priors when optimizing for discrete answers. Furthermore, reasoning-based approaches like Chain-of-Thought proved effective for sequential tasks like Orientation but failed to correct fundamental perceptual errors in Depth and Scale, often amplifying hallucinations instead of resolving them. Agentic decompositions showed similar trade-offs, improving specific alignment tasks while degrading performance in holistic scene understanding contexts. These divergent behaviors confirm that scaling laws and reasoning scaffolds alone are insufficient; achieving human-level spatial intelligence will require architectural innovations that explicitly integrate geometric grounding and physics-aware representations.

## 7 CONCLUDING REMARKS

Spatial reasoning remains a core bottleneck for VLMs, particularly in visually complex, real-world settings. SPATIALAB introduces 1,400 diverse QA pairs across six spatial categories and thirty sub-categories, enabling fine-grained evaluation via both MCQ and open-ended formats. Despite applying several performance-improving methods, our experiments reveal substantial gaps between state-of-the-art models and human performance, with open-ended generation showing an additional 10–25% drop relative to multiple-choice accuracy. Error analysis reveals persistent failures in depth, occlusion, 3D geometry, and multi-step navigation, reflecting reliance on superficial cues over grounded spatial representations. SPATIALAB exposes these limitations and provides a structured testbed for diagnosing spatial competence. Our benchmark enables holistic evaluation and provides actionable insights for developing VLMs with human-aligned spatial understanding. Future progress will require richer training signals (e.g., embodied interaction, action-conditioned feedback) and architectures capable of geometric grounding, relational chaining, and consistent multi-scale reasoning.

ETHICS STATEMENT

We have taken multiple steps to ensure ethical compliance throughout the development of this work. All metadata associated with the images was removed to protect privacy, and any personally identifiable features (such as human faces or license plates) were minimized or excluded wherever possible. The dataset was carefully curated to avoid copyright concerns, with diligent attention paid to sourcing and licensing, though minor oversights may occur despite best efforts. The benchmark construction process was designed with transparency and fairness in mind, ensuring that no demographic groups were singled out or negatively represented. We adhered to the ICLR Code of Ethics and confirm that no human subjects or sensitive personal data were involved.

REPRODUCIBILITY STATEMENT

To facilitate reproducibility, we provide a detailed description of the dataset construction process, annotation pipeline, and evaluation protocol in the main text and appendix. All filtering, quality-control, and validation steps are explicitly documented, with clear definitions of task categories and annotation criteria. Hyperparameters, model configurations, and training procedures are reported in the appendix to support consistent replication of our results. SPATIALAB is available at: `https://spatialab-reasoning.github.io/`.

ACKNOWLEDGMENT

We sincerely thank **Mahir Absar Khan** and **Enjamamul Haque Eram** for their assistance with data collection. We also acknowledge the Qatar Computing Research Institute (QCRI) for providing computing resources and technical support.

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

## A    RELATED WORK

**Spatial reasoning in multimodal models.**    Spatial reasoning, the capacity to understand geometric structures and relational layouts in visual scenes, is central to embodied intelligence. Early multimodal benchmarks such as CLEVR (Johnson et al., 2016) and GQA (Hudson & Manning, 2019) tested compositional reasoning with synthetic or curated templates, which allowed controlled analysis but provided limited ecological validity. With the advent of large vision–language models (VLMs) (Li et al., 2023; Alayrac et al., 2022; OpenAI, 2023), spatial reasoning has been revisited in more naturalistic datasets, though systematic weaknesses remain. Despite strong performance on captioning and visual QA, these models often fail in tasks requiring depth ordering, occlusion inference, or consistent reference-frame alignment. This discrepancy highlights a broader problem: scaling alone does not guarantee progress in structured spatial understanding.

**Spatial reasoning benchmarks.**    A range of benchmarks have recently been proposed to probe these gaps (Table 1). SCANQA (Azuma et al., 2022) was among the first embodied QA datasets, requiring reasoning about 3D indoor environments, but its limited scale (800 scenes) and relatively shallow complexity restrict broader generalization. Datasets such as VISUAL SPATIAL (Liu et al., 2023) and WHAT'S UP (Kamath et al., 2023) expanded scope to real-world or household contexts, but relied on template-driven questions, leading to reduced linguistic variability and an overrepresentation of simple binary or multiple-choice relationships. More recent efforts, including EMBSPATIAL-BENCH (Du et al., 2024) and SPACE3D-BENCH (Szymanska et al., 2024), increased task variety yet remained constrained by scale (2.2K and 211 samples, respectively) and annotation style (template-heavy or manual but narrow). SPATIALRGPT-BENCH and BLINK-SPATIAL introduced mixed domains and up to 14 categories, improving diversity, but still emphasized either static snapshots or limited multi-view simulation.

**Embodiment, difficulty, and diversity.**    The trend toward embodiment is visible in datasets like ROBOSPATIAL (Song et al., 2024), which scales up to 1M scenes with robot-centric questions. However, its template-based annotation makes questions less natural and easier to game with heuristic shortcuts. Similarly, SPATIALVLM (Chen et al., 2024b) offers unprecedented scale (billions of QA pairs), but its two-category coverage renders it unsuitable for fine-grained diagnostic evaluation. On the other end, high-quality manual datasets such as VSI-BENCH (Yang et al., 2024) achieve improved annotation fidelity, but with only 288 scenes, they remain insufficient for training or robust cross-model comparison. In terms of complexity, most existing benchmarks cap at "medium" (Mm in Table 1), rarely extending to the "hard" or "extremely hard" levels that require multi-step causal inference. This leaves a wide evaluation gap for next-generation models.

**Evaluation protocols.**    Equally important are evaluation types and analysis depth. Most prior work employs close-ended QA or binary judgments (e.g., VISUAL SPATIAL, EMBSPATIAL-BENCH), which simplify scoring but restrict the expressivity of model outputs. A few datasets such as ROBOSPATIAL, SPATIAL-MM, and SPATIALAB-OPEN introduce open-ended evaluation, but only SPATIALAB-OPEN couples this with extremely high complexity, manual puzzle-style annotations, and comprehensive analysis of errors. This enables a deeper understanding of systematic weaknesses beyond accuracy, including calibration and cross-model divergence, which existing datasets rarely capture. Without this granularity, performance gains can be misleading, as models may overfit to dataset-specific biases rather than generalizing robust spatial competence.

**Positioning of SPATIALAB.**    SPATIALAB directly addresses these limitations. Compared to prior datasets that span between 2 and 14 task categories, SPATIALAB covers 30 distinct categories across geometry, occlusion, orientation, relative positioning, size and scale, and navigation, offering unprecedented task diversity. Unlike template-based benchmarks, all questions are manually authored and puzzle-inspired, ensuring linguistic richness and reducing shortcut opportunities. Furthermore, SPATIALAB is fully embodied and multi-modal, including both static and dynamic visual contexts, thus bridging gaps between robotics-style datasets (ROBOSPATIAL) and web-scale corpora (SPATIALVLM). As shown in Table 1, even the strongest baselines, such as InternVL3.5-72B and GPT-5-mini, perform well below 60%, with the open-ended variant yielding only 40.93% accuracy. This large performance gap demonstrates the difficulty of the benchmark and its utility for driving future research.

**Toward next-generation evaluation.** By combining high difficulty, broad coverage, manual annotation, and open-ended tasks, SPATIALAB extends the landscape of spatial reasoning benchmarks from template-driven, narrow evaluations to comprehensive, embodied testing. This design makes it possible not only to assess whether a model answers correctly, but also to identify systematic error patterns, probe causal and counterfactual reasoning, and measure calibration reliability. In this sense, SPATIALAB provides a rigorous foundation for the next phase of multimodal research, complementing datasets like MIND THE GAP and VLM4D that explore temporal or generative aspects but remain narrower in coverage. We expect future work to leverage SPATIALAB as a high-bar diagnostic tool, pushing models beyond surface-level recognition toward genuine grounded spatial intelligence.

## A.1 DETAILED ANALYSIS OF LIMITATIONS IN PRIOR BENCHMARKS

To situate SPATIALAB within the broader landscape, we conducted a detailed examination of existing spatial reasoning benchmarks. Earlier efforts have contributed important foundations, yet they present structural and methodological constraints that limit their value for evaluating current VLMs. These constraints fall into three areas: synthetic-domain bias, limited task scope, and restrictive evaluation formats. Table 4 summarizes the major limitations identified.

**Synthetic and Template-Based Constraints.** Benchmarks such as CLEVR and GQA rely on synthetic environments or template-generated linguistic structures. Although these settings offer controlled conditions for isolating compositional reasoning, the visual domain lacks real-world complexity and the linguistic space is narrow. Models can rely on template regularities rather than grounded spatial interpretation. Related simulated benchmarks such as ROBOSPATIAL face similar issues, resulting in a large sim-to-real gap.

**Domain and Scale Limitations.** Datasets grounded in 3D reconstruction, including ScanQA and Space3D-Bench, provide depth supervision but remain restricted to indoor scenes. This prevents assessment in outdoor or mixed environments that are essential for embodied agents. High-quality manual datasets such as VSI-Bench offer richer annotation but remain small in scale, limiting their usefulness for high-resolution subcategory analysis.

**Taxonomic and Evaluation Narrowness.** Large-scale datasets such as SpatialVLM cover broad tasks but lack detailed taxonomies, making them unsuitable for diagnosing specific weaknesses in orientation, occlusion, or layout reasoning. Many recent benchmarks, including EMBSPATIAL-BENCH and WHAT'S UP, rely heavily on Binary or MCQ formats. These formats can inflate performance by enabling option elimination rather than requiring generative spatial reasoning. SPATIALAB mitigates these issues by combining MCQ and Open-Ended formats across a detailed taxonomy of 30 spatial subcategories.

Table 4: Limitations identified in representative prior spatial benchmarks compared with SPATIALAB. Limitations are categorized by domain constraints, evaluation format, and task taxonomy.

| Benchmark | Domain | Eval. Format | Taxonomy | Primary Limitations Identified |
|---|---|---|---|---|
| **CLEVR** and **GQA** | Synthetic | Closed-Ended | Relational | Artificial visual domain; template-driven language; prone to shortcut learning. |
| **ScanQA** | Indoor 3D | Open-Ended | Navigation | Restricted to indoor point clouds; limited scale ($< 1K$ scenes). |
| **SpatialVLM** | Web Data | Open-Ended | General | Covers only a small number of high-level spatial categories. |
| **EmbSpatial-Bench** | Indoor | MCQ / Binary | 6 Categories | Relies on closed-ended formats; limited complexity in reasoning chains. |
| **RoboSpatial** | Simulator | Hybrid | Embodied | Template-generated questions in simulated scenes produce a sim-to-real gap. |
| **Space3D-Bench** | Indoor | MCQ | 6 Categories | Very small scale ($\sim$200 samples) prevents robust statistical evaluation. |
| **OmniSpatial** | Mix | MCQ / Binary | Metric | Emphasizes metric outputs; limited support for explanatory reasoning. |
| **SpatiaLab (Ours)** | In-the-Wild | MCQ + Open | 30 Sub-cats | Addresses identified limitations with real-world images, detailed taxonomy, and diverse evaluation formats. |

## B TASK DESIGN AND DESCRIPTIONS

Understanding spatial reasoning requires more than simple recognition of objects; it depends on how those objects relate, align, and interact in physical space. To capture this complexity, we introduce *SpatiaLab*, a benchmark designed to test fine-grained aspects of spatial understanding across diverse categories. The tasks in SpatiaLab reflect how humans naturally describe and reason about environments, ranging from relative positioning (e.g., *What's to the left of the microwave?*) to geometric inference (e.g., *Would this stack fall if another book is added?*). Each category is further broken into subcategories that highlight subtle but essential dimensions of reasoning, such

as occlusion, orientation, navigation, and physical effects like gravity. By framing each subtask in accessible, everyday terms, SpatiaLab encourages models to go beyond object detection and engage in structured spatial interpretation. This design allows us to probe how well systems generalize across contexts that matter for robotics, AR/VR, autonomous driving, and human-computer interaction. The following sections provide short, task-level descriptions for every category and subcategory in SpatiaLab.

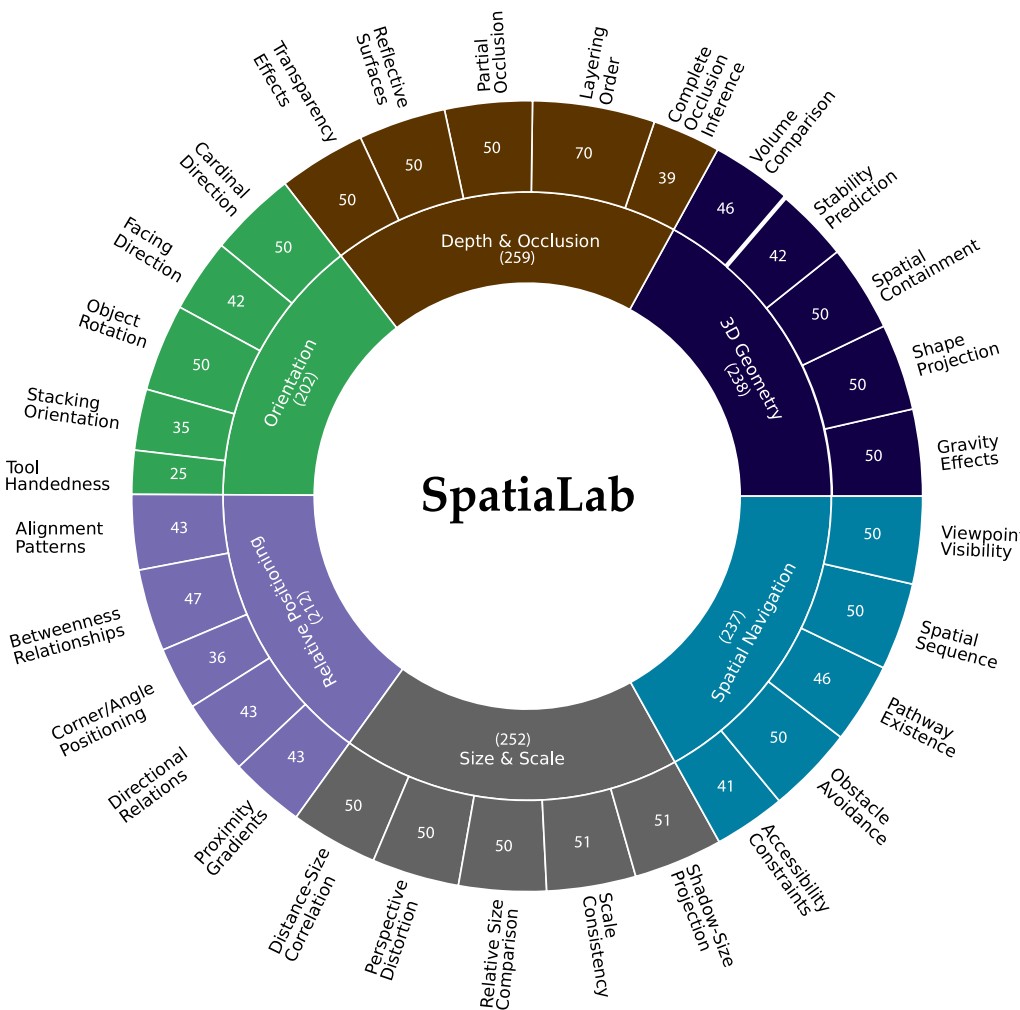

Figure 5: **Categories and subcategories of spatial reasoning in SPATIALAB**. Each category decomposes into five subcategories, yielding thirty task types in total.

The following outlines the rationale and real-world utility guiding each task design, per category:

## B.1 RELATIVE POSITIONING

**Directional Relations.** Directional relations describe how objects are positioned relative to one another, such as left, right, above, or below. Understanding these relations enables models to interpret spatial layouts more naturally, just like how humans describe scenes. For example, identifying what lies immediately to the left of an appliance helps in scene understanding and robot navigation.

**Proximity Gradients.** Proximity reasoning involves recognizing which objects are closest or farthest from a reference point. This is key for questions like "Which object is nearest to the doorway?" Models capable of estimating gradients of closeness can support navigation tasks and assistive technologies.

**Alignment Patterns.** Alignment concerns whether objects follow a certain orientation or order, such as being stacked vertically or arranged in a straight line. Recognizing alignment patterns helps in understanding organization in environments like shelves, tables, or warehouses. Such reasoning is also critical for tasks like robotic sorting and automated inspection.

**Betweenness Relationships.** Betweenness focuses on identifying objects that occupy the middle ground between two references. For instance, asking what sits between the lamp and the armchair requires reasoning about spatial continuity. This helps in navigation, retrieval, and situational awareness tasks.

**Corner/Angle Positioning.** Corners and angled positions often serve as reference anchors in spatial reasoning. Questions such as "What's positioned in the back-right corner?" require the model to localize objects within constrained regions. This kind of reasoning is vital in robotics for mapping and placement.

## B.2 Depth & Occlusion

**Layering Order.** Layering requires distinguishing which objects appear in front versus behind. For example, identifying the frontmost item on a table reflects depth reasoning. This is particularly useful in AR/VR applications where visual stacking affects user experience.

**Partial Occlusion.** Partial occlusion occurs when one object hides part of another from view. Recognizing the hidden item behind a coffee cup, for instance, requires filling in missing information. This strengthens robustness in perception systems operating in cluttered environments.

**Complete Occlusion Inference.** Sometimes objects are fully hidden, yet clues like shadows or bulges reveal their presence. Inferring what lies under a blanket requires the model to use indirect evidence. Such reasoning mirrors human inference in uncertain visual conditions.

**Transparency Effects.** Transparent surfaces, such as glass or water, allow partial visibility of objects behind them. Understanding what's visible through a fish tank requires blending occlusion reasoning with transparency perception. This plays an important role in simulation, graphics, and embodied AI.

**Reflective Surfaces.** Reflective reasoning asks models to interpret mirrors, shiny metals, or glass reflections. Identifying what is reflected in a mirror involves indirect spatial mapping. This is essential for safety in autonomous vehicles and situational awareness in robotics.

## B.3 Orientation

**Cardinal Direction.** Cardinal reasoning relates to compass directions like north, south, east, and west. Asking which way a front door faces requires integrating global orientation cues. This is critical for navigation and geospatial alignment.

**Object Rotation.** Rotation reasoning measures how much an object has been turned around an axis. Estimating that a chair is rotated by some degrees is crucial for accurate 3D modeling. This helps robots align objects correctly for manipulation.

**Facing Direction.** Facing direction specifies the orientation of an object's front side. For example, determining whether a monitor faces north requires mapping its surface orientation to global references. This is important in surveillance and robotics.

**Stacking Orientation.** Stacking involves understanding whether objects are piled vertically, horizontally, or in mixed orientations. Recognizing such layouts ensures stability in warehouses and homes. It is also critical for safe robotic stacking.

**Tool Handedness.** Certain tools, like scissors or gloves, are designed for left- or right-handed use. Identifying handedness requires fine-grained shape and orientation perception. This enhances assistive robotics in human-centered environments.

### B.4 SIZE & SCALE

**Relative Size Comparison.** Relative comparisons assess whether one object is larger, taller, or smaller than another. Asking if a vase is taller than a statue requires proportional reasoning. This is crucial for packing, fitting, and recognition tasks.

**Perspective Distortion.** Perspective can make distant objects appear smaller than nearby ones, even if the actual size differs. Identifying illusions like a faraway car looking smaller than a nearby bicycle requires compensating for depth cues. This ability makes models more robust to camera perspectives.

**Distance-Size Correlation.** Here, size judgments depend on distance estimation. For instance, a nearby bicycle appearing larger than a far car should not confuse true size reasoning. This helps in navigation, AR/VR, and visual grounding.

**Scale Consistency.** Scale reasoning involves checking whether object sizes are reasonable in context. A mouse larger than a chair signals inconsistency, which models must detect. This strengthens realism in simulated environments.

**Shadow-Size Projection.** Shadows provide indirect cues about object size and light source position. Estimating how tall a light must be based on shadow length requires geometric reasoning. This is useful in graphics, architecture, and physics-based modeling.

### B.5 SPATIAL NAVIGATION

**Pathway Existence.** Pathway reasoning checks if a clear route exists between two points. For example, determining if one can reach the desk without moving chairs tests navigability. This ability underpins indoor robot planning.

**Obstacle Avoidance.** Avoidance requires finding routes that minimize collisions. Asking for the clearest path from door to window directly maps to robotic and human navigation. It is essential in autonomous driving and indoor mobility aids.

**Viewpoint Visibility.** From a given viewpoint, certain objects may be blocked or visible. For example, checking what is obscured when sitting on the couch requires simulating perspective. This strengthens AR/VR realism and robotic vision.

**Spatial Sequence.** Sequence reasoning identifies the order of encounters along a path. Walking from the kitchen to the balcony, one might pass the dining table first. Such reasoning mimics human navigation memory.

**Accessibility Constraints.** Constraints identify which objects are blocked from access due to obstacles. For instance, a cabinet may be unreachable because of a table in front. This is crucial for assistive AI and ergonomic design.

### B.6 3D GEOMETRY

**Volume Comparison.** Comparing volume asks which object occupies more three-dimensional space. Judging whether a box is larger than a sphere helps in packing and containment tasks. This is important for logistics and simulation.

**Stability Prediction.** Stability reasoning predicts whether adding or moving objects will topple a stack. For instance, checking if placing a book will collapse a pile mirrors human intuition. This supports safety in robotics and construction.

**Shape Projection.** Projection tasks ask what shape an object would cast under certain rotations or lighting. For example, a sphere always projects a circle, while a cube may project a square or hexagon. Such reasoning strengthens geometric understanding.

**Spatial Containment.** Containment checks whether one object can fit inside another. For example, reasoning if a ball can fit in a box supports packing and design. This task aligns with intuitive geometry.

**Gravity Effects.** Gravity reasoning predicts motion outcomes under physical forces. For example, a rolling ball may stop under a table due to friction and barriers. This strengthens real-world simulation capabilities.

## C  MORE DETAILS ON BENCHMARK CONSTRUCTION

### C.1  ANNOTATOR RECRUITMENT

We recruited annotators aged between 24 and 30 years, all of whom were engineering graduates. They participated voluntarily and worked independently without external incentives. Before starting the annotation process, we provided structured training sessions to ensure clarity of task requirements and consistency across annotators. We also maintained all ethical and quality control standards throughout the process. This approach ensured that the resulting annotations were both reliable and aligned with the dataset design.

### C.2  ANNOTATOR TRAINING AND PROTOCOL

Before beginning annotation, we designed a structured training program to ensure consistency and reliability across the full taxonomy of tasks. We first introduced annotators to the six core categories (*Relative Positioning, Depth & Occlusion, Orientation, Size & Scale, Spatial Navigation, and 3D Geometry*) and their thirty subcategories. For each subcategory, we provided curated examples that illustrated the corresponding spatial relations (e.g., directional relations, occlusion inference, stability prediction) along with counter-examples to highlight common pitfalls. Beyond the taxonomy, we trained annotators to account for six meta-dimensions of image variability: lighting, texture complexity, edge density, dominant relation, material type, and gravity constraints, so that questions explicitly leveraged visual diversity. We demonstrated how these factors interact in real-world settings, such as reasoning about occlusion under low-contrast lighting or maintaining scale consistency under perspective distortion.

We also introduced the two evaluation modes: multiple-choice and open-ended. In multiple-choice settings, annotators learned to design precise distractors to test fine-grained discrimination, while in open-ended settings they practiced formulating prompts that encourage compositional reasoning. To ground this training, we ran calibration sessions in which annotators generated both types of questions for sample images and iteratively refined their designs with feedback. To situate task design within the broader research landscape, we reviewed prior benchmarks (e.g., SpatialBot-Bench, EmbSpatial, SpatialMM) with annotators and analyzed their limitations. These resources often emphasize simplified relations, synthetic or staged scenes, and narrow reasoning types, while neglecting natural complexity such as shadows, reflections, or gravity-sensitive stability. This gap analysis underscored the need for richer, more ambiguous, and dynamically layered spatial contexts.

Finally, we provided detailed demonstrations of the annotation protocol. These demonstrations specified how to select images with sufficient spatial complexity, how to map questions unambiguously to subcategories, and how to avoid trivial formulations. We instructed annotators to prioritize tasks requiring multi-step reasoning, such as depth layering, relational chains, or cross-view scale inference, over binary labels. Through this systematic training and iterative feedback, we ensured that the final dataset balanced taxonomic coverage with real-world relevance.

## D  MORE DETAILS ON EXPERIMENTS AND EVALUATION

### D.1  MULTIPLE CHOICE EVALUATION

The prompts used for the answer model are shown in Figure 6. We set `temperature` to 0 and `max_tokens` to 10, while keeping all other generation parameters at their default values.

---

**SpatiaLab-MCQ : Direct : Prompt for Answering Model**

```
You are a spatial reasoning assistant. Given a multiple choice question about an image,
Answer with a single number (1-4) corresponding to the option.
Question:
{question}
Options:
1. {opt1}
2. {opt2}
3. {opt3}
4. {opt4}
```

**SpatiaLab-Open : Direct : Prompt for Answering Model**

```
You are a spatial reasoning assistant.
Answer the following question in 2-4 sentences about the given image.
Question:
{question}
```

**SpatiaLab-Open : Direct : Prompt for Judge Model**

```
You are a strict evaluator.
You will receive a question, the ground truth answer, and a model's answer.
Return only 1 if the model's answer is correct, otherwise 0.

Question:
{question}

Ground Truth Answer:
{ground_truth}

Model Answer:
{model_answer}

Output only a single digit: 0 or 1.
```

Figure 6: Prompts used for evaluation.

## D.2 OPEN-ENDED EVALUATION

The prompts used for both the answer model and the judge model are provided in Figure 6. We set `temperature` to 0.7 and `max_tokens` to 1200, with all other parameters kept at their default values.

### D.2.1 EVALUATION OF LLM-AS-A-JUDGE FOR OPEN-ENDED EVALUATION

To evaluate *Gemini-2.5-Flash* as an automatic judge, we selected a balanced set of 240 samples covering all categories. Each sample included the evaluation setup (question, ground truth, and model output), which was scored independently by three human annotators. Annotators assigned binary scores: 1 for correct and 0 for incorrect. We compared the LLM's judgments against the human annotations using multiple complementary agreement metrics, computed both overall and per category. Specifically, we employed **Cohen's Kappa**, **Accuracy with Majority Vote**, and **Fleiss' Kappa**, defined as follows.

**Cohen's Kappa.** Cohen's kappa ($\kappa$) measures pairwise agreement between two raters while correcting for chance agreement. For binary task, it is defined as $\kappa = \frac{p_o - p_e}{1 - p_e}$, where $p_o$ is the observed agreement and $p_e$ is the expected agreement by chance: $p_e = \sum_{c \in \{0,1\}} p_c^{(1)} p_c^{(2)}$. Here, $p_c^{(1)}$ and $p_c^{(2)}$ are the marginal probabilities that each rater assigns class $c$. An $\kappa = 1$: perfect agreement; $\kappa = 0$: agreement equals chance; $\kappa < 0$: systematic disagreement) We computed Cohen's kappa between the LLM and each annotator individually ($\kappa_{\text{Human1}}, \kappa_{\text{Human2}}, \kappa_{\text{Human3}}$), as well as between the LLM and the majority vote of the annotators ($\kappa_{\text{majority}}$).

**Accuracy with Majority Vote.** We also measured raw agreement between the LLM and the annotator majority. Accuracy is defined as $\text{Accuracy} = \frac{1}{N} \sum_{i=1}^{N} \mathbf{1}\left(y_i^{\text{LLM}} = y_i^{\text{maj}}\right)$, where $y_i^{\text{LLM}}$ is the LLM's decision for sample $i$, and $y_i^{\text{maj}}$ is the majority vote among annotators. Unlike Cohen's kappa, accuracy does not correct for chance agreement; it simply reflects the proportion of identical labels.

**Fleiss' Kappa.** To assess inter-annotator reliability, we computed Fleiss' kappa ($\kappa_F$), which generalizes Cohen's kappa to multiple raters. Let $n_{ij}$ denote the number of annotators assigning category $j$ to item $i$, and define $P_i = \frac{1}{n(n-1)} \sum_j n_{ij}(n_{ij} - 1)$, where $n$ is the number of annotators. The observed agreement is $\bar{P} = \frac{1}{N} \sum_{i=1}^{N} P_i$, and the expected agreement is $\bar{P}_e = \sum_j p_j^2$, with $p_j = \frac{1}{Nn} \sum_i n_{ij}$. Fleiss' kappa is then computed as $\kappa_F = \frac{\bar{P}-\bar{P}_e}{1-\bar{P}_e}$. Higher values indicate stronger inter-annotator reliability beyond chance.

Table 5: Agreement metrics between Gemini-2.5-Flash as an automatic judge and three human annotators across all categories. The table reports Cohen's kappa with the majority vote and each annotator, raw accuracy against the majority, and Fleiss' kappa among all annotators. Bold values highlight the highest agreement within each category.

| Category | $\kappa$Majority | $\kappa$Human1 | $\kappa$Human2 | $\kappa$Human3 | Acc. (majority) | Fleiss $\kappa$ (all) |
|---|---|---|---|---|---|---|
| Overall | 0.738 | 0.738 | 0.681 | 0.795 | 0.880 | 0.774 |
| 3D Geometry | **1.000** | **1.000** | 0.875 | **1.000** | **1.000** | **0.916** |
| Depth & Occlusion | 0.805 | 0.805 | **0.899** | 0.697 | 0.909 | 0.871 |
| Orientation | 0.500 | 0.500 | 0.426 | 0.588 | 0.810 | 0.571 |
| Relative Positioning | 0.833 | 0.657 | 0.676 | 0.833 | 0.917 | 0.778 |
| Size & Scale | 0.417 | 0.571 | 0.417 | 0.696 | 0.714 | 0.704 |
| Spatial Navigation | 0.815 | 0.815 | 0.667 | **1.000** | 0.933 | 0.773 |

Interpretation Table 5 shows that the LLM judge (Gemini-2.5-Flash) demonstrates strong overall agreement with human annotators, achieving a Cohen's kappa of 0.738 with the majority vote and 0.681–0.795 against individual annotators. Raw accuracy against the majority is 0.880 overall, while Fleiss' kappa among human annotators is 0.774, indicating substantial consistency. Certain categories, such as 3D Geometry, reach perfect agreement with $\kappa = 1.000$ and accuracy of 1.000. Depth & Occlusion also shows high agreement ($\kappa$_majority $= 0.805$, accuracy = 0.909). Categories like Orientation ($\kappa = 0.500$, accuracy = 0.810) and Size & Scale ($\kappa = 0.417$, accuracy = 0.714) show moderate agreement but remain within acceptable reliability ranges.

These results indicate that Gemini-2.5-Flash performs comparably to human evaluators in scoring correctness, with Cohen's kappa values mostly in the substantial range (0.61–0.80) and some categories reaching almost perfect agreement (higher than 0.81). Fleiss' kappa of 0.774 confirms that human annotators themselves are highly consistent, validating the reliability of the reference scores. The LLM achieves perfect or near-perfect agreement in clearly defined categories such as 3D Geometry ($\kappa = 1.000$, accuracy = 1.000) and Spatial Navigation ($\kappa = 0.815$, accuracy = 0.933), demonstrating accurate pattern recognition. Moderate agreement in categories like Orientation and Size & Scale reflects task complexity and inherent subjectivity rather than model shortcomings. Overall, it shows that LLM-as-judge shows robust reliability and can serve as a practical automated scoring tool, closely mirroring human evaluation.

# E    Error Analysis of SpatiaLab-MCQ

## E.1    Sub-Category-wise Quantitative Error Analysis

**Closed-source Models.** Closed-source models (Table 6) produce some of the strongest results across the benchmark. The best case is **GPT-4o-mini**, which reaches 85.71% in *Stacking Orientation*, the highest score overall. **Gemini-2.0-Flash** performs well in occlusion and projection, achieving 64.0% in *Partial Occlusion* and 64.71% in *Shadow-Size Projection*. Similarly, **GPT-5-mini** scores 68.0% in *Tool Handedness* and 55.71% in *Layering Order*. Weaknesses remain, as **Claude 3.5 Haiku** only reaches 32.0% in *Relative Size Comparison*. On average, closed-source systems outperform open-source ones by 15–25 percentage points across several categories. However, within-model variation remains large, often exceeding 50 percentage points. We therefore conclude that while closed-source training pipelines deliver higher ceilings, they do not solve the challenge of uniform spatial competence.

Table 6: Multiple-choice Evaluation Accuracy on SPATIALAB-MCQ by Question Sub-Categories for Closed-source Models. We bold and underline the best score within each model category.

| Category | Sub-Category | GPT-4o-mini | GPT-5-mini | Gemini-2.0-Flash | Gemini-2.5-Flash | Claude 3.5 Haiku | Mistral Medium 3.1 |
|---|---|---|---|---|---|---|---|
| 3D Geometry | Gravity Effects | **52.0** | 50.0 | 46.0 | **52.0** | **52.0** | 46.0 |
| | Shape Projection | 54.0 | 48.0 | 50.0 | 50.0 | 46.0 | **60.0** |
| | Spatial Containment | 36.0 | **46.0** | 44.0 | 38.0 | 34.0 | 38.0 |
| | Stability Prediction | 52.38 | 52.38 | **59.52** | 50.0 | 45.24 | 52.38 |
| | Volume Comparison | 41.3 | **47.83** | 39.13 | 34.78 | 34.78 | 36.96 |
| Depth and Occlusion | Complete Occlusion Inference | 61.54 | **69.23** | 61.54 | 61.54 | 61.54 | 46.15 |
| | Layering Order | 32.86 | **55.71** | 50.0 | 42.86 | 32.86 | 45.71 |
| | Partial Occlusion | 42.0 | 54.0 | **64.0** | 50.0 | 44.0 | 52.0 |
| | Reflective Surfaces | 34.0 | **54.0** | 46.0 | 48.0 | 44.0 | 44.0 |
| | Transparency Effects | 32.0 | 44.0 | 60.0 | 44.0 | 36.0 | **62.0** |
| Orientation | Cardinal Direction | 26.0 | **48.0** | 44.0 | 30.0 | 44.0 | 38.0 |
| | Facing Direction | 45.24 | **57.14** | **57.14** | 47.62 | 35.71 | 45.24 |
| | Object Rotation | 40.0 | **60.0** | 50.0 | 46.0 | 38.0 | 44.0 |
| | Stacking Orientation | **85.71** | 77.14 | 62.86 | 65.71 | 68.57 | 62.86 |
| | Tool Handedness | 52.0 | **68.0** | 64.0 | 64.0 | 56.0 | 56.0 |
| Relative Positioning | Alignment Patterns | 44.19 | **58.14** | 55.81 | 53.49 | 46.51 | **58.14** |
| | Betweenness Relationships | 51.06 | 63.83 | **70.21** | 63.83 | 44.68 | 61.7 |
| | Corner/Angle Positioning | 47.22 | 63.89 | 50.0 | 58.33 | 50.0 | **77.78** |
| | Directional Relations | 48.84 | **79.07** | 67.44 | 69.77 | 53.49 | 69.77 |
| | Proximity Gradients | 44.19 | **48.84** | 44.19 | 34.88 | 37.21 | 44.19 |
| Size and Scale | Distance-Size Correlation | **56.0** | **56.0** | **56.0** | 42.0 | 34.0 | 42.0 |
| | Perspective Distortion | **50.0** | 40.0 | 46.0 | 36.0 | 38.0 | 34.0 |
| | Relative Size Comparison | 42.0 | 44.0 | **58.0** | 46.0 | 32.0 | 38.0 |
| | Scale Consistency | **47.06** | 45.1 | **47.06** | 39.22 | 33.33 | 41.18 |
| | Shadow-Size Projection | 52.94 | 39.22 | **64.71** | 49.02 | 41.18 | 52.94 |
| Spatial Navigation | Accessibility Constraints | 56.1 | 60.98 | **65.85** | **65.85** | 48.78 | 56.1 |
| | Obstacle Avoidance | 38.0 | **48.0** | 34.0 | 34.0 | 46.0 | 40.0 |
| | Pathway Existence | **54.35** | 41.3 | 43.48 | 45.65 | 39.13 | 36.96 |
| | Spatial Sequence | 66.0 | **78.0** | 48.0 | 62.0 | 56.0 | 44.0 |
| | Viewpoint Visibility | 36.0 | **54.0** | 48.0 | 50.0 | 40.0 | 34.0 |

Table 7: Multiple-choice Evaluation Accuracy on SPATIALAB-MCQ by Question Sub-Categories for Open-source Large Models. We bold and underline the best score within each model category.

| Category | Sub-Category | llama-3.2-11b-vision-instruct | Gemma-3-27B-it | qwen-2.5-vl-32b-instruct | InternVL3-5-72B | qwen-2.5-vl-72b-instruct | llama-3.2-90b-vision-instruct |
|---|---|---|---|---|---|---|---|
| 3D Geometry | Gravity Effects | 22.0 | 48.0 | 54.0 | 52.0 | 52.0 | **56.0** |
| | Shape Projection | 32.0 | 42.0 | 44.0 | **50.0** | 46.0 | 36.0 |
| | Spatial Containment | 32.0 | 34.0 | 36.0 | 44.0 | **46.0** | 36.0 |
| | Stability Prediction | 33.33 | **54.76** | 38.1 | 52.38 | 47.62 | 52.38 |
| | Volume Comparison | 13.04 | 39.13 | 32.61 | **52.17** | 43.48 | 36.96 |
| Depth and Occlusion | Complete Occlusion Inference | 30.77 | 61.54 | 58.97 | **66.67** | 48.72 | **66.67** |
| | Layering Order | 34.29 | 34.29 | 37.14 | **50.0** | **50.0** | 38.57 |
| | Partial Occlusion | 32.0 | 34.0 | 44.0 | **60.0** | 48.0 | 58.0 |
| | Reflective Surfaces | 28.0 | 38.0 | 26.0 | **54.0** | 50.0 | 44.0 |
| | Transparency Effects | 26.0 | 40.0 | 40.0 | 60.0 | 46.0 | **62.0** |
| Orientation | Cardinal Direction | 22.0 | **48.0** | 34.0 | 46.0 | **48.0** | 40.0 |
| | Facing Direction | 21.43 | 42.86 | 52.38 | **54.76** | 50.0 | 50.0 |
| | Object Rotation | 2.0 | 40.0 | 38.0 | **48.0** | **48.0** | 44.0 |
| | Stacking Orientation | 28.57 | 65.71 | 65.71 | 68.57 | 62.86 | **71.43** |
| | Tool Handedness | 40.0 | 48.0 | 52.0 | **56.0** | **56.0** | **56.0** |
| Relative Positioning | Alignment Patterns | 27.91 | **55.81** | 48.84 | 53.49 | **55.81** | 53.49 |
| | Betweenness Relationships | 42.55 | 55.32 | 55.32 | **65.96** | 61.7 | 61.7 |
| | Corner/Angle Positioning | 58.33 | 55.56 | 36.11 | **80.56** | 50.0 | 75.0 |
| | Directional Relations | 60.47 | 62.79 | 51.16 | **83.72** | 62.79 | 65.12 |
| | Proximity Gradients | 27.91 | 41.86 | 32.56 | **48.84** | 39.53 | 41.86 |
| Size and Scale | Distance-Size Correlation | 36.0 | 50.0 | **54.0** | 52.0 | 44.0 | 46.0 |
| | Perspective Distortion | 22.0 | 42.0 | 36.0 | 46.0 | 34.0 | **52.0** |
| | Relative Size Comparison | 28.0 | 48.0 | 40.0 | 48.0 | **52.0** | 46.0 |
| | Scale Consistency | 29.41 | **47.06** | 41.18 | 43.14 | 37.25 | 37.25 |
| | Shadow-Size Projection | 37.25 | 52.94 | 54.9 | **56.86** | 50.98 | 52.94 |
| Spatial Navigation | Accessibility Constraints | 29.27 | 58.54 | 43.9 | **75.61** | 53.66 | 43.9 |
| | Obstacle Avoidance | 24.0 | 38.0 | 38.0 | **46.0** | 42.0 | 40.0 |
| | Pathway Existence | 28.26 | **54.35** | 36.96 | 39.13 | **54.35** | 47.83 |
| | Spatial Sequence | 34.0 | 52.0 | 56.0 | **66.0** | 54.0 | 62.0 |
| | Viewpoint Visibility | 44.0 | 36.0 | 34.0 | **50.0** | 42.0 | 48.0 |

**Open-source Large Models.** When scaling to large open-source models (Table 7), we see clear improvements but also severe fragilities. The **InternVL3.5-72B** achieves 80.56% in *Corner/Angle Positioning*, one of the highest open-source results. Both **InternVL3.5-72B** and **llama-3.2-90b-vision-instruct** reach 66.67% in *Complete Occlusion Inference*. Yet weaknesses persist, such as the **llama-3.2-11b-vision-instruct** scoring only 2.0% in *Object Rotation*. Gains are also evident in *Stacking Orientation*, where **llama-3.2-90b-vision-instruct** scores 71.43%, but this comes with losses in other categories. The within-model spread often exceeds 70 percentage points, highlighting brittle specialization. Gravity reasoning remains modest at 56.0%. We find that scaling yields higher peaks but does not resolve structural weaknesses. Our conclusion is that size amplifies selective strengths while leaving foundational representation gaps intact.

**Open-source Small Models.** We observe that open-source small models (Table 8) display sharp contrasts in performance across sub-categories. For example, **qwen-2.5-vl-7b-instruct** reaches 74.29% in *Stacking Orientation*, while **InternVL3.5-1B** collapses to 12.0% in *Cardinal Direction*. Scores in *Pathway Existence* remain modest, with the best models tied at 34.78%. Gravity-related

Table 8: Multiple-choice Evaluation Accuracy on SPATIALAB-MCQ by Question Sub-Categories for Open-source Small Models. We bold and underline the best score within each model category.

| Category | Sub-Category | InternVL3.5-1B | InternVL3.5-2B | qwen-2.5-vl-3b-instruct | InternVL3.5-4B | Gemma-3-4B-it | qwen-2.5-vl-7b-instruct |
|---|---|---|---|---|---|---|---|
| 3D Geometry | Gravity Effects | 46.0 | 38.0 | 48.0 | **50.0** | **50.0** | **50.0** |
| | Shape Projection | 30.0 | 42.0 | 38.0 | 42.0 | 40.0 | **44.0** |
| | Spatial Containment | 30.0 | 28.0 | 34.0 | 34.0 | **36.0** | 34.0 |
| | Stability Prediction | 30.95 | 35.71 | 50.0 | **52.38** | 45.24 | **52.38** |
| | Volume Comparison | 30.43 | 26.09 | 36.96 | 36.96 | **47.83** | 34.78 |
| Depth and Occlusion | Complete Occlusion Inference | 30.77 | 30.77 | 41.03 | **64.1** | 33.33 | 41.03 |
| | Layering Order | 30.0 | 24.29 | 30.0 | **32.86** | 22.86 | 27.14 |
| | Partial Occlusion | 38.0 | 30.0 | 38.0 | **54.0** | 46.0 | 42.0 |
| | Reflective Surfaces | 38.0 | 34.0 | 36.0 | 32.0 | **42.0** | 40.0 |
| | Transparency Effects | 26.0 | 42.0 | 36.0 | 40.0 | 32.0 | **44.0** |
| Orientation | Cardinal Direction | 12.0 | 18.0 | 36.0 | 24.0 | **38.0** | 30.0 |
| | Facing Direction | 21.43 | 28.57 | **52.38** | 40.48 | 42.86 | 38.1 |
| | Object Rotation | 12.0 | 28.0 | **42.0** | 34.0 | **42.0** | 34.0 |
| | Stacking Orientation | 45.71 | 57.14 | 62.86 | 68.57 | 68.57 | **74.29** |
| | Tool Handedness | 40.0 | 36.0 | 40.0 | **60.0** | 48.0 | 48.0 |
| Relative Positioning | Alignment Patterns | 27.91 | 30.23 | 39.53 | **44.19** | 41.86 | 41.86 |
| | Betweenness Relationships | 38.3 | 38.3 | 46.81 | **65.96** | 46.81 | 53.19 |
| | Corner/Angle Positioning | 50.0 | 55.56 | 47.22 | **69.44** | 55.56 | 50.0 |
| | Directional Relations | 51.16 | 58.14 | 27.91 | **60.47** | 46.51 | 48.84 |
| | Proximity Gradients | 20.93 | 23.26 | **39.53** | 34.88 | **39.53** | 37.21 |
| Size and Scale | Distance-Size Correlation | 32.0 | 20.0 | **40.0** | 36.0 | 38.0 | 36.0 |
| | Perspective Distortion | 26.0 | 40.0 | 42.0 | 38.0 | **44.0** | **44.0** |
| | Relative Size Comparison | 34.0 | 32.0 | **46.0** | 38.0 | 30.0 | 42.0 |
| | Scale Consistency | 27.45 | 35.29 | **41.18** | 31.37 | 27.45 | **41.18** |
| | Shadow-Size Projection | 39.22 | 35.29 | **66.67** | 39.22 | 47.06 | 47.06 |
| Spatial Navigation | Accessibility Constraints | 36.59 | 29.27 | **51.22** | 41.46 | 39.02 | 43.9 |
| | Obstacle Avoidance | 24.0 | 26.0 | **38.0** | **38.0** | **38.0** | 36.0 |
| | Pathway Existence | **34.78** | 30.43 | 32.61 | **34.78** | **34.78** | 28.26 |
| | Spatial Sequence | 24.0 | 42.0 | 40.0 | **56.0** | 42.0 | 44.0 |
| | Viewpoint Visibility | 36.0 | 34.0 | 36.0 | **40.0** | 36.0 | 26.0 |

reasoning is similarly weak, with only 50.0% achieved by **InternVL3.5-4B** and **Gemma-3-4B-it**. The within-model spread often exceeds 60 percentage points, showing brittle generalization. These patterns suggest that smaller models are highly cue-dependent, performing well where visual templates are strong but failing in tasks requiring deeper spatial abstractions. We interpret this inconsistency as evidence that representation learning is uneven across sub-categories. Overall, we find that small models provide localized competence but lack robustness across the full spectrum of spatial reasoning.

**Reasoning Models.** Reasoning-enhanced models (Table 9) show notable benefits in select tasks but remain inconsistent overall. **o4-mini** reaches 65.12% in *Alignment Patterns* and 56.0% in *Gravity Effects*, outperforming most open baselines. **Gemini-2.0-Flash-Thinking** achieves 62.75% in *Shadow-Size Projection* and 52.17% in *Volume Comparison*, demonstrating stronger geometric reasoning. The strongest results come from **Gemini-2.5-Flash-Thinking**, which scores 71.79% in *Complete Occlusion Inference* and 70.21% in *Betweenness Relationships*. Yet the same family fails in others, such as 34.0% in *Cardinal Direction* and 34.0% in *Shape Projection*. This variance shows that reasoning mechanisms improve depth in certain sub-categories but do not broaden coverage. Within-model spreads frequently exceed 35 percentage points. We conclude that reasoning augmentation adds localized capacity but leaves structural gaps in spatial understanding. Overall, these models demonstrate promise but not generalizable competence.

**Spatial Reasoning Models.** Spatially specialized models (Table 10) reveal targeted improvements but lack generality. **SpaceQwen2.5-VL-3B-Instruct** achieves 48.0% in *Distance-Size Correlation* and 49.02% in *Scale Consistency*, clearly surpassing many open-source baselines in size–scale reasoning. The same model leads in navigation tasks, with 53.66% in *Accessibility Constraints* and 44.0% in *Obstacle Avoidance*. **SpaceOm** and **SpaceThinker-Qwen2.5VL-3B** reach 50.0% in *Stability Prediction*, matching expectations for domain-specific reasoning. However, performance drops sharply in other tasks, such as only 24.0% for *Spatial Containment*. Rarely do these models exceed 55.0%, even in their strongest areas. The spread across sub-categories frequently surpasses 30 percentage points. We find that specialization drives narrow peaks but does not produce holistic competence. Our interpretation is that domain targeting enhances select abilities but fails to integrate them into a robust, general framework.

**Overall Discussion.** Across all categories (Table 8, Table 7, Table 6, Table 10, Table 9), we observe consistent fragmentation. Small open-source models peak at 74.29% in *Stacking Orientation* but collapse to 12.0% in *Cardinal Direction*. Larger models raise ceilings, such as 80.56% in *Corner/Angle Positioning*, but still fail catastrophically with 2.0% in *Object Rotation*. Closed-source

Table 9: Multiple-choice Evaluation Accuracy on SPATIALAB-MCQ by Question Sub-Categories for Reasoning Models. We bold and underline the best score within each model category.

| Category | Sub-Category | o4-mini | Gemini-2.0-Flash-Thinking | Gemini-2.5-Flash-Thinking | Kimi-VL-A3B-Thinking-2506 |
|---|---|---|---|---|---|
| 3D Geometry | Gravity Effects | **56.0** | 38.0 | 46.0 | 50.0 |
| | Shape Projection | **50.0** | 36.0 | 40.0 | 34.0 |
| | Spatial Containment | **56.0** | 28.0 | 48.0 | 42.0 |
| | Stability Prediction | **50.0** | 35.71 | **50.0** | **50.0** |
| | Volume Comparison | 43.48 | **52.17** | 45.65 | 39.13 |
| Depth and Occlusion | Complete Occlusion Inference | 69.23 | 51.28 | **71.79** | 51.28 |
| | Layering Order | **57.14** | 30.0 | 41.43 | 35.71 |
| | Partial Occlusion | 50.0 | 42.0 | **58.0** | 40.0 |
| | Reflective Surfaces | **54.0** | 40.0 | 52.0 | 42.0 |
| | Transparency Effects | **64.0** | 50.0 | 54.0 | 42.0 |
| Orientation | Cardinal Direction | 46.0 | 34.0 | **50.0** | 28.0 |
| | Facing Direction | 50.0 | 42.86 | **54.76** | 45.24 |
| | Object Rotation | **56.0** | 36.0 | 48.0 | 30.0 |
| | Stacking Orientation | **71.43** | 57.14 | 57.14 | 65.71 |
| | Tool Handedness | 56.0 | 44.0 | **60.0** | 44.0 |
| Relative Positioning | Alignment Patterns | **65.12** | 41.86 | 44.19 | 34.88 |
| | Betweenness Relationships | 65.96 | 55.32 | **70.21** | 63.83 |
| | Corner/Angle Positioning | **75.0** | 55.56 | 61.11 | 47.22 |
| | Directional Relations | **65.12** | 44.19 | **65.12** | 60.47 |
| | Proximity Gradients | **51.16** | 32.56 | 41.86 | 48.84 |
| Size and Scale | Distance-Size Correlation | 44.0 | 42.0 | **56.0** | 44.0 |
| | Perspective Distortion | 40.0 | 44.0 | **50.0** | 36.0 |
| | Relative Size Comparison | 46.0 | 46.0 | **58.0** | 32.0 |
| | Scale Consistency | 39.22 | **56.86** | 52.94 | 43.14 |
| | Shadow-Size Projection | 35.29 | **62.75** | 58.82 | 43.14 |
| Spatial Navigation | Accessibility Constraints | **56.1** | 41.46 | 53.66 | 31.71 |
| | Obstacle Avoidance | 32.0 | **44.0** | **44.0** | 36.0 |
| | Pathway Existence | 45.65 | 41.3 | **47.83** | 34.78 |
| | Spatial Sequence | 64.0 | 48.0 | **68.0** | 54.0 |
| | Viewpoint Visibility | **60.0** | 40.0 | 54.0 | 48.0 |

Table 10: Multiple-choice Evaluation Accuracy on SPATIALAB-MCQ by Question Sub-Categories for Spatial Reasoning Models. We bold and underline the best score within each model category.

| Category | Sub-Category | SpaceOm | SpaceThinker-Qwen2.5VL-3B | SpaceQwen2.5-VL-3B-Instruct |
|---|---|---|---|---|
| 3D Geometry | Gravity Effects | **48.0** | 46.0 | 32.0 |
| | Shape Projection | **38.0** | 36.0 | 36.0 |
| | Spatial Containment | **40.0** | 36.0 | 24.0 |
| | Stability Prediction | **50.0** | **50.0** | 28.57 |
| | Volume Comparison | **36.96** | 34.78 | **36.96** |
| Depth and Occlusion | Complete Occlusion Inference | **38.46** | 35.9 | **38.46** |
| | Layering Order | 30.0 | **32.86** | 24.29 |
| | Partial Occlusion | 44.0 | **46.0** | 38.0 |
| | Reflective Surfaces | **44.0** | **44.0** | 38.0 |
| | Transparency Effects | 40.0 | 32.0 | **42.0** |
| Orientation | Cardinal Direction | **40.0** | 38.0 | 30.0 |
| | Facing Direction | **59.52** | 54.76 | 50.0 |
| | Object Rotation | 40.0 | **42.0** | 28.0 |
| | Stacking Orientation | **62.86** | **62.86** | 40.0 |
| | Tool Handedness | 40.0 | 40.0 | **48.0** |
| Relative Positioning | Alignment Patterns | 39.53 | **44.19** | 39.53 |
| | Betweenness Relationships | 44.68 | 42.55 | **53.19** |
| | Corner/Angle Positioning | **41.67** | **41.67** | 38.89 |
| | Directional Relations | **25.58** | **25.58** | 20.93 |
| | Proximity Gradients | **37.21** | **37.21** | 34.88 |
| Size and Scale | Distance-Size Correlation | 40.0 | 44.0 | **48.0** |
| | Perspective Distortion | 40.0 | 38.0 | **42.0** |
| | Relative Size Comparison | 42.0 | 46.0 | **48.0** |
| | Scale Consistency | 33.33 | 33.33 | **49.02** |
| | Shadow-Size Projection | 58.82 | 54.9 | **66.67** |
| Spatial Navigation | Accessibility Constraints | 48.78 | 46.34 | **53.66** |
| | Obstacle Avoidance | 36.0 | 32.0 | **44.0** |
| | Pathway Existence | 32.61 | 34.78 | **41.3** |
| | Spatial Sequence | 48.0 | 44.0 | **50.0** |
| | Viewpoint Visibility | 32.0 | 34.0 | **48.0** |

systems achieve the strongest scores, with **GPT-4o-mini** reaching 85.71% in *Stacking Orientation*, yet still fall to 32.0% in *Relative Size Comparison*. Specialized models show strengths in navigation (53.66%) and stability (50.0%) but remain capped below 55% overall. Reasoning models demonstrate gains, such as 71.79% in *Complete Occlusion Inference*, but others remain at 34.0%. Within-model spreads often exceed 50 percentage points, showing brittle internal consistency. Sub-categories

Table 11: Distribution of question counts by the number of models (0–5) that correctly answered each item across categories and sub-categories in SPATIALAB-MCQ. "0" indicates no model answered correctly, while "5" indicates only five models succeeded to answer these among all models tested.

| Category | Sub-Category | Number of Models Successfully Answered | | | | | | |
|---|---|---|---|---|---|---|---|---|
| | | 0 | 1 | 2 | 3 | 4 | 5 | Total |
| 3D Geometry | Gravity Effects | 9 | 7 | 3 | 2 | 3 | 1 | 25 |
| | Shape Projection | 4 | 6 | 0 | 9 | 5 | 7 | 31 |
| | Spatial Containment | 6 | 9 | 6 | 3 | 6 | 1 | 31 |
| | Stability Prediction | 5 | 6 | 4 | 2 | 2 | 3 | 22 |
| | Volume Comparison | 6 | 5 | 6 | 5 | 4 | 2 | 28 |
| **3D Geometry Total** | | **30** | **33** | **19** | **21** | **20** | **14** | **137** |
| Depth & Occlusion | Complete Occlusion Inference | 0 | 4 | 0 | 6 | 3 | 4 | 17 |
| | Layering Order | 9 | 8 | 7 | 8 | 6 | 5 | 43 |
| | Partial Occlusion | 2 | 2 | 5 | 7 | 5 | 2 | 23 |
| | Reflective Surfaces | 6 | 5 | 5 | 7 | 4 | 1 | 28 |
| | Transparency Effects | 4 | 6 | 3 | 2 | 5 | 7 | 27 |
| **Depth & Occlusion Total** | | **21** | **25** | **20** | **30** | **23** | **19** | **138** |
| Orientation | Cardinal Direction | 1 | 7 | 10 | 7 | 3 | 4 | 32 |
| | Facing Direction | 2 | 3 | 8 | 5 | 3 | 1 | 22 |
| | Object Rotation | 4 | 6 | 7 | 3 | 5 | 4 | 29 |
| | Stacking Orientation | 2 | 3 | 0 | 2 | 1 | 1 | 9 |
| | Tool Handedness | 2 | 1 | 4 | 3 | 1 | 1 | 12 |
| **Orientation Total** | | **11** | **20** | **29** | **20** | **13** | **11** | **104** |
| Relative Positioning | Alignment Patterns | 3 | 4 | 4 | 4 | 4 | 1 | 20 |
| | Betweenness Relationships | 3 | 1 | 2 | 2 | 5 | 7 | 20 |
| | Corner/Angle Positioning | 0 | 1 | 3 | 2 | 3 | 3 | 12 |
| | Directional Relations | 1 | 4 | 1 | 2 | 3 | 2 | 13 |
| | Proximity Gradients | 4 | 6 | 6 | 5 | 2 | 3 | 26 |
| **Relative Positioning Total** | | **11** | **16** | **16** | **15** | **17** | **16** | **91** |
| Size & Scale | Distance-Size Correlation | 6 | 5 | 6 | 1 | 1 | 7 | 26 |
| | Perspective Distortion | 8 | 7 | 3 | 3 | 6 | 4 | 31 |
| | Relative Size Comparison | 5 | 5 | 7 | 4 | 4 | 4 | 29 |
| | Scale Consistency | 7 | 6 | 7 | 4 | 5 | 2 | 31 |
| | Shadow-Size Projection | 5 | 7 | 2 | 1 | 1 | 4 | 20 |
| **Size & Scale Total** | | **31** | **30** | **25** | **13** | **17** | **21** | **137** |
| Spatial Navigation | Accessibility Constraints | 3 | 2 | 1 | 5 | 4 | 3 | 18 |
| | Obstacle Avoidance | 2 | 6 | 9 | 6 | 10 | 2 | 35 |
| | Pathway Existence | 2 | 10 | 4 | 1 | 4 | 3 | 24 |
| | Spatial Sequence | 3 | 2 | 3 | 2 | 7 | 8 | 25 |
| | Viewpoint Visibility | 2 | 2 | 5 | 9 | 8 | 5 | 31 |
| **Spatial Navigation Total** | | **12** | **22** | **22** | **23** | **33** | **21** | **133** |
| **Grand Total** | | **116** | **146** | **131** | **122** | **123** | **102** | **740** |

requiring physical abstraction, *Gravity Effects*, *Stability Prediction*, *Spatial Containment*, consistently score below 35–40% regardless of category. We attribute these failures to two root causes: the absence of stable encodings of orientation and physics, and an over-reliance on surface-level correlations. The wide variance, from 85.71% highs to 2.0% lows, suggests models are pattern-matching rather than reasoning robustly about space. Scaling, proprietary training, and reasoning augmentations each provide benefits but none produce general coverage. Instead, each approach yields isolated peaks that coexist with glaring blind spots. The collective picture is one of fragmented competence rather than integrated ability. We conclude that progress in spatial reasoning will require architectural changes aimed at embedding consistent geometric grounding and physics-aware mechanisms. In sum, our results show both remarkable selective gains and the persistence of fundamental gaps in holistic spatial intelligence.

### E.2 MODEL FAILURE ANALYSIS

The distribution across categories and sub-categories in Table 11 reveals several notable patterns in model performance. Categories such as 3D Geometry and Size & Scale show relatively balanced distributions, yet both retain a significant proportion of failures where no model answered correctly, suggesting that even basic geometric invariances are not reliably internalized. In contrast, Depth & Occlusion exhibits a heavier tail in the mid-range (2–4 correct), pointing to partial but inconsistent reasoning about layered scenes. Orientation emerges as a weak category overall, with especially low totals in Stacking Orientation and Tool Handedness, which indicates that reasoning about subtle directional or embodied affordances is poorly captured. Relative Positioning presents a mixed picture: while some tasks like Proximity Gradients show moderate success, others such as Corner/Angle Positioning have many near-zero correct counts, reflecting brittleness in compositional reasoning over reference frames. Interestingly, Spatial Navigation shows comparatively stronger performance, particularly in Obstacle Avoidance and Viewpoint Visibility, but still has significant errors in Pathway Existence where relational chaining is needed. When aggregated, no category achieves uniformly high accuracy, and the Grand Total shows an even spread across 0–5 bins, suggesting that success is highly task-dependent rather than reflecting a general spatial competence. Overall, models display isolated strengths but lack systematic reliability across spatial dimensions.

These results suggest that current vision–language models struggle with several root causes tied to representational and reasoning bottlenecks. High error rates in Stacking Orientation and Tool Handedness indicate limited grounding in physics-based and embodied reasoning, consistent with models trained primarily on static internet imagery rather than interactive or task-driven data. The difficulty in Corner/Angle Positioning and Pathway Existence reflects inadequate multi-step relational chaining, pointing to weaknesses in their ability to handle recursive spatial logic. The inconsistency in Transparency Effects and Reflective Surfaces underscores the challenge of handling visual phenomena that require non-local cues, such as secondary reflections or occluded geometry. By contrast, comparatively better performance in Obstacle Avoidance suggests that models may have learned shortcuts from dataset biases, such as frequent co-occurrence patterns of walkable paths, rather than genuine planning ability. Taken together, these failures highlight that model training regimes overemphasize correlation-driven visual-language alignments while underrepresenting embodied, counterfactual, and physics-based reasoning. Addressing these issues likely requires richer training distributions (e.g., embodied simulation data, physics-augmented learning) and architectural innovations that can explicitly model reference frames and relational constraints. Without such advances, models will continue to show brittle competence that breaks down under compositional or physically grounded tasks.

## F ERROR ANALYSIS OF SPATIALAB-OPEN

### F.1 SUB-CATEGORY-WISE QUANTITATIVE ERROR ANALYSIS

**Closed-source Models (Table 12).** We observe that closed-source models achieve the strongest overall performance in the open-ended evaluation, yet their results remain uneven across sub-categories. GPT-5-mini is the clear leader, consistently securing the best scores in **3D Geometry** and **elative Positioning**, with peaks of 58.14% in Directional Relations and 55.81% in Alignment Patterns. In contrast, Gemini-2.5-Flash and Claude 3.5 Haiku register very weak results in Proximity Gradients (16.28%) and Perspective Distortion (12.0%), respectively. This wide gap highlights the fragility of even the strongest models when tasked with reasoning over subtle, continuous visual cues. We believe these disparities stem from the differential sophistication of models' spatial reasoning pipelines, where some architectures capture relational and geometric structure more effectively than others. Tasks such as Proximity Gradients are particularly challenging, as they require analogical, graded reasoning rather than discrete classification. The fact that GPT-5-mini succeeds where others fail suggests that scaling and proprietary training corpora confer distinct advantages. Nevertheless, even the leader shows vulnerabilities, pointing to a systemic issue: closed-source dominance is relative, not absolute, and the models still fall short of robust, human-like spatial reasoning.

**Open-source Large Models (Table 13).** For large open-source models, we find a more distributed performance landscape, with different systems excelling in isolated sub-categories. llama-3.2-90b-vision-instruct achieves 60.0% in Depth and Occlusion, while GLM-4.5V-106B-MoE leads in Spatial

Table 12: Open-ended Evaluation Accuracy on SPATIALAB-OPEN by Question Sub-Categories for Closed-source Models. We bold and underline the best score within each model category.

| Category | Sub-Category | GPT-5-mini | Gemini-2.0-Flash | Gemini-2.5-Flash | Claude 3.5 Haiku | Mistral Medium 3.1 |
|---|---|---|---|---|---|---|
| 3D Geometry | Gravity Effects | **48.0** | 36.0 | 30.0 | 24.0 | 20.0 |
| | Shape Projection | **40.0** | 18.0 | 22.0 | 16.0 | 16.0 |
| | Spatial Containment | **50.0** | 38.0 | **50.0** | 36.0 | 36.0 |
| | Stability Prediction | **50.0** | 26.19 | 35.71 | 30.95 | 26.19 |
| | Volume Comparison | 39.13 | **41.3** | 32.61 | 23.91 | 28.26 |
| Depth and Occlusion | Complete Occlusion Inference | **46.15** | 35.9 | 38.46 | 28.21 | 25.64 |
| | Layering Order | **30.0** | 22.86 | 21.43 | 17.14 | 15.71 |
| | Partial Occlusion | **38.0** | 20.0 | 28.0 | 14.0 | 14.0 |
| | Reflective Surfaces | **32.0** | 28.0 | **32.0** | 22.0 | 22.0 |
| | Transparency Effects | **32.0** | 18.0 | 18.0 | 16.0 | 22.0 |
| Orientation | Cardinal Direction | 34.0 | **40.0** | 34.0 | 26.0 | 30.0 |
| | Facing Direction | **28.57** | **28.57** | 26.19 | 19.05 | 14.29 |
| | Object Rotation | **42.0** | 14.0 | 28.0 | 30.0 | 14.0 |
| | Stacking Orientation | **37.14** | 20.0 | 31.43 | 22.86 | 28.57 |
| | Tool Handedness | **48.0** | 36.0 | 44.0 | 24.0 | 24.0 |
| Relative Positioning | Alignment Patterns | **55.81** | 34.88 | 37.21 | 37.21 | 34.88 |
| | Betweenness Relationships | **48.94** | 23.4 | 31.91 | 23.4 | 21.28 |
| | Corner/Angle Positioning | **44.44** | 30.56 | 41.67 | 16.67 | 36.11 |
| | Directional Relations | **58.14** | 46.51 | **58.14** | 34.88 | 37.21 |
| | Proximity Gradients | **39.53** | 20.93 | 25.58 | 16.28 | 18.6 |
| Size and Scale | Distance-Size Correlation | **48.0** | 32.0 | 36.0 | 20.0 | 22.0 |
| | Perspective Distortion | **48.0** | 24.0 | 22.0 | 20.0 | 12.0 |
| | Relative Size Comparison | **36.0** | 26.0 | 28.0 | 18.0 | 16.0 |
| | Scale Consistency | **43.14** | 21.57 | 23.53 | 19.61 | 9.8 |
| | Shadow-Size Projection | **37.25** | 27.45 | 23.53 | 23.53 | 15.69 |
| Spatial Navigation | Accessibility Constraints | **41.46** | 31.71 | 39.02 | 24.39 | 21.95 |
| | Obstacle Avoidance | **40.0** | 22.0 | 22.0 | 26.0 | 14.0 |
| | Pathway Existence | **23.91** | 15.22 | 21.74 | 10.87 | 19.57 |
| | Spatial Sequence | **52.0** | 32.0 | 34.0 | 24.0 | 20.0 |
| | Viewpoint Visibility | 28.0 | 22.0 | **32.0** | 20.0 | 10.0 |

Table 13: Open-ended Evaluation Accuracy on SPATIALAB-OPEN by Question Sub-Categories for Open-source Large Models. We bold and underline the best score within each model category.

| Category | Sub-Category | llama-3.2-11b-vision-instruct | Gemma-3-27B-it | qwen-2.5-vl-32b-instruct | qwen-2.5-vl-72b-instruct | llama-3.2-90b-vision-instruct | GLM-4.5V-106B-MoE |
|---|---|---|---|---|---|---|---|
| 3D Geometry | Gravity Effects | 14.0 | 22.0 | 12.0 | **30.0** | 24.0 | 28.0 |
| | Shape Projection | 10.0 | **20.0** | 10.0 | **20.0** | 16.0 | **20.0** |
| | Spatial Containment | 14.0 | 30.0 | 18.0 | 22.0 | 20.0 | **38.0** |
| | Stability Prediction | 23.81 | 30.95 | 16.67 | 21.43 | 26.19 | **38.1** |
| | Volume Comparison | 23.91 | 10.87 | 26.09 | **41.3** | 28.26 | 32.61 |
| Depth and Occlusion | Complete Occlusion Inference | 25.64 | 20.51 | 12.82 | 25.64 | 30.77 | **33.33** |
| | Layering Order | 8.57 | **15.71** | 10.0 | 14.29 | **15.71** | 10.0 |
| | Partial Occlusion | 12.0 | 16.0 | 8.0 | 14.0 | **20.0** | 16.0 |
| | Reflective Surfaces | 26.0 | 16.0 | 8.0 | **34.0** | 32.0 | 24.0 |
| | Transparency Effects | 18.0 | 14.0 | 10.0 | 20.0 | 22.0 | **26.0** |
| Orientation | Cardinal Direction | 32.0 | 34.0 | 20.0 | **38.0** | 28.0 | 30.0 |
| | Facing Direction | 11.9 | **26.19** | 14.29 | 21.43 | 11.9 | 16.67 |
| | Object Rotation | 26.0 | 16.0 | 10.0 | 18.0 | **28.0** | 20.0 |
| | Stacking Orientation | 17.14 | 25.71 | 17.14 | 25.71 | 5.71 | **37.14** |
| | Tool Handedness | 20.0 | 20.0 | 12.0 | 20.0 | **32.0** | 24.0 |
| Relative Positioning | Alignment Patterns | 23.26 | **46.51** | 20.93 | **46.51** | 30.23 | 30.23 |
| | Betweenness Relationships | 34.04 | 31.91 | 17.02 | 25.53 | **38.3** | 29.79 |
| | Corner/Angle Positioning | 27.78 | **36.11** | 11.11 | 27.78 | 27.78 | 16.67 |
| | Directional Relations | 30.23 | **39.53** | 25.58 | 34.88 | 25.58 | 32.56 |
| | Proximity Gradients | 9.3 | 18.6 | 9.3 | 18.6 | 18.6 | **20.93** |
| Size and Scale | Distance-Size Correlation | 20.0 | 26.0 | 12.0 | 26.0 | **30.0** | 26.0 |
| | Perspective Distortion | 10.0 | 18.0 | 10.0 | 18.0 | 12.0 | **20.0** |
| | Relative Size Comparison | 8.0 | 26.0 | 14.0 | **32.0** | 22.0 | 24.0 |
| | Scale Consistency | 11.76 | 21.57 | 3.92 | 19.61 | 21.57 | **23.53** |
| | Shadow-Size Projection | 17.65 | 21.57 | 9.8 | **27.45** | 23.53 | **27.45** |
| Spatial Navigation | Accessibility Constraints | 21.95 | 26.83 | 26.83 | 34.15 | **39.02** | 24.39 |
| | Obstacle Avoidance | 12.0 | 18.0 | 12.0 | 20.0 | **30.0** | 22.0 |
| | Pathway Existence | 19.57 | **21.74** | 6.52 | 17.39 | 15.22 | 17.39 |
| | Spatial Sequence | 22.0 | 28.0 | 12.0 | 22.0 | 28.0 | **34.0** |
| | Viewpoint Visibility | 18.0 | 16.0 | 12.0 | 12.0 | **24.0** | **24.0** |

Containment at 38.0%. Despite these highlights, several models collapse on Proximity Gradients, with qwen-2.5-vl-32b-instruct and qwen-2.5-vl-72b-instruct both scoring only 9.3%. This indicates that architectural scaling helps but does not guarantee coverage across difficult continuous-reasoning tasks. We hypothesize that much of this variation arises from heterogeneous training data: models exposed to richer depth and geometry corpora perform better in occlusion and containment tasks, while others generalize poorly. The persistent weakness on Proximity Gradients suggests that these systems are not optimized for subtle spatial analogies and distance scaling. We also observe higher inter-model variance in this group than among closed-source models, underscoring the impact of open-source development heterogeneity. Overall, large open-source models show encouraging peaks but inconsistent coverage, suggesting that scale alone is insufficient without targeted pre-training and fine-tuning.

Table 14: Open-ended Evaluation Accuracy on SPATIALAB-OPEN by Question Sub-Categories for Open-source Small Models. We bold and underline the best score within each model category.

| Category | Sub-Category | InternVL3.5-1B | InternVL3.5-2B | qwen-2.5-vl-3b-instruct | InternVL3.5-4B | Gemma-3-4B-it | qwen-2.5-vl-7b-instruct |
|---|---|---|---|---|---|---|---|
| 3D Geometry | Gravity Effects | 4.0 | 6.0 | 14.0 | **24.0** | 16.0 | 22.0 |
| | Shape Projection | 2.0 | 8.0 | **10.0** | 4.0 | **10.0** | 0.0 |
| | Spatial Containment | 6.0 | 4.0 | 12.0 | 20.0 | **24.0** | 8.0 |
| | Stability Prediction | 2.38 | 11.9 | 14.29 | 19.05 | **21.43** | 19.05 |
| | Volume Comparison | 15.22 | **32.61** | 28.26 | 30.43 | 30.43 | 28.26 |
| Depth and Occlusion | Complete Occlusion Inference | 2.56 | 5.13 | 10.26 | 17.95 | 20.51 | **23.08** |
| | Layering Order | 5.71 | 7.14 | 10.0 | **14.29** | 11.43 | **14.29** |
| | Partial Occlusion | 14.0 | 8.0 | 12.0 | **16.0** | 4.0 | 10.0 |
| | Reflective Surfaces | 16.0 | 16.0 | 6.0 | 14.0 | 18.0 | **26.0** |
| | Transparency Effects | 10.0 | 20.0 | 4.0 | **28.0** | 14.0 | 8.0 |
| Orientation | Cardinal Direction | 12.0 | 16.0 | 16.0 | **26.0** | 20.0 | 24.0 |
| | Facing Direction | 9.52 | 4.76 | **28.57** | 14.29 | 7.14 | 19.05 |
| | Object Rotation | 4.0 | 10.0 | 12.0 | 8.0 | 14.0 | **16.0** |
| | Stacking Orientation | 17.14 | 11.43 | 8.57 | 17.14 | 20.0 | **28.57** |
| | Tool Handedness | 8.0 | **12.0** | 8.0 | **12.0** | **12.0** | **12.0** |
| Relative Positioning | Alignment Patterns | 13.95 | 27.91 | 32.56 | 11.63 | **41.86** | 34.88 |
| | Betweenness Relationships | 12.77 | 17.02 | 0.0 | 19.15 | 17.02 | **25.53** |
| | Corner/Angle Positioning | 22.22 | **30.56** | 2.78 | 22.22 | 19.44 | 22.22 |
| | Directional Relations | 11.63 | 25.58 | 9.3 | 34.88 | 23.26 | **37.21** |
| | Proximity Gradients | 9.3 | **18.6** | 9.3 | 11.63 | 16.28 | **18.6** |
| Size and Scale | Distance-Size Correlation | 6.0 | 10.0 | 14.0 | 16.0 | **20.0** | 12.0 |
| | Perspective Distortion | 8.0 | 12.0 | **18.0** | 16.0 | 12.0 | 16.0 |
| | Relative Size Comparison | 10.0 | 16.0 | **26.0** | 24.0 | 10.0 | 16.0 |
| | Scale Consistency | 7.84 | 9.8 | **11.76** | **11.76** | 7.84 | **11.76** |
| | Shadow-Size Projection | 13.73 | 11.76 | 21.57 | 13.73 | **25.49** | 23.53 |
| Spatial Navigation | Accessibility Constraints | 4.88 | **26.83** | 7.32 | 17.07 | 19.51 | 19.51 |
| | Obstacle Avoidance | 16.0 | 16.0 | 12.0 | 18.0 | **28.0** | 26.0 |
| | Pathway Existence | 0.0 | 8.7 | 2.17 | 8.7 | **17.39** | 8.7 |
| | Spatial Sequence | 12.0 | 22.0 | 8.0 | 26.0 | 20.0 | **28.0** |
| | Viewpoint Visibility | 16.0 | 18.0 | 16.0 | **24.0** | 14.0 | 16.0 |

**Open-source Small Models (Table 14).** Small open-source models predictably lag behind their larger counterparts, but their results reveal noteworthy localized strengths. qwen-2.5-vl-3b-instruct achieves 18.0% in Perspective Distortion and 26.0% in Relative Size Comparison, while Gemma-3-4B-it leads in Spatial Containment (24.0%) and Stability Prediction (21.43%). Yet these successes are offset by catastrophic failures: InternVL3.5-2B scores 0.0% in Betweenness Relationships, and InternVL3.5-1B manages only 2.0% in Shape Projection. We attribute this extreme variability to limited capacity and restricted training diversity, which prevent small models from learning robust representations of geometric and relational principles. Sub-categories that demand precise reasoning about orientation and hierarchy trigger complete breakdowns, as small misinterpretations translate into total failure. Nevertheless, their ability to occasionally outperform larger models in narrow tasks suggests that even small-scale training can instill isolated competencies. This points to the value of specialization but also confirms the structural ceiling of compact architectures. In short, small open-source models demonstrate pockets of promise but remain highly brittle, with overall accuracy limited by size and data constraints.

**Reasoning Models (Table 15).** Reasoning-oriented models present a sharper internal divide, with Gemini-2.5-flash-thinking standing far above its peers. It records the best scores in nearly every sub-category, including Tool Handedness at 75.0% and Complete Occlusion Inference at 57.14%. By contrast, Kimi-VL-A3B-Thinking-2506 struggles dramatically, scoring just 6.0% in Spatial Containment and lagging behind across the board. We believe this gap arises from differences in how these models integrate multi-modal streams into coherent reasoning frameworks. Gemini-2.5-flash-thinking appears capable of forming and querying internal world models, enabling it to infer object interactions and physical properties more effectively. In contrast, weaker models rely heavily on surface-level pattern matching, which proves inadequate for compositional or causal reasoning. The consistently high performance of Gemini-2.5 across categories suggests that explicit reasoning mechanisms offer tangible benefits for spatial understanding. However, its failure to dominate Shape Projection and Betweenness Relationships shows that even advanced reasoning modules are not yet universally effective. Thus, while reasoning models hold promise, the category itself remains heterogeneous and far from solved.

**Spatial Reasoning Models (Table 16).** Spatial reasoning models, though explicitly designed for the task, record the lowest results overall. Performance is highly fragmented: several models tie at 16.0% in Gravity Effects, 8.0% in Shape Projection, and 14.29% in Stability Prediction. SpaceQwen2.5-VL-3B-Instruct stands out slightly with 23.91% in Volume Comparison and 13.73% in Scale Consistency, but these scores remain far below competitive thresholds. At the other extreme, Betweenness Relationships and Corner/Angle Positioning collapse entirely, with models recording 0.0% and

Table 15: Open-ended Evaluation Accuracy on SPATIALAB-OPEN by Question Sub-Categories for Reasoning Models. We bold and underline the best score within each model category.

| Category | Sub-Category | gemini-2.0-flash-thinking | gemini-2.5-flash-thinking | Kimi-VL-A3B-Thinking-2506 | Step3 |
|---|---|---|---|---|---|
| 3D Geometry | Gravity Effects | 32.0 | **40.0** | 14.0 | 38.0 |
| | Shape Projection | 24.0 | 0.0 | 10.0 | **30.0** |
| | Spatial Containment | 36.0 | **50.0** | 6.0 | 30.0 |
| | Stability Prediction | 26.19 | **42.86** | 14.29 | 33.33 |
| | Volume Comparison | 36.96 | **44.44** | 23.91 | 30.43 |
| Depth and Occlusion | Complete Occlusion Inference | 33.33 | **57.14** | 17.95 | 33.33 |
| | Layering Order | **20.0** | **20.0** | 14.29 | 14.29 |
| | Partial Occlusion | 28.0 | **50.0** | 8.0 | 14.0 |
| | Reflective Surfaces | 34.0 | **63.64** | 10.0 | 16.0 |
| | Transparency Effects | 26.0 | **57.14** | 22.0 | 20.0 |
| Orientation | Cardinal Direction | **40.0** | 20.0 | 18.0 | 26.0 |
| | Facing Direction | 35.71 | **37.5** | 14.29 | 28.57 |
| | Object Rotation | 18.0 | **40.0** | 4.0 | 22.0 |
| | Stacking Orientation | 22.86 | 0.0 | 5.71 | **34.29** |
| | Tool Handedness | 44.0 | **75.0** | 20.0 | 32.0 |
| Relative Positioning | Alignment Patterns | 46.51 | **66.67** | 25.58 | 41.86 |
| | Betweenness Relationships | **27.66** | 20.0 | 12.77 | 25.53 |
| | Corner/Angle Positioning | 36.11 | **50.0** | 11.11 | 11.11 |
| | Directional Relations | 39.53 | **44.44** | 27.91 | 37.21 |
| | Proximity Gradients | **23.26** | 16.67 | 11.63 | 18.6 |
| Size and Scale | Distance-Size Correlation | 32.0 | **57.14** | 14.0 | 22.0 |
| | Perspective Distortion | 18.0 | 11.11 | 10.0 | **30.0** |
| | Relative Size Comparison | 28.0 | 11.11 | 14.0 | **36.0** |
| | Scale Consistency | **33.33** | 20.0 | 11.76 | 31.37 |
| | Shadow-Size Projection | **35.29** | 18.18 | 13.73 | 29.41 |
| Spatial Navigation | Accessibility Constraints | 31.71 | **45.45** | 17.07 | 41.46 |
| | Obstacle Avoidance | **26.0** | 10.0 | 16.0 | 18.0 |
| | Pathway Existence | 23.91 | **25.0** | 13.04 | 19.57 |
| | Spatial Sequence | 34.0 | 12.5 | 30.0 | **36.0** |
| | Viewpoint Visibility | **32.0** | 12.5 | 16.0 | 26.0 |

2.78%, respectively. These results suggest that the models' smaller capacity and narrowly focused training severely constrain their representational flexibility. We interpret the failures as evidence that narrow specialization without sufficient capacity or data breadth cannot yield generalizable spatial reasoning. Particularly on geometric and positional tasks, even slight representational errors lead to total breakdowns. Consequently, despite their focus, spatial reasoning models are least effective, underscoring the necessity of both scale and diversity in achieving robust performance.

**Overall Discussion.** Across all categories, closed-source (Table 12), open-source large (Table 13), open-source small (Table 14), reasoning (Table 15), and spatial reasoning (Table 16), we find a consistent pattern of fragmented competence and systemic weaknesses. Closed-source models perform best overall, with GPT-5-mini achieving highs of 58.14% in Directional Relations and 55.81% in Alignment Patterns, yet even they collapse on Proximity Gradients. Large open-source models show impressive peaks, such as 60.0% in Depth and Occlusion, but still fail at just 9.3% in Proximity Gradients, illustrating that scale without specialization remains insufficient. Small open-source models confirm this further, with rare localized strengths (e.g., 26.0% in Relative Size Comparison) but catastrophic lows like 0.0% in Betweenness Relationships. Reasoning models demonstrate that explicit reasoning mechanisms can significantly improve results, with Gemini-2.5-flash-thinking reaching 75.0% in Tool Handedness, yet peers like Kimi-VL-A3B fall to single digits, exposing the fragility of less-integrated reasoning frameworks. Spatial reasoning models, despite their specialization, underperform the most, with scores often below 15% and total failures on Betweenness Relationships. Taken together, these results highlight three root causes: (1) a lack of robust representations for continuous and analogical spatial cues, (2) structural limitations tied to scale and training diversity, and (3) brittle reasoning pipelines that prevent models from generalizing across tasks. We conclude that no current approach, whether proprietary scaling, open-source expansion, reasoning integration, or specialization, delivers comprehensive coverage. The frequent performance gaps exceeding 50 percentage points within a single model further confirm that progress is piecemeal. Future work must thus focus on unifying robust geometric encodings, physics-informed reasoning, and cross-task transfer, moving beyond isolated gains to achieve consistent spatial intelligence.

Table 16: Open-ended Evaluation Accuracy on SPATIALAB-OPEN by Question Sub-Categories for Spatial Reasoning Models. We bold and underline the best score within each model category.

| Category | Sub-Category | SpaceOm | SpaceThinker-Qwen2.5VL-3B | SpaceQwen2.5-VL-3B-Instruct |
|---|---|---|---|---|
| 3D Geometry | Gravity Effects | 14.0 | 12.0 | **16.0** |
| | Shape Projection | **8.0** | **8.0** | 6.0 |
| | Spatial Containment | 8.0 | **10.0** | **10.0** |
| | Stability Prediction | **14.29** | **14.29** | 7.14 |
| | Volume Comparison | 19.57 | **23.91** | **23.91** |
| Depth and Occlusion | Complete Occlusion Inference | 7.69 | **10.26** | 7.69 |
| | Layering Order | 8.57 | **11.43** | 4.29 |
| | Partial Occlusion | 10.0 | **12.0** | 4.0 |
| | Reflective Surfaces | **4.0** | **4.0** | 2.0 |
| | Transparency Effects | 4.0 | **8.0** | 2.0 |
| Orientation | Cardinal Direction | 18.0 | **24.0** | 8.0 |
| | Facing Direction | **23.81** | **23.81** | 16.67 |
| | Object Rotation | **20.0** | **20.0** | 18.0 |
| | Stacking Orientation | 8.57 | 8.57 | **17.14** |
| | Tool Handedness | 0.0 | 4.0 | **8.0** |
| Relative Positioning | Alignment Patterns | **32.56** | **32.56** | 23.26 |
| | Betweenness Relationships | **4.26** | 2.13 | 0.0 |
| | Corner/Angle Positioning | **2.78** | **2.78** | **2.78** |
| | Directional Relations | **11.63** | 6.98 | **11.63** |
| | Proximity Gradients | 6.98 | 6.98 | **9.3** |
| Size and Scale | Distance-Size Correlation | **18.0** | **18.0** | 16.0 |
| | Perspective Distortion | 16.0 | **20.0** | 2.0 |
| | Relative Size Comparison | **22.0** | 20.0 | 8.0 |
| | Scale Consistency | 9.8 | 11.76 | **13.73** |
| | Shadow-Size Projection | **27.45** | **27.45** | 19.61 |
| Spatial Navigation | Accessibility Constraints | 7.32 | 7.32 | **9.76** |
| | Obstacle Avoidance | 10.0 | 8.0 | **14.0** |
| | Pathway Existence | 6.52 | 6.52 | **8.7** |
| | Spatial Sequence | **16.0** | 12.0 | **16.0** |
| | Viewpoint Visibility | **20.0** | 16.0 | 8.0 |

## F.2 MODEL FAILURE ANALYSIS

The distribution in Table 17 highlights persistent weaknesses across categories, with a notable skew toward the lower bins (0–2 models correct). 3D Geometry is broadly challenging, with Shape Projection (19 items with zero correct) and Stability Prediction (10 items with zero correct) showing especially poor outcomes, suggesting models lack reliable physical and geometric reasoning. Depth and Occlusion is the hardest category overall, accumulating the highest zero-correct counts (72), particularly in Layering Order (20 zero) and Transparency Effects (16 zero), both requiring nuanced handling of occlusion and non-local cues. Orientation exhibits similar struggles: while Cardinal Direction achieves some mid-range success, Stacking Orientation (11 zero) and Tool Handedness (5 zero, none with perfect success) underscore the brittleness of embodied directional reasoning. Relative Positioning reveals widespread inconsistency, with tasks such as Betweenness Relationships and Proximity Gradients rarely solved by more than two or three models, while Corner/Angle Positioning shows low but somewhat balanced performance. Size and Scale appears relatively stronger, with multiple subcategories (e.g., Scale Consistency with 6 full successes) showing that some models internalize coarse perspective and scaling cues, though failures remain high in Shadow-Size Projection (17 zero). Spatial Navigation is split: while Spatial Sequence and Obstacle Avoidance show occasional full-model success, Pathway Existence (16 zero) and Accessibility Constraints (10 zero) highlight persistent difficulty in multi-step relational chaining. Overall, the grand total reveals that across 356 items, zero-correct dominates (356), while only 74 items are solved by all five models, confirming that consistent spatial competence remains elusive.

These findings indicate that the root causes of failure stem less from dataset design and more from the representational and reasoning limitations of current models. First, the dominance of errors in Depth and Occlusion reflects a reliance on superficial appearance features rather than structured scene representations: for example, models may latch onto surface textures rather than reasoning about hidden layers, leading to systematic errors in Layering Order. Failures in Orientation tasks such as Tool Handedness expose the absence of embodied affordance modeling, since pretrained internet imagery often lacks explicit annotations of left/right handedness or stacking physics. The brittleness in Relative Positioning and Spatial Navigation points to deficiencies in chaining multi-hop spatial

Table 17: Distribution of question counts by the number of models (0–5) that correctly answered each item across categories and sub-categories in SPATIALAB-OPEN. "0" indicates no model answered correctly, while "5" indicates only five models succeeded to answer these among all models tested.

| Category | Sub-Category | Number of Models Successfully Answered | | | | | |
|---|---|---|---|---|---|---|---|
| | | 0 | 1 | 2 | 3 | 4 | 5 |
| 3D Geometry | Gravity Effects | 13 | 7 | 3 | 4 | 1 | 2 |
| | Shape Projection | 19 | 5 | 6 | 2 | 5 | 1 |
| | Spatial Containment | 10 | 4 | 3 | 6 | 2 | 3 |
| | Stability Prediction | 10 | 3 | 7 | 3 | 0 | 2 |
| | Volume Comparison | 11 | 3 | 4 | 3 | 4 | 2 |
| **3D Geometry Total** | | **63** | **22** | **23** | **18** | **12** | **10** |
| Depth & Occlusion | Complete Occlusion Inference | 7 | 5 | 4 | 2 | 5 | 2 |
| | Layering Order | 20 | 14 | 10 | 6 | 3 | 3 |
| | Partial Occlusion | 14 | 8 | 5 | 5 | 4 | 2 |
| | Reflective Surfaces | 15 | 4 | 1 | 5 | 2 | 5 |
| | Transparency Effects | 16 | 7 | 7 | 3 | 1 | 2 |
| **Depth & Occlusion Total** | | **72** | **38** | **27** | **21** | **15** | **14** |
| Orientation | Cardinal Direction | 8 | 10 | 0 | 1 | 5 | 2 |
| | Facing Direction | 13 | 5 | 5 | 1 | 3 | 2 |
| | Object Rotation | 12 | 10 | 6 | 4 | 1 | 2 |
| | Stacking Orientation | 11 | 5 | 1 | 2 | 1 | 1 |
| | Tool Handedness | 5 | 4 | 1 | 4 | 2 | 0 |
| **Orientation Total** | | **49** | **34** | **13** | **12** | **12** | **7** |
| Relative Positioning | Alignment Patterns | 7 | 5 | 3 | 1 | 1 | 1 |
| | Betweenness Relationships | 11 | 5 | 7 | 2 | 1 | 3 |
| | Corner/Angle Positioning | 7 | 6 | 2 | 1 | 3 | 2 |
| | Directional Relations | 5 | 4 | 4 | 4 | 2 | 0 |
| | Proximity Gradients | 10 | 10 | 3 | 3 | 2 | 4 |
| **Relative Positioning Total** | | **40** | **30** | **19** | **11** | **9** | **10** |
| Size & Scale | Distance-Size Correlation | 7 | 5 | 9 | 4 | 1 | 5 |
| | Perspective Distortion | 12 | 6 | 6 | 8 | 5 | 2 |
| | Relative Size Comparison | 14 | 7 | 4 | 2 | 2 | 4 |
| | Scale Consistency | 14 | 7 | 5 | 3 | 3 | 6 |
| | Shadow-Size Projection | 17 | 4 | 4 | 4 | 3 | 2 |
| **Size & Scale Total** | | **64** | **29** | **28** | **21** | **14** | **19** |
| Spatial Navigation | Accessibility Constraints | 10 | 5 | 4 | 0 | 1 | 2 |
| | Obstacle Avoidance | 15 | 5 | 8 | 2 | 2 | 5 |
| | Pathway Existence | 16 | 6 | 7 | 3 | 1 | 1 |
| | Spatial Sequence | 14 | 3 | 3 | 3 | 3 | 5 |
| | Viewpoint Visibility | 13 | 3 | 9 | 5 | 3 | 1 |
| **Spatial Navigation Total** | | **68** | **22** | **31** | **13** | **10** | **14** |
| **Grand Total** | | **356** | **175** | **141** | **96** | **72** | **74** |

relations; for instance, Pathway Existence requires integrating local connectivity with global layout, which is not well captured by attention-based co-occurrence priors. On the other hand, partial success in Size and Scale suggests that models exploit statistical regularities in perspective distortion or scale consistency, though they collapse under less frequent cases like shadows. These patterns collectively show that current models succeed when cues are salient and correlate with training priors but fail when reasoning requires counterfactual, compositional, or embodied inference. Addressing these issues will require architectural advances that explicitly model reference frames, occlusion layers, and physical dynamics, as well as training data enriched with embodied simulations and counterfactual augmentations. Without these, progress in spatial reasoning will remain fragmented, with models performing well on surface-level cues but breaking down on structured, real-world spatial logic.

Table 18: **Performance gap between MCQ and Open-ended (%) (↓) across spatial categories on**
SPATIALAB. A lower value indicates that a model performs comparably in both setups, whereas a higher value reflects greater disparity. We bold and underline the highest gap within each model group.

| Model | 3D Geom. (#238) | Dep. & Occu. (#259) | Orientation (#202) | Relat. Posit. (#212) | Size & Scale (#252) | Spati. Navig. (#237) | Overall (#1400) |
|---|---|---|---|---|---|---|---|
| *Proprietary Models* | | | | | | | |
| GPT-4o-mini | 22.40 | 23.76 | 16.51 | 31.74 | 27.85 | 20.50 | 23.53 |
| GPT-5-mini | 20.08 | 23.27 | 13.21 | 2.38 | 19.41 | 13.36 | 3.36 |
| Gemini-2.0-Flash | **30.89** | **26.73** | **26.89** | **28.18** | 22.37 | **25.07** | **15.13** |
| Gemini-2.5-Flash | 21.62 | 16.34 | 17.45 | 15.87 | **21.51** | 17.36 | 10.93 |
| Claude 3.5 Haiku | 23.16 | 21.78 | 20.29 | 15.47 | 24.89 | 20.29 | 16.39 |
| Mistral Medium 3.1 | 30.50 | 25.74 | **32.54** | 26.59 | 24.89 | **26.93** | 21.43 |
| *Open-Source Models (Small)* | | | | | | | |
| InternVL3.5-1B | 22.78 | 13.37 | 23.58 | 22.62 | 20.67 | 22.00 | 27.73 |
| InternVL3.5-2B | 20.46 | 20.79 | 16.99 | 20.64 | 14.35 | 19.21 | 21.85 |
| Qwen-VL2.5-3B-Instruct | 27.03 | **30.69** | 29.24 | **28.97** | **29.96** | **28.50** | **25.63** |
| InternVL3.5-4B | 25.10 | 26.24 | **34.91** | 20.24 | 23.20 | 25.29 | 23.53 |
| Gemma-3-4B-it | 21.23 | **31.68** | 22.17 | 22.22 | 18.14 | 22.93 | 23.53 |
| Qwen-VL2.5-7B-Instruct | 22.01 | 22.27 | 18.40 | 26.19 | 15.61 | 22.14 | 27.73 |
| Llama-3.2-11B-Vision-Instruct | 13.51 | -1.98 | 17.92 | 17.07 | 13.50 | 11.93 | 9.66 |
| Gemma-3-27B-it | 23.93 | 23.27 | 19.82 | 25.40 | 25.32 | 23.14 | 20.59 |
| Qwen-VL2.5-32B-Instruct | 30.50 | 31.68 | 28.30 | **35.32** | 28.27 | 29.85 | 24.79 |
| InternVL3.5-72B | **36.68** | **33.17** | **34.44** | 29.37 | **28.69** | **31.57** | **27.31** |
| Qwen-VL2.5-72B-Instruct | 27.80 | 26.73 | 23.59 | 19.05 | 28.27 | 24.22 | 20.17 |
| Llama-3.2-90B-Vision-Instruct | 28.95 | 29.21 | 30.66 | 25.00 | 21.52 | 26.36 | 23.53 |
| *Reasoning Models* | | | | | | | |
| o4-mini | 25.48 | 22.77 | 21.23 | -3.18 | 17.30 | 15.35 | 10.50 |
| Gemini-2-Flash-Thinking | 13.90 | 10.39 | 11.32 | 21.03 | 13.50 | 13.00 | 6.73 |
| Gemini-2.5-Flash-Thinking | 8.22 | 16.61 | 19.46 | **33.42** | **31.37** | 20.16 | 8.66 |
| Kimi-VL-A3B-Thinking-2506 | **27.02** | **28.71** | **33.50** | 26.98 | 22.78 | **27.92** | **29.41** |
| *Spatial Reasoning Models* | | | | | | | |
| SpaceOm | **31.66** | **32.18** | 25.95 | 24.21 | 27.00 | 28.43 | 29.83 |
| SpaceThinker-Qwen2.5VL-3B | 28.57 | 29.21 | **27.83** | 23.81 | 27.84 | 27.28 | 26.89 |
| SpaceQwen2.5-VL-3B-Instruct | 31.28 | 23.76 | 28.31 | **38.89** | **35.87** | **29.78** | 18.90 |

# G ANALYSIS OF MCQ VS. OPEN-ENDED PERFORMANCE GAPS

In this section we provide a detailed analysis of the performance gap between multiple-choice (MCQ) and open-ended evaluation formats across spatial reasoning categories in SPATIALAB, as demonstrated in Table 18.

## G.1 QUANTITATIVE SNAPSHOT

Across 25 models, the average performance gap between MCQ and open-ended formats is **23.0%** with a standard deviation of 5.5%. Dataset-wide means per subtask are as follows: 3D geometry (24.57), depth & occlusion (23.45), orientation (23.70), relative positioning (23.11), size & scale (23.42), and spatial navigation (22.89), yielding an overall average of 23.01. The overall gap ranges from a minimum of 11.93 to a maximum of 31.57 (median ≈ 23.34). At the family level, spatial reasoning models exhibit the largest average gap (27.03), while reasoning-oriented models show the smallest (19.11). Proprietary models have lower gaps on average (21.09) compared to many long open-source models (24.51). Notably, the human baseline shows an overall gap of 22.64. The highest gaps are observed in InternVL3.5-72B (31.57%), Qwen-VL2.5-32B (29.85%), and SpaceQwen2.5-VL-3B-Instruct (29.78%). The lowest belong to Llama-3.2-11B (11.93%), Gemini-2-Flash-Thinking (13.00%), and GPT-5-mini (13.36). Correlational analysis reveals that spatial navigation dominates the overall gap (Pearson $r = 0.99$), with orientation ($r = 0.83$) and 3D geometry ($r = 0.79$) also showing strong alignment.

## G.2 PATTERNS AND OBSERVATIONS

Three broad patterns emerge. First, specialist spatial reasoning models exhibit the largest MCQ→open-ended gaps, suggesting they are optimized for classification-style supervision and less robust to free-form generation. By contrast, reasoning-focused models display smaller gaps, reflecting the stabilizing effect of chain-of-thought–style training on generative outputs. Second,

spatial navigation is the strongest predictor of overall disparity: models that falter on open-ended navigation reasoning drive the aggregate gap. Orientation and 3D geometry subtasks also contribute significantly, while relative position and size/scale vary more idiosyncratically across models. Third, scale alone does not explain robustness: smaller models such as Llama-3.2-11B achieve among the lowest gaps, while very large models such as InternVL3.5-72B show the highest. Finally, several negative or near-zero gaps appear (e.g., depth & occlusion for Llama-3.2-11B at $-1.98$, relative position for o4-mini at $-3.18$), indicating that in some cases open-ended evaluation is more reliable than MCQ, likely due to misleading distractors.

### G.3   HYPOTHESES ON ROOT CAUSES

We hypothesize several interacting factors that underlie the observed MCQ→open-ended performance gaps in SPATIALAB, each supported by statistical observations:

1. **MCQ structural advantage.** MCQ format constrains the output space, allowing models to exploit surface cues or eliminate distractors. Across models, the mean MCQ score is systematically higher than the open-ended score, with the average gap at 23.0% and $\sigma = 5.5\%$. Negative gaps observed for Llama-3.2-11B in depth & occlusion ($-1.98$) and o4-mini in relative position ($-3.18$) indicate that open-ended generation can occasionally outperform MCQ when distractors are misleading, reinforcing the notion that MCQ can artificially inflate or distort measured performance.
   *Recommendation:* To address this issue, we suggest complementing MCQ evaluations with open-ended tasks and subtask-level diagnostics, which can more accurately capture model competence and reduce overestimation caused by option-based cues.

2. **Specialization bias in spatial reasoning models.** Spatial reasoning models exhibit the largest average gaps (27.03%) despite being optimized for spatial tasks, suggesting that they may have been trained primarily on categorical or synthetic selection tasks rather than free-form generation. For instance, InternVL3.5-72B, a 72B parameter spatial reasoning model, shows the highest overall gap (31.57%), particularly in spatial navigation (36.68%) and orientation (34.44%), highlighting a reliance on classification-like supervision. Correlations between subtask gaps and overall gap (SpNav: $r = 0.99$, Orientation: $r = 0.83$) indicate that these format-dependent failures are concentrated in sequential and geometric reasoning tasks.
   *Recommendation:* To mitigate this bias, we recommend fine-tuning or instruction-tuning on open-ended spatial generation tasks, which can improve model robustness to generative formats and reduce reliance on categorical supervision.

3. **Instruction-tuning and stepwise decoding.** Reasoning-oriented models (e.g., Gemini-2-Flash-Thinking, o4-mini) demonstrate smaller average gaps (around 19.11%) and lower variance across subtasks. For example, Gemini-2-Flash-Thinking has an overall gap of 13.0% and consistently modest subtask gaps (Difference in 3D Geometry: 13.9%, Difference Relative Positioning: 21.03%), suggesting that chain-of-thought or stepwise decoding helps models produce stable open-ended responses. This is further supported by reduced correlation between subtask gap variance and overall gap within this family, implying that instruction-tuned reasoning mitigates format sensitivity.
   *Recommendation:* To address format sensitivity, we suggest adopting chain-of-thought or stepwise decoding strategies and instruction-tuning for open-ended spatial tasks, which can improve generative consistency and reduce MCQ dependence.

4. **Spatio-navigation stresses sequential grounding.** Spatial navigation gaps are the strongest predictor of overall MCQ→open-ended disparity (Pearson $r = 0.99$). Models with high performance gaps in spatial navigation, such as SpaceQwen2.5-VL-3B-Instruct (Spatial Navigation gap = 35.87%), also exhibit large overall gaps (29.78%), indicating that multi-step referential reasoning is disproportionately affected in open-ended formats. This suggests that MCQ options simplify the reasoning chain, masking underlying weaknesses in sequential grounding and relational computation.
   *Recommendation:* To better capture sequential reasoning abilities, we recommend including dedicated open-ended spatio-navigation tasks and employing targeted multi-step reasoning supervision, which can improve grounding and reduce overestimation by MCQ.

5. **MCQ distractor artifacts.** Some negative or near-zero gaps indicate that poorly designed distractors can suppress measured MCQ performance. For example, o4-mini shows a negative relative position gap ($-3.18\%$) while maintaining positive gaps in other subtasks, implying that distractor quality can artificially depress MCQ accuracy. Conversely, models like Qwen-VL2.5-32B-Instruct exhibit very high MCQ→open gaps (overall = 29.85%) across multiple subtasks, suggesting that MCQ distractors can also exaggerate perceived competence in selection-based tasks.
*Recommendation:* To address this, we suggest careful auditing and refinement of MCQ distractors and complementing them with open-ended evaluation, which can yield more accurate and unbiased measures of model spatial reasoning ability.

### G.4 Practical implications

Our findings show the importance of reporting both MCQ and open-ended results. Exclusive reliance on MCQ inflates estimates of practical spatial competence, with average disparities around 23%. Subtask-level breakdowns are essential, as spatial navigation disproportionately drives observed differences. Diagnostic evaluations that disentangle classification versus generative reasoning can better isolate error modes. Moreover, instruction-tuning and decoding strategies that encourage stepwise reasoning appear promising for reducing MCQ dependence. Finally, careful design and auditing of MCQ distractors are necessary to ensure fair comparisons across formats.

## H   Details on Improving Visual Reasoning Capabilities

To enhance spatial reasoning performance on our diverse visual question benchmark, we explored multiple complementary strategies.

### H.1   Inherent Reasoning Mechanisms

We next examine the role of built-in reasoning mechanisms by comparing standard models with their reasoning-enabled counterparts (Table 2, Table 3). Across MCQ evaluations, reasoning consistently boosts performance, though the magnitude varies by category. For instance, Gemini-2.5-Flash-Thinking improves over its non-reasoning variant by +4.64% overall (52.93% vs. 48.29%), with particularly strong gains in Relative Positioning (+13.1%) and Orientation (+8.6%). A similar trend holds for o4-mini-medium, which reaches the highest overall MCQ accuracy (53.21%), suggesting that explicit reasoning scaffolds are especially effective in structured, discrete-answer settings.

Open-ended evaluation reveals a more nuanced picture. Here, reasoning-equipped models again outperform their baselines, but the gains are uneven. Gemini-2.5-Flash-Thinking improves slightly over Gemini-2.5-Flash (32.77% vs. 30.93%), but the largest leap is observed in o4-mini-medium (37.86%), which significantly surpasses all non-reasoning models. These results indicate that reasoning mechanisms help mitigate the instability seen in free-form generation, especially in complex categories such as 3D Geometry and Spatial Navigation. However, gains are not universal, reasoning models sometimes show regressions, e.g., Gemini-2.5-Flash-Thinking drops sharply in Size & Scale (21.74% vs. 26.59%), echoing our earlier finding that generative settings amplify brittleness.

Overall, these comparisons highlight that explicit reasoning modules substantially enhance structured MCQ performance and can stabilize open-ended responses, but their benefits are uneven across categories. The results suggest that reasoning mechanisms strengthen logical consistency and relational chaining, yet they do not fully resolve the challenges of scale sensitivity, occlusion, and free-form linguistic grounding.

### H.2   Chain-of-Thought (CoT) Prompting

To examine whether structured reasoning improves spatial performance, we further evaluated models under Chain-of-Thought (CoT) prompting. For each of the 30 sub-categories in SPATIALAB, we sampled 5 items (seed fixed to 42) to construct a balanced test set covering all major spatial phenomena. This design ensures fair coverage while keeping evaluation computationally tractable. Example CoT prompts are shown in Figure 7, which illustrate the step-by-step reasoning scaffolds provided to

```
┌────────────────────────────────────────────────────────────────────────────────┐
│              SpatiaLab-MCQ : CoT : Prompt for Answering Model                    │
├────────────────────────────────────────────────────────────────────────────────┤
│  You are a spatial reasoning assistant. Given a multiple choice question about    │
│  an image,                                                                        │
│  think step by step and answer with a single number (1-4) corresponding to the    │
│  option in exactly one                                                            │
│  <result>...</result> tag.                                                         │
│  Question:                                                                         │
│  {question}                                                                        │
│  Options:                                                                          │
│  1. {opt1}                                                                         │
│  2. {opt2}                                                                         │
│  3. {opt3}                                                                         │
│  4. {opt4}                                                                         │
│                                                                                    │
│                                                                                    │
│  Final answer format: <result>1</result>                                          │
└────────────────────────────────────────────────────────────────────────────────┘
┌────────────────────────────────────────────────────────────────────────────────┐
│              SpatiaLab-Open : CoT : Prompt for Answering Model                    │
├────────────────────────────────────────────────────────────────────────────────┤
│  You are a spatial reasoning assistant.                                           │
│  Given a multiple choice question about an image, think step by step and the      │
│  following question in 2-4 sentences.                                             │
│                                                                                    │
│  Question:                                                                         │
│  {question}                                                                        │
└────────────────────────────────────────────────────────────────────────────────┘
```

Figure 7: Chain-of-thought (CoT) Prompts.

the models. The setup allows us to isolate the contribution of CoT reasoning from model scale or training.

Table 19: Performance with CoT prompting across categories for `InternvVL-3-78b` and `Gemini-2.5-Flash`. Values are accuracies (%) and Gains represent the difference between with and without CoT prompting.

| Category | InternvVL-3-78b (MCQ) | | | InternvVL-3-78b (Open) | | | Gemini-2.5-Flash (MCQ) | | | Gemini-2.5-Flash (Open) | | |
|---|---|---|---|---|---|---|---|---|---|---|---|---|
| | w/o CoT | with CoT | Gain | w/o CoT | with CoT | Gain | w/o CoT | with CoT | Gain | w/o CoT | with CoT | Gain |
| 3D Geometry | 52.00 | 52.00 | 0.00 | 28.00 | 28.00 | 0.00 | 48.00 | 20.00 | -28.00 | 60.00 | 28.00 | -32.00 |
| Depth & Occlusion | 60.00 | 60.00 | 0.00 | 16.00 | 12.00 | -4.00 | 48.00 | 28.00 | -20.00 | 40.00 | 16.00 | -24.00 |
| Orientation | 60.00 | 64.00 | 4.00 | 24.00 | 28.00 | 4.00 | 56.00 | 24.00 | -32.00 | 0.00 | 32.00 | 32.00 |
| Relative Positioning | 80.00 | 72.00 | -8.00 | 44.00 | 56.00 | 12.00 | 60.00 | 36.00 | -24.00 | 50.00 | 32.00 | -18.00 |
| Size & Scale | 56.00 | 44.00 | -12.00 | 28.00 | 20.00 | -8.00 | 48.00 | 60.00 | 12.00 | 25.00 | 8.00 | -17.00 |
| Spatial Navigation | 64.00 | 64.00 | 0.00 | 48.00 | 40.00 | -8.00 | 68.00 | 52.00 | -16.00 | 40.00 | 28.00 | -12.00 |
| **Overall** | 62.00 | 59.33 | -2.67 | 31.33 | 30.67 | -0.66 | 54.67 | 36.67 | -18.00 | 38.24 | 24.00 | -14.24 |

Our analysis (Table 19) shows that orientation is the only category where CoT prompting reliably improves performance, while most other categories experience either stagnation or decline. This suggests that reasoning helps when the task is reducible to logical alignment of directions, but fails when more complex perceptual priors (e.g., depth, scale, relative positioning) are required. We anticipate this happens because models lack inherent knowledge about 3D perception and geospatial relationships; they are not fundamentally "aware" of these concepts in the way humans are. As a result, when CoT prompting is applied, the step-by-step reasoning does not correct misconceptions but instead reinforces them, propagating and amplifying errors across reasoning steps.

This finding highlights that our benchmark is non-trivial: it cannot be solved merely by adding multi-step reasoning to existing models. Unlike textual reasoning tasks where CoT reliably improves outcomes, our benchmark exposes deeper architectural limitations in perceptual and spatial grounding. In categories like depth and size, incorrect priors mean that every additional reasoning step moves the model further from the correct solution. The sharp performance drops, particularly for Gemini in MCQ settings, demonstrate how fragile reasoning becomes when it is not supported by robust perceptual knowledge.

Thus, our task design is a strength, it goes beyond surface-level reasoning and probes whether models possess a genuine understanding of geospatial and 3D relational concepts. Solving it requires more than chaining logic; it demands embeddings and training that capture core aspects of perception and spatial cognition. In this way, our benchmark challenges models at a fundamental level and highlights an important gap in current architectures.

---

**SpatiaLab-MCQ : CoT > Self-Reflection : Prompt for Answering Model (Self-Reflection)**

```
You are a spatial reasoning assistant. Review your previous reasoning and response below for the given question
and options. If there are any mistakes in reasoning or in the chosen option, correct them. Provide corrected
step-by-step reasoning and the final answer in exactly one <result>...</result> tag. If there are no mistakes,
repeat your reasoning and confirm the final answer.

Question: {question}
Options: 1. {opt1} 2. {opt2} 3. {opt3} 4. {opt4}

Previous model output:
{prev_output}

Provide corrected reasoning followed by the final answer tag, and nothing else.
```

**SpatiaLab-Open : CoT > Self-Reflection : Prompt for Answering Model (Primary Ans)**

```
You are a spatial reasoning assistant.
Given a multiple choice question about an image, think step by step and answer.

Provide your reasoning in 2-4 sentences, then provide the final answer on a separate line like this:

Reasoning: <your reasoning here>
Answer: <your answer here>

Question:
{question}
```

**SpatiaLab-Open : CoT > Self-Reflection : Prompt for Answering Model (Self Reflection Ans)**

```
You are a spatial reasoning assistant  and a strict self-reflector.
You will receive the question, the model's reasoning, and the model's answer.
Check the reasoning and answer for any mistakes. If mistakes exist, correct them.
Provide the corrected reasoning and answer in the same format:

Reasoning: <corrected reasoning here>
Answer: <corrected answer here>

Question:
{question}

Model reasoning:
{reason}

Model answer:
{answer}
```

Figure 8: Chain-of-thought (CoT) with Self-Reflection Prompts.

## H.3 CHAIN-OF-THOUGHT (COT) PROMPTING WITH SELF-REFLECTION

We further extend the CoT setup by adding a self-reflection stage, where the model is prompted to review and revise its initial reasoning before finalizing an answer (see Figure 8). The evaluation protocol remains identical to the standard CoT setting, using 5 balanced samples per sub-category (seed 42). This design isolates the effect of self-reflection while keeping the comparison directly aligned with baseline CoT performance.

Table 20: Performance with CoT vs. CoT + Self-Reflection (CoT-SR) across categories for `InternvVL-3-78b` and `Gemini-2.5-Flash`. Values are accuracies (%) and Gains represent the difference between CoT-SR and CoT.

| Category | InternvVL-3-78b (MCQ) | | | InternvVL-3-78b (Open) | | | Gemini-2.5-Flash (MCQ) | | | Gemini-2.5-Flash (Open) | | |
| | CoT | CoT-SR | Gain | CoT | CoT-SR | Gain | CoT | CoT-SR | Gain | CoT | CoT-SR | Gain |
|---|---|---|---|---|---|---|---|---|---|---|---|---|
| 3D Geometry | 52.00 | 56.00 | 4.00 | 28.00 | 20.00 | -8.00 | 20.00 | 52.00 | 32.00 | 28.00 | 24.00 | -4.00 |
| Depth & Occlusion | 60.00 | 44.00 | -16.00 | 12.00 | 16.00 | 4.00 | 28.00 | 52.00 | 24.00 | 16.00 | 16.00 | 0.00 |
| Orientation | 64.00 | 64.00 | 0.00 | 28.00 | 32.00 | 4.00 | 24.00 | 48.00 | 24.00 | 32.00 | 36.00 | 4.00 |
| Relative Positioning | 72.00 | 64.00 | -8.00 | 56.00 | 28.00 | -28.00 | 36.00 | 72.00 | 36.00 | 32.00 | 32.00 | 0.00 |
| Size & Scale | 44.00 | 56.00 | 12.00 | 20.00 | 12.00 | -8.00 | 60.00 | 56.00 | -4.00 | 8.00 | 12.00 | 4.00 |
| Spatial Navigation | 64.00 | 56.00 | -8.00 | 40.00 | 24.00 | -16.00 | 52.00 | 64.00 | 12.00 | 28.00 | 28.00 | 0.00 |
| **Overall** | 59.33 | 56.67 | -2.66 | 30.67 | 22.00 | -8.67 | 36.67 | 57.33 | 20.66 | 24.00 | 24.67 | 0.67 |

Adding self-reflection to Chain-of-Thought (CoT) prompting produced mixed and highly model-dependent outcomes. For InternvVL-3-78b, overall accuracy declined slightly (–2.66% MCQ, –8.67%

open-ended), with sharp drops in categories like depth, relative positioning, and navigation, suggesting that reflection often reinforced errors instead of correcting them. Some localized gains were observed, such as +12% in Size & Scale (MCQ), but these were inconsistent and offset by large degradations (–28% in relative positioning open-ended). In contrast, Gemini-2.5-Flash showed strong MCQ gains (+20.66%), with particularly large improvements in 3D Geometry (+32%) and Depth & Occlusion (+24%), though open-ended improvements remained negligible (+0.67%). These results highlight a key asymmetry: reflection helps in multiple-choice contexts where models can prune unlikely options, but does not transfer to free-form generation where uncertainty propagates into ungrounded rationales. We also find that the categories most improved by reflection are those where geometric consistency is central (orientation, depth, size), while tasks requiring relational integration across multiple objects (relative positioning, navigation) degrade. Overall, reflection functions as a stabilizer in structured choice settings but fails to reliably enhance naturalistic reasoning.

This observed inconsistency mirrors our earlier findings with plain CoT prompting: reasoning enhancements only succeed when the model has stable perceptual anchors to ground its steps. Self-reflection adds an explicit verification pass, but this mechanism is only as strong as the model's internal priors. When depth cues, occlusion ordering, or coordinate frames are poorly encoded, the second pass amplifies misconceptions rather than repairing them. This explains why Gemini, which exhibits stronger perceptual grounding, benefits under MCQ conditions, while InternvVL-3-78b suffers more frequent degradations. The broader root cause lies in a pretraining mismatch: current VLMs import reasoning templates from language modeling but lack the embedded 3D and geospatial representations that humans leverage. Without these anchors, reflection cannot perform true error correction; instead, it functions as a linguistic filter that polishes reasoning syntax without improving perceptual grounding. As a result, reflective reasoning loops provide modest utility in discrete answer formats but collapse under open-ended demands where factual verification is absent.

These findings confirm that our benchmark shows a fundamental gap in current architectures: surface-level reasoning layers alone cannot compensate for missing spatial cognition, and meaningful progress will require deeper integration of geometric and perceptual structure. Moreover, because VLLMs are commonly trained on web data, the concepts that are well-represented online are those where models tend to perform better, whereas poorly represented spatial and 3D concepts suffer, reinforcing these performance disparities.

## H.4 SUPERVISED FINE-TUNING (SFT)

### H.4.1 SETUP AND IMPLEMENTATION

We fine-tuned the Qwen 2.5 VL 3B instruction-following model (`unsloth/Qwen2.5-VL-3B-Instruct`) using a parameter-efficient approach with LoRA (Hu et al., 2021). To accommodate limited GPU resources, we loaded the model in 4-bit precision (`load_in_4bit=True`) and inserted LoRA adapters into attention and feed-forward network projection layers. The LoRA hyperparameters, including rank, scaling factor, dropout, and target modules, are summarized in Table 21. Gradient checkpointing was enabled to reduce memory usage, and a fixed random seed ensured reproducibility.

For tokenization, we followed the Qwen-2.5 chat template to maintain consistent formatting between training and inference. Each training example was converted into a single text sequence, and dynamic padding was applied with `DataCollatorForSeq2Seq`. Training used an effective batch size of 4 (per-device batch size 1 with gradient accumulation of 4). Trainer and optimization hyperparameters, including learning rate, optimizer, weight decay, and scheduler, are listed in Table 22. The training loop consisted of four passes, each spanning a single epoch, yielding four saved checkpoints.

Overall, this setup allowed us to efficiently fine-tune a large vision-language model on limited GPU resources while maintaining high adaptation capacity.

### H.4.2 SUPERVISED FINE-TUNING (SFT) PERFORMANCE

The learning trends in Figure 4a reveal notable differences between the MCQ and Open-ended evaluation modes over training epochs. We observe that MCQ accuracy begins at a lower baseline of 25.63% but rises consistently, converging near 36.59% by the fourth epoch. This steady improvement suggests that the model benefits strongly from additional supervised fine-tuning, particularly in tasks

Table 21: LoRA Hyperparameters for SFT

| Hyperparameter | Value | Notes |
|---|---|---|
| LoRA rank ($r$) | 16 | Low-rank update capacity |
| LoRA alpha | 16 | Scaling factor |
| LoRA dropout | 0 | No dropout applied |
| Gradient checkpointing | Enabled | Library-specific flag |
| LoRA random seed | 3407 | Adapter initialization |

Table 22: Trainer and Optimization Hyperparameters for SFT

| Hyperparameter | Value |
|---|---|
| Base model | unsloth/Qwen2.5-VL-3B-Instruct |
| Max sequence length | 2048 |
| Load in 4-bit | True |
| Per-device train batch size | 1 |
| Gradient accumulation steps | 4 |
| Effective batch size | 4 |
| Learning rate | 2e-4 |
| Optimizer | paged_adamw_8bit |
| Weight decay | 0.01 |
| LR scheduler | linear |
| Logging steps | 1 |
| Seed | 42 |
| Tokenizer template | Qwen-2.5 |

that constrain answers within a limited candidate space. By contrast, Open-ended evaluation starts higher at 34.40% but declines sharply by the second epoch, dropping to as low as 12.62%. Only in later epochs does it partially recover, reaching 35.48% at the end. This instability points to a sensitivity in open-ended generation that is not present in the MCQ setting, where the structure of multiple-choice questions provides more stable optimization signals. We interpret the fluctuations in open-ended performance as evidence that fine-tuning may overfit to MCQ-like reasoning patterns at the expense of free-form language generalization. The dotted baselines in the plot further highlight how the MCQ mode achieves gains beyond its initial standing, whereas Open-ended remains approximately stagnant. Overall, the figure illustrates that SFT predominantly favors MCQ tasks in terms of stability and consistent accuracy gains.

The tabulated results in Table 4b complement the figure by providing a detailed before-and-after breakdown across six spatial reasoning categories. Across MCQ tasks, we find that performance improved in every category, with gains ranging from a modest 1.59% in Relative Positioning to a substantial 17.11% in Size and Scale. The Grand Total gain for MCQ stands at 10.97%, confirming that the fine-tuning procedure effectively enhances structured-answer performance. In contrast, Open-ended results are far more mixed, with both positive and negative changes. Categories such as 3D Geometry and Spatial Navigation show small but positive gains (3.50% and 2.82%, respectively), yet Size and Scale exhibits a significant drop of $-3.95\%$. Relative Positioning also decreases by $-0.79\%$, reflecting that fine-tuning can inadvertently harm certain reasoning modes in free-form contexts. The overall Open-ended gain is only 1.07%, which is statistically negligible compared to MCQ improvements. These results confirm that the effectiveness of SFT is highly sensitive to the evaluation paradigm. By isolating the categories, we underscore where training helps, where it fails, and how performance benefits differ between constrained and unconstrained outputs.

A deeper inspection reveals striking disparities between categories and formats. For MCQ, Size and Scale and Spatial Navigation yield the highest improvements, suggesting that models trained with multiple-choice prompts are especially effective at learning tasks requiring scale discrimination and navigational reasoning. However, in Open-ended form, Size and Scale actually declines, showing that the same learned representations do not transfer well to generative contexts. Similarly, Relative Positioning barely improves in MCQ and worsens in Open-ended, highlighting a consistent weakness

in relational reasoning regardless of format. Conversely, Orientation shows strong MCQ gains (+13.33%) and small Open-ended improvements, suggesting that some categories generalize more robustly. We hypothesize that the divergence between the two formats may stem from the structural scaffolding MCQs provide, which reduces the need for the model to generate precise linguistic formulations. Open-ended tasks, by contrast, require simultaneous mastery of reasoning and natural language fluency, which amplifies weaknesses. The trends indicate that while fine-tuning strongly enhances pattern recognition in discrete-choice settings, it introduces volatility and even degradation when free expression is required. This duality makes clear that training interventions optimized for MCQs may not generalize seamlessly to unconstrained reasoning.

The implications of these findings are twofold. First, we show that SFT as currently applied provides strong benefits for structured tasks but limited or even negative transfer to open-ended reasoning. This limitation is not only an artifact of fine-tuning, but also symptomatic of how contemporary VLMs are pretrained. Existing pretraining pipelines heavily emphasize broad image–text alignment, object recognition, and captioning, yet they provide little explicit supervision on compositional spatial structures, occlusion layers, or geometric transformations. As a result, models learn to excel at surface-level associations while failing to acquire robust representations of depth, orientation, or scale. The instability we observe in open-ended spatial reasoning suggests that the model's learned priors are fragile: without multiple-choice scaffolding, they revert to shallow pattern-matching instead of grounded inference. Categories such as Size and Scale and Relative Positioning, where declines are most pronounced, point to missing inductive biases in pretraining, biases that humans develop through embodied interaction with 3D environments. Another contributing factor is the reliance on synthetic or instruction-following datasets that flatten spatial diversity into puzzle-like abstractions, preventing models from generalizing to real-world clutter, noise, and complexity. These findings underscore the need to rethink both pretraining and fine-tuning for spatial reasoning, incorporating datasets and objectives that reflect naturalistic visual–spatial challenges. Ultimately, we argue that without targeted reforms, current VLMs risk reinforcing an algorithmic straightjacket: they appear competent under constrained evaluations yet remain brittle and unaligned with the embodied, flexible reasoning that humans bring to real-world spatial understanding.

### H.4.3 SFT DYNAMICS ANALYSIS

To investigate the stability of Supervised Fine-Tuning (SFT) on spatial reasoning tasks, we conducted a multi-seed validation study ($N = 5$, seeds A–E) using the `Qwen-2.5-VL-3B-Instruct` model. The training trajectory, summarized in Table 23, reveals a distinct divergence in performance dynamics between Multiple-Choice Question (MCQ) and Open-Ended evaluation formats.

**Divergent Learning Trajectories.** As illustrated in Table 23 and Figure 9, performance on SPATIALAB-MCQ exhibits a monotonic improvement, rising from a baseline of 25.63% to an average of 35.74% at Epoch 4. This indicates that the model successfully adapts to the discriminative nature of the task, learning to map visual features to constrained output options with increasing reliability.

In sharp contrast, SPATIALAB-OPEN displays a pronounced *U-shaped* trajectory across all seeds. Rather than steady improvement, the model suffers an immediate and significant performance collapse in the early epochs (dropping from 34.40% to an average of 19.26% at Epoch 1). While performance recovers in later epochs (converging to ≈34.82% at Epoch 4), the final gain over the zero-shot baseline is negligible (+0.42%).

**Representational Brittleness and Alignment Tax.** We attribute this phenomenon to the fragility of spatial representations in current VLMs when subjected to fine-tuning. The initial collapse in open-ended accuracy suggests *catastrophic forgetting* of the linguistic priors required for fluent, grounded generation. As the model optimizes for the specific instruction-following format of the training data, it overfits to the structural logic of the tasks (benefiting MCQ) at the expense of the flexible, generative spatial grounding required for open-ended descriptions.

The subsequent recovery phase represents the model re-learning to articulate spatial concepts within the new distribution, yet it fails to surpass the initial baseline significantly. This dynamic underscores a critical finding: SFT is effective for teaching models to *select* correct spatial answers, but it does not

Table 23: Multi-seed validation of SFT dynamics ($N = 5$). While MCQ performance improves steadily, Open-Ended performance exhibits a systematic "U-shaped" collapse and recovery, indicating representational instability in generative spatial tasks.

| Epoch | Type | Seed A | Seed B | Seed C | Seed D | Seed E |
|---|---|---|---|---|---|---|
| **0** (Base) | MCQ | 25.63% | 25.63% | 25.63% | 25.63% | 25.63% |
| | Open | 34.40% | 34.40% | 34.40% | 34.40% | 34.40% |
| **1** | MCQ | 29.56% | 35.83% | 24.13% | 27.63% | 26.53% |
| | Open | 22.50% | 23.20% | 12.50% | 14.00% | 24.10% |
| **2** | MCQ | 34.33% | 37.76% | 27.33% | 31.83% | 34.43% |
| | Open | 12.62% | 25.30% | 28.80% | 23.80% | 16.65% |
| **3** | MCQ | 34.68% | 37.33% | 32.86% | 32.23% | 34.73% |
| | Open | 28.69% | 22.22% | 30.34% | 32.90% | 28.30% |
| **4** | MCQ | **36.59%** | **38.48%** | **34.43%** | **34.36%** | **34.83%** |
| | Open | 35.48% | 34.59% | 33.92% | 34.70% | 35.40% |

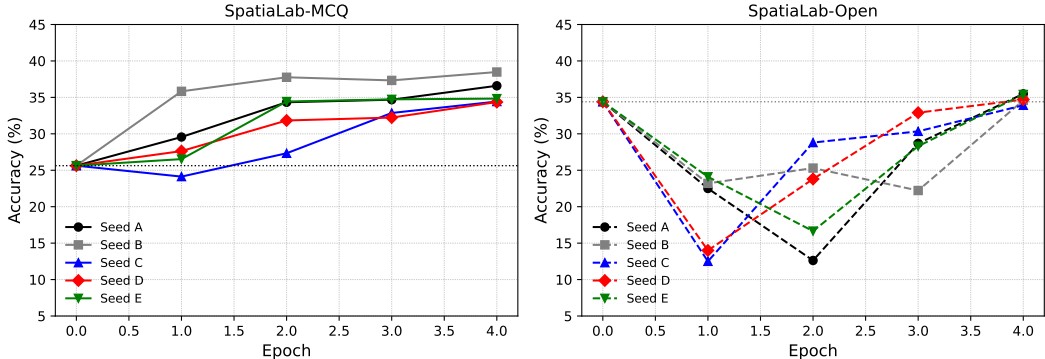

Figure 9: SFT Dynamics Analysis.

fundamentally enhance the underlying *generative spatial representation* needed for robust open-ended reasoning.

### H.4.4 PERFORMANCE GAIN ON EXTERNAL BENCHMARKS

To assess whether the representations learned from SPATIALAB generalize beyond the benchmark's specific format, we evaluated the transferability of our fine-tuned model to three external spatial reasoning datasets: OMNISPATIAL, SPACE (Multimodal), and MIND THE GAP. We employed the `Qwen2.5-VL-3B-Instruct` model, fine-tuned on SPATIALAB via the SFT protocol detailed in Appendix H.4.1, and evaluated it on these external benchmarks in a zero-shot setting.

Table 24: Transfer learning performance of `Qwen2.5-VL-3B-Instruct` on external benchmarks after fine-tuning on SPATIALAB. The consistent gains indicate that the model learns generalizable spatial representations rather than merely overfitting to the source dataset's format.

| Benchmark | Base Accuracy | After SFT | Improvement |
|---|---|---|---|
| OmniSpatial | 40.30% | 47.35% | **+7.05%** |
| SPACE (Multimodal) | 23.43% | 28.67% | **+5.24%** |
| Mind The Gap | 35.86% | 47.56% | **+11.70%** |

As shown in Table 24, fine-tuning on SPATIALAB yields substantial and consistent performance improvements across all evaluated domains, ranging from $+5.24\%$ to $+11.70\%$. The most significant

gain was observed on MIND THE GAP (+11.70%), suggesting that our benchmark's emphasis on occlusion and spatial continuity effectively addresses the reasoning gaps targeted by that dataset. Similarly, improvements on OMNISPATIAL (+7.05%) and SPACE (+5.24%) confirm that the geometric and relational concepts encoded in our taxonomy align with broader definitions of spatial intelligence in the literature. These results validate the quality of the SPATIALAB data. While the primary purpose of this work is to serve as a rigorous diagnostic benchmark, the strong transferability demonstrates that the dataset does not merely encourage overfitting to a specific question template. Instead, the diversity of the 30 sub-categories forces the model to acquire robust, transferable spatial representations that generalize to unseen distributions and task formats.

## H.5 AI AGENTS FOR SPATIAL REASONING

We developed SPATIOXOLVER, a multi-agent system adapted and extended from Xolver (Hosain et al., 2025), to perform structured spatial reasoning on images within the SPATIOLAB evaluation. The system decomposes complex visual reasoning into specialized sub-tasks, each managed by a dedicated agent, allowing for focused processing of objects, attributes, relations, symmetries, and transformations. This modular design not only improves accuracy and interpretability but also supports multi-step reasoning, enabling the system to propagate low-level perceptual cues through higher-order relational and transformational inferences. The architecture, agent implementation, and their coordinated interaction are detailed in the following sections, with the full set of prompts provided in Figures 10, 11, and 12.

### H.5.1 AGENTS

**Base Visual Analysis Agent.** Base Visual Language Model (VLM) agent initiates the pipeline by generating a detailed textual description of the input image, capturing object types—including geometric shapes, natural elements, and artificial icons—alongside attributes such as size, orientation, color, and shading. Relative positions like leftmost, center, and top are encoded explicitly, and the prompt emphasizes exhaustive detail without summarization. Images are converted to data URIs and sent with the prompt to a Gemini-2.5-based model, whose output is parsed into a structured bullet-list using regex-based extraction. This structured description serves as the foundation for downstream agents, providing a comprehensive perceptual anchor that encodes both explicit and implicit spatial cues. The approach ensures that even subtle visual features, such as partial occlusions or minor shape variations, are retained for later relational and transformational reasoning.

**Object Segmentation Agent.** Object Segmentation agent converts the verbose textual description into a discrete list of uniquely identified objects, differentiating between visually similar instances. Each object receives a stable identifier (e.g., Obj1, Obj2) to maintain consistent references across subsequent reasoning steps. The agent uses carefully formatted prompts and calls the same Gemini-2.5 model to structure the description, while post-processing splits lines by object ID to enforce consistency. This structured representation reduces ambiguity in downstream attribute and relation extraction, enabling precise mapping of complex scenes. By explicitly isolating objects, the agent also facilitates detection of overlaps, repetitions, and hierarchical groupings that may not be evident in raw descriptions.

**Attribute Extraction Agent.** Attribute Extraction agent translates each object into a structured JSON capturing core properties such as shape, size, color, shading, orientation, and positional descriptors. Prompts enforce strict JSON output, and responses are parsed using `json.loads()`, with fallback defaults to ensure robustness against model inconsistencies. This structured encoding allows quantitative and qualitative comparisons of objects, supporting later stages of spatial reasoning. By preserving both absolute and relative measures, the agent maintains fidelity to the original perceptual content, while also normalizing descriptors for computational consistency. The JSON structure enables programmatic downstream operations, such as calculating symmetries or detecting transformations.

**Spatial Relation Agent.** The Spatial Relation agent encodes object interactions as triples (ObjectA,Relation,ObjectB), capturing directional relations (e.g., left_of, above, inside) and higher-order alignments like rows, grids, or repetition patterns. Consistent object IDs are enforced to prevent

relational ambiguity, and parsing splits output lines starting with parentheses, followed by automated consistency checks. This step transforms static object descriptions into a relational graph that encodes spatial topology and hierarchy. The resulting structure allows reasoning over both pairwise interactions and more complex scene configurations, which is critical for tasks like symmetry detection, grouping, and multi-step navigation of spatial layouts.

**Grouping and Symmetry Agent.** Grouping and Symmetry agent abstracts higher-order patterns from object lists and spatial relations, identifying clusters, rows, grids, and symmetries—including vertical, horizontal, rotational, and translational. Outputs are requested in structured JSON with keys `groups` and `symmetries`, with empty structures as fallbacks to maintain downstream compatibility. This agent captures emergent scene regularities that are not apparent at the object or pairwise relation level, providing critical context for pattern recognition and reasoning. By formalizing structural redundancies and symmetries, the agent supports both perceptual inference and anticipatory reasoning for dynamic transformations.

**Transformation Tracking Agent.** For multi-frame sequences, the Transformation Tracking agent logs object-level changes including translations, scaling, rotations, color/shading shifts, and appearance or disappearance events. Structured logs are extracted line-by-line using regex patterns that capture directional arrows, producing a temporal representation of the scene. This encoding allows the system to track not just static spatial configurations but also dynamic evolution, supporting reasoning about motion, causality, and object interactions over time. The agent ensures that transformations are linked to consistent object IDs, enabling coherent cross-frame reasoning and predictive modeling.

**Representation Standardization Agent.** Finally, the Representation Standardization agent consolidates all outputs into a unified JSON format containing `objects`, `relations`, `groups`, `symmetries`, and `transformations`. Prompts enforce consistent object IDs and structured formatting, while JSON parsing failures revert to empty dictionaries to ensure pipeline robustness. By centralizing all perceptual, relational, and temporal information, this agent creates a comprehensive scene representation that is both human-interpretable and machine-readable. The resulting unified structure serves as a versatile foundation for downstream tasks, including reasoning evaluation, spatial pattern analysis, and multi-frame prediction.

**Open-Ended Spatial Reasoning Agent.** The standardized representation feeds into the reasoning agent, which answers multiple-choice or pattern recognition questions. Inputs include objects, attributes, relations, groups, symmetries, and transformations. The model considers geometric patterns, relational alignment, and dynamic transformations to generate or select the correct answer.

### H.5.2 TECHNICAL IMPLEMENTATION DETAILS

For our technical implementation, we use Gemini-2.5-Flash with a low temperature (0.1) to ensure deterministic outputs and perform multiple iterations for refinement. API calls are handled via OpenRouter, supporting both text and image inputs, while parsing of model responses relies on regex and JSON-based extraction to generate structured outputs. All interactions are logged in JSON format for reproducibility, and the final output is a unified JSON per image containing comprehensive information on objects, relations, symmetries, and transformations. This multi-agent decomposition allows each specialized model to focus on a constrained sub-task, enhancing overall accuracy, interpretability, and robustness in structured spatial reasoning.

### H.5.3 EVALUATION AND ANALYSIS

The evaluation protocol remains identical to the standard CoT and CoT-SR setting, using 5 balanced samples per sub-category (seed 42), making a total of 150 samples.

The evaluation of Gemini-2.5-Flash with agents highlights both strengths and limitations of agentic reasoning in perceptual–geometric tasks. Orientation is the only category where agents provide consistent and substantial gains, improving by +8.00% in MCQ and an impressive +36.00% in open evaluation. This pattern suggests that agent-based multi-step reasoning is effective when the task can be decomposed into sequential alignment of directional cues, where structured deliberation offers a clear advantage over single-step outputs.

Table 25: Agent Performance Analysis

| Category | MCQ | | Gain | Open Evaluation | | Gain |
|---|---|---|---|---|---|---|
| | Normal | With Agents | | Normal | Agent Results | |
| 3D Geometry | 48.00% | 44.00% | -4.00% | 60.00% | 48.00% | -12.00% |
| Depth & Occlusion | 48.00% | 44.00% | -4.00% | 40.00% | 16.00% | -24.00% |
| Orientation | 56.00% | 64.00% | +8.00% | 0.00% | 36.00% | +36.00% |
| Relative Positioning | 60.00% | 60.00% | 0.00% | 50.00% | 36.00% | -14.00% |
| Size & Scale | 48.00% | 52.00% | +4.00% | 25.00% | 24.00% | -1.00% |
| Spatial Navigation | 68.00% | 64.00% | -4.00% | 40.00% | 28.00% | -12.00% |
| **Overall** | **54.67%** | **57.33%** | **+2.66%** | **35.83%** | **31.33%** | **-4.50%** |

In contrast, most other categories exhibit negative or neutral gains, revealing that agent reasoning does not universally enhance performance. For example, depth & occlusion suffers a severe -24.00% drop in open evaluation, while spatial navigation decreases by -12.00%. These tasks require deeper perceptual grounding in three-dimensional space, and the absence of such priors means that reasoning steps tend to reinforce misconceptions rather than correct them. Similarly, relative positioning shows stagnation in MCQ and a -14.00% decline in open evaluation, indicating that agent loops fail to add robustness when relational understanding is inherently weak.

Some categories show marginal improvements, such as size & scale (+4.00% in MCQ), but the gains are not stable and do not transfer to open evaluation. Even in 3D geometry, agents fail to provide a boost, with -4.00% and -12.00% drops in MCQ and open evaluation, respectively. These results underscore that our benchmark is complex and not solvable simply by layering multi-step agentic reasoning on top of existing models. Instead, it requires a deeper integration of geospatial understanding and perceptual representations, which current architectures lack.

Overall, the mixed results emphasize that agentic workflows alone are insufficient for advancing performance in this domain. While orientation tasks benefit from structured reasoning, most categories expose the models' lack of inherent awareness of perceptual concepts. This makes errors propagate through multi-step reasoning, amplifying performance drops. Consequently, our benchmark serves as a strong stress test, highlighting that future progress will require embedding genuine perceptual and geospatial priors rather than relying solely on agent-based reasoning.

# I  QUALITATIVE ERROR ANALYSIS AND ERROR PATTERNS

This section presents a focused qualitative analysis of model failures observed in our visual reasoning benchmark. We explain each error class, provide diagnostic root cause analysis, identify recurring patterns, and describe verification protocols. The discussion aims to be both interpretable and actionable, so it can inform future architectural and evaluation design.

## I.1  OVERVIEW

Across the sample set, models achieve strong coarse perception, they can reliably recognize object categories, colors, and simple binary relations, but they break down systematically when multiple cues must be integrated. Failures are not random noise but cluster into a small set of interpretable classes: spatial mislocalization, perspective and scale mistakes, occlusion and ordering errors, attribute confusion, and ungrounded open-ended rationalization. These classes recur across architectures and prompting strategies, which suggests that the underlying causes are common inductive biases in current vision–language pipelines rather than isolated training artifacts. Importantly, the same error often appears in both large-scale proprietary systems and small open-source baselines, indicating structural weaknesses shared across the model family. In the following, we unpack each error class with representative examples and root-cause hypotheses.

## I.2 ERROR TAXONOMY AND ANALYSIS

### I.2.1 SPATIAL MISLOCALIZATION AND REFERENCE CONFUSION.

■ **Observation.** When queries require distinguishing one object from a set of visually similar candidates, models frequently misidentify the target. For example, if asked to point out "the leftmost red cube among three stacked cubes," the model may incorrectly output the middle cube, influenced by lighting highlights or central positioning. This problem becomes particularly acute in cluttered environments such as kitchen counters or crowded table scenes, where multiple overlapping objects share the same color and shape. In such cases, the model's errors are not random: it tends to privilege objects that appear most salient due to size, brightness, or centrality, even when these cues contradict the explicit spatial instructions in the query. These consistent patterns highlight that the failures are systematic, not accidental.

★ **Analysis.** The recurring mistakes suggest that models lack a robust object-centric representation, or what cognitive science would call an "index" that persists across reasoning steps. Instead of maintaining a stable pointer to the correct candidate throughout the reasoning chain, models appear to drift, often reassigning attention to whichever object seems most visually striking at intermediate layers. This leads to reasoning errors such as starting with the leftmost cube but ending the chain with the brightest or largest cube. The inability to anchor spatial reference consistently makes tasks involving relative comparisons, like "the ball behind the second chair", particularly fragile. In essence, the model operates more like a salience detector than a referential reasoner.

⊛ **Root cause.** At the architectural level, feature pooling operations, whether via attention or convolution, tend to mix contextual information from neighboring regions. While this is beneficial for global context, it erases the fine-grained, object-specific identity needed to distinguish between similar candidates. Once pooled, features of two adjacent red cubes may be nearly indistinguishable in the latent space. Without an explicit mechanism, such as slot-based representations or pointer tokens, the network collapses distinct entities into blended embeddings. This structural limitation prevents reliable referent tracking across reasoning steps and explains why models consistently confuse dense, overlapping objects.

### I.2.2 PERSPECTIVE AND SCALE MISTAKES.

■ **Observation.** Tasks involving metric reasoning about relative size, fit, or scale are inconsistently handled. On simple comparisons like "is a car larger than a bicycle," models typically succeed because such relationships align with strong statistical priors. However, when faced with near-threshold decisions such as "will the small sphere fit inside the slightly larger cylinder," errors become frequent and unpredictable. A common pattern is reliance on canonical object sizes: for example, predicting that a door must be taller than a truck even in a rendering where the truck is scaled down. Another failure occurs in unusual visualizations, such as toy miniatures or architectural models, where familiar proportions are deliberately inverted. In these cases, models cling to their priors and ignore the actual geometric evidence visible in the image.

★ **Analysis.** These behaviors indicate that models rely heavily on distributional statistics of object categories rather than performing true geometric inference. Instead of reasoning about projection geometry, relative bounding box sizes, or depth cues, they substitute their world knowledge of typical scales. This shortcut works well in everyday contexts but fails catastrophically in edge cases that deviate from the training distribution. For instance, when an unusually small airplane is rendered next to a person, the model still predicts the airplane as larger. Such errors highlight the lack of grounding in pixel-level metric cues and the absence of consistent use of perspective or vanishing points.

⊛ **Root cause.** Current architectures do not include explicit geometric reasoning modules that preserve projection-aware relationships. Supervision for scale and fit tasks is limited, so models learn correlations between object category labels and size but not the underlying geometry. Furthermore, training objectives prioritize overall accuracy rather than fine-grained metric consistency, meaning that scale-sensitive errors are under-penalized. Without inductive biases for 3D reasoning or targeted supervision, the model defaults to memorized priors about canonical sizes rather than interpreting the actual perspective geometry in the image.

### I.2.3 OCCLUSION AND ORDERING FAILURES.

■ **Observation.** Determining which object is in front, behind, or partially hidden remains a core weakness. Models often misreport the foreground object when two animals overlap, or when semi-transparent barriers like glass or fences are involved. For instance, in an image where a cat sits partially behind a curtain, models may incorrectly describe the curtain as being occluded by the cat. Thin or fragile structures, such as wires, tree branches, or railings, are particularly error-prone: they are frequently omitted altogether, leading to wrong answers in questions like "what object lies behind the fence." These mistakes occur consistently, especially when occlusion cues are subtle or when the depth difference is small.

★ **Analysis.** Such failures indicate that models treat depth and occlusion as weak, secondary signals. While the pixel data contains clear occlusion cues (e.g., edge boundaries, T-junctions), the intermediate features learned by the model do not retain them with sufficient resolution. Aggregation layers blur boundaries and downweight thin structures, effectively removing the very cues necessary to resolve ordering. This is why models succeed in gross depth distinctions (foreground vs. background) but collapse in fine-grained cases where partial occlusion must be inferred. In practice, they guess rather than infer, with systematic biases toward treating the visually larger or brighter object as foreground.

⊛ **Root cause.** The architecture lacks explicit mechanisms for multi-scale feature retention and has no dedicated supervision for depth or occlusion reasoning. Thin or partially occluded signals are easily lost when features are pooled across large receptive fields. Furthermore, training data often underrepresents subtle occlusion cases, especially those involving transparency or fine boundaries. As a result, the learned representations flatten layered scenes into 2D appearance maps, preventing robust construction of depth-aware relational reasoning.

### I.2.4 ATTRIBUTE CONFUSION AND SEMANTIC SWAP.

■ **Observation.** Models sometimes conflate perceptual attributes (such as color, texture, or material) with functional or semantic roles. A classic example is misjudging whether a crosswalk is safe: models may base the answer solely on the presence of painted stripes while ignoring the state of the traffic light or approaching vehicles. Similarly, a surface with a metallic texture may be misclassified as "steel" even if it is painted plastic. In tool-use contexts, the presence of handles or familiar shapes may lead the model to infer functionality that the object does not possess. These confusions are frequent in tasks requiring fine-grained visual grounding of attributes.

★ **Analysis.** These errors reveal that language priors strongly bias the model's predictions when visual input is ambiguous. The decoder favors plausible narratives based on co-occurrence in the training data: "crosswalks are safe when stripes are visible," "shiny textures correspond to metal," and so on. This shortcut allows models to produce semantically coherent answers even when they do not inspect the relevant visual evidence. The result is superficially correct reasoning in familiar contexts but systematic errors in atypical or adversarial scenarios. In practice, this manifests as overfitting to textual stereotypes rather than grounding in perceptual details.

⊛ **Root cause.** The underlying training objective, maximizing likelihood of fluent and plausible text, does not enforce visual grounding. As long as the generated answer is semantically reasonable, the loss is minimized, even if the reasoning is incorrect. Consequently, the model is incentivized to substitute common-sense priors for genuine perceptual inference. Without auxiliary losses or verification steps to tie textual claims to image regions, models will continue to swap perceptual and semantic attributes.

### I.2.5 OPEN-ENDED RATIONALIZATION WITHOUT VERIFICATION.

■ **Observation.** For explanatory or why-style questions, models often generate coherent but unsupported narratives. For example, when asked "why is the object leaning," a model might answer "because the surface is slippery" even though slipperiness cannot be visually inferred. In another case, a tilted shelf might be explained as "due to an earthquake," a claim entirely outside the evidence in the image. These errors highlight a tendency to fill gaps with imaginative but ungrounded rationales, producing fluent explanations that mislead users about the true evidence.

★ **Analysis.** The issue stems from a decoupling between text generation and visual grounding. The model generates high-quality language that reads convincingly, but there is no mechanism ensuring that these rationales are tied to actual visual features. Confidence calibration compounds the problem:

models often assign high certainty to fabricated explanations, giving the false impression of reliability. As a result, open-ended outputs may look authoritative while being factually baseless. This mode of failure is particularly concerning in safety-critical domains where plausible but false rationales could misguide decision-making.

⊛ **Root cause.** Training is dominated by language-modeling objectives that reward fluency and coherence but not factual grounding. Decoders are optimized to produce the most probable continuation of text given the prompt, without constraints linking the explanation back to specific image evidence. The lack of an explicit verification layer or grounding requirement means that models default to linguistic plausibility over perceptual truth. Without additional architectural or training constraints, open-ended rationalizations will continue to prioritize surface coherence at the expense of evidential accuracy.

### I.3 ERROR PATTERNS AND CROSS-MODEL SIGNALS

When comparing failures across different models, several consistent patterns appear, revealing insights into the underlying mechanisms of errors. One clear trend is that disagreements between models are systematic rather than random: some models consistently rely on geometric cues such as relative size, depth ordering, and occlusion relationships, while others fall back on linguistic priors or common-sense assumptions derived from training text. This leads to predictable divergences, for example when two models give different answers about which object is "in front" despite seeing the same scene. Errors also spike when multiple cues must be integrated simultaneously. Tasks that require reasoning about occlusion, relative scale, and perspective together are particularly challenging; for instance, understanding whether a larger object partially occluding a smaller one is in front or behind often triggers consistent mispredictions. Calibration is another noticeable issue: high-confidence predictions frequently correspond to incorrect answers, especially in open-ended explanations or complex spatial queries. These patterns indicate that errors arise from how models balance perceptual grounding and linguistic plausibility. Different architectures exhibit distinct tendencies, some over-rely on visual geometry, others overfit to language correlations, so failures are not just isolated mistakes but reflect deeper trade-offs in reasoning strategies. This also implies that any attempt to improve performance should account for both sources of bias simultaneously rather than treating errors as independent events.

### I.4 MITIGATION STRATEGIES

Given these error patterns, several targeted strategies can be pursued to reduce failures without requiring complete architectural redesign. For instance, geometry-aware supervision can be incorporated by adding auxiliary tasks like depth prediction, occlusion ordering, or overlap estimation. This encourages models to respect spatial constraints and improves consistency across predictions. Multi-scale feature retention is another approach: by using edge-aware pooling or attention mechanisms, models can maintain information from thin or partially occluded structures that would otherwise vanish in standard downsampling. Coordinate anchoring can help resolve confusion between viewer-centric and object-centric frames by explicitly representing reference points for relative positioning. Counterfactual augmentation can also be applied, where distractor objects are systematically swapped or perturbed, forcing the model to rely on the true referent cues rather than heuristics. Lightweight verification modules can be added on top of existing outputs, cross-checking claims against visual or geometric constraints to filter implausible predictions. Each of these strategies can be implemented incrementally, providing clear paths to reduce systematic failures while keeping the core model largely unchanged.

In addition to these architectural and training augmentations, supervised fine-tuning (SFT) offers a practical pathway to instill more reliable spatial reasoning. By curating datasets that explicitly highlight failure cases, such as near-threshold scale comparisons, occlusion ambiguities, or reference resolution traps, SFT can directly expose models to the edge scenarios where they otherwise default to heuristics. Carefully designed instruction-style fine-tuning further encourages step-by-step reasoning, prompting the model to verbalize intermediate spatial relations rather than skipping to a guess. This not only strengthens grounding but also improves interpretability, as errors become traceable to specific reasoning steps. Moreover, SFT can be paired with counterfactual or adversarial augmentations, ensuring that the model does not simply memorize but instead learns invariances to distractors and context shifts. Importantly, these interventions require no wholesale changes to the architecture: they

adapt existing models by reshaping their inductive biases through supervision. In this sense, SFT serves as a lightweight yet powerful complement to geometry-aware and verification-based strategies.

In this work, we additionally performed SFT using 40% of the training data and observed clear performance gains, particularly on multiple-choice questions. This suggests that targeted supervision helps models better internalize spatial cues and resolve distractor confusions when answer options are well-defined. However, improvements on open-ended responses were more limited. The model continues to produce fluent but weakly grounded explanations, highlighting that SFT alone cannot fully address rationalization without verification. These findings reinforce the need to combine SFT with geometry-aware supervision, counterfactual augmentations, and lightweight verifiers to achieve balanced progress across both structured and free-form tasks.

Our analysis shows that failures are not arbitrary but structured and interpretable, emerging from recurring patterns rather than random noise. This indicates that VLM errors follow systematic trajectories tied to the inductive biases of their architectures and training regimes. For example, geometric reasoning often collapses when occlusion cues are weak, and in such cases language priors take over, producing answers that sound plausible but lack visual grounding. Similarly, scale inference drifts when canonical object size priors dominate over pixel-level metric cues, leading to consistent misjudgments in near-threshold scenarios. These findings highlight that the gap is not simply one of capacity but of representation: current models lack explicit mechanisms to anchor visual features to grounded geometric relations.

By adopting the proposed verification protocols and targeted mitigation strategies, researchers can address these weaknesses directly. Geometry-aware supervision and counterfactual augmentation would help models resist shortcuts, while multi-scale feature retention ensures thin structures and partial occlusions are not discarded. SFT, when carefully applied, can further strengthen structured reasoning by aligning model predictions with task constraints, though it must be combined with grounding objectives to avoid overfitting to discrete-choice formats. Importantly, verification modules provide a lightweight way to enforce consistency between claimed relations and image evidence, counteracting the tendency to rationalize without checking. Taken together, these steps chart a practical pathway toward systems that reason more like humans: robust to clutter, interpretable in their failures, and capable of explaining not only what they predict but why.

---

**Base VLM Agent Prompt**

```
You are a Visual Scene Describer. Produce a raw, exhaustive text description of the given image.

Input image:
{image_input}

Your focus:
1. Object Identification:
   - Shapes (circle, square, triangle, polygon, etc.)
   - Natural elements (tree, sun, water, etc.)
   - Artificial objects (arrows, patterns, icons)

2. Attribute Noticing:
   - Size (small, medium, large, relative comparison)
   - Orientation (upright, rotated, tilted)
   - Shading or texture (solid, striped, shaded, transparent)
   - Color (if available)

3. Positioning:
   - Relative location (leftmost, rightmost, top, bottom, center)
   - Proximity to other objects

4. Level of Detail:
   - Do not summarize
   - Be verbose and explicit about every visible detail
```

---

**Object Segmentation Agent Prompt**

```
You are an Object Segmentation Specialist. Convert a verbose image description into a structured list of
distinct objects.

Input description:
{visual_description}

Your focus:
1. Object Isolation:
   - Treat each unique instance as a separate object
   - If multiple objects are similar, assign separate IDs

2. Labeling:
   - Assign IDs: Obj1, Obj2, Obj3...
   - Keep descriptions short but complete

3. Example Output:
   Obj1: Small shaded circle (leftmost)
   Obj2: Large unshaded square (center)
   Obj3: Triangle rotated 45° (rightmost)
```

---

**Attribute Extraction Agent Prompt**

```
You are an Attribute Extraction Specialist. For each object, extract its properties into structured JSON.

Your focus:
1. Attributes to Capture:
   - Shape
   - Size (relative if necessary)
   - Color or shading
   - Orientation or rotation
   - Position (leftmost, rightmost, center, top, bottom, etc.)

2. Output Format:
   JSON array, e.g.:
   [
      {{"ID": "Obj1", "Shape": "Circle", "Size": "Small", "Shading": "Yes", "Rotation": "0°", "Position":
"Leftmost"}},
      {{"ID": "Obj2", "Shape": "Square", "Size": "Large", "Shading": "No", "Rotation": "45°", "Position":
"Center"}}
   ]
```

Figure 10: Prompts for SPATIOXOLVER (Part 1).

**Spatial Relation Agent Prompt**

```
You are a Spatial Relation Specialist. Extract spatial relationships among objects.

Your focus:
1. Core Relations:
   - left_of, right_of, above, below
   - inside, surrounding
   - touching, overlapping
   - aligned, centered

2. Precision:
   - Always specify directionality (ObjA left_of ObjB is not the same as ObjB left_of ObjA)
   - Use consistent object IDs

3. Structural Patterns:
   - Detect repeated relations (e.g., "each circle is above a square")
   - Note spacing and alignment

4. Output Format:
   Triples in form: (ObjectA, Relation, ObjectB)
   Example:
   (Obj1, left_of, Obj2)
   (Obj3, inside, Obj4)
```

**Grouping Symmetry Agent Prompt**

```
You are a Grouping and Symmetry Specialist. Identify higher-order structures from objects and relations.

Relations:
{relation_list}

Your focus:
1. Grouping:
   - Clusters of objects (rows, columns, triangles, grids)
   - Repeated patterns

2. Symmetry:
   - Vertical, horizontal, rotational
   - Mirror or reflective symmetry
   - Translational repetition

3. Output Format:
   JSON with two fields: "groups" and "symmetries"
   Example:
   {{

     "groups": ["Obj1, Obj2, Obj3 form a horizontal row"],
     "symmetries": ["Vertical symmetry across Obj2"]
   }}
```

**Transformation Tracking Prompt**

```
You are a Transformation Tracker. Analyze changes across multiple frames of an image sequence.

Your focus:
1. Types of Transformations:
   - Movement (left, right, up, down)
   - Scaling (larger, smaller)
   - Rotation (degrees)
   - Shading or color change
   - Shape change (square → circle)
   - Appearance/disappearance

2. Output Format:
   Per-object transformation logs:
   Obj1: Frame1 → Frame2 (rotated 90° clockwise)
   Obj2: Frame2 → Frame3 (moved right)
```

Figure 11: Prompts for SPATIOXOLVER (Part 2).

---

**Representation Standardization Agent Prompt**

```
You are a Symbolic Abstraction Specialist. Consolidate all perception results into a standardized structured
format.

Inputs:
- Objects: {object_list}
- Attributes: {attributes_json}
- Relations: {relation_list}
- Groups & Symmetries: {grouping_symmetry}
- Transformations: {transformations}

Your focus:
1. Create a unified JSON representation containing:
    - Objects with attributes
    - Relations as triples
    - Groups and symmetries
    - Transformations across frames

2. Ensure consistency in IDs and formatting

3. Example Output:
{{
  "objects": [...],
  "relations": [...],
  "groups": [...],
  "symmetries": [...],
  "transformations": [...]}}
```

---

**MCQ Spatial Option Prompt**

```
You are a Spatial Reasoning Expert. Analyze the image data and select the correct option for the multiple-
choice question.

Image Analysis Data:
- Objects and Attributes: {attributes_json}
- Spatial Relations: {relations_list}
- Groups & Symmetries: {grouping_symmetry_json}
- Transformations Across Frames: {transformations_json}

MCQ:
Question: {question_text}
Options: {options_text}

Instructions:
- Consider shapes, positions, shading, rotations, symmetries, and transformations.
- Identify the pattern or rule in the sequence of images.
- Select only the correct option Only (1, 2, 3, 4 ).
- Do NOT provide explanations, reasoning, or extra text. Only the option letter.

Output format: "<option_number>"
```

---

**MCQ Spatial Option Prompt**

```
You are a Spatial Reasoning Expert. Analyze the image data and give correct answer for the  question.

Image Analysis Data:
- Objects and Attributes: {attributes_json}
- Spatial Relations: {relations_list}
- Groups & Symmetries: {grouping_symmetry_json}
- Transformations Across Frames: {transformations_json}

MCQ:
Question: {question_text}

Instructions:
- Consider shapes, positions, shading, rotations, symmetries, and transformations.
- Identify the pattern or rule in the sequence of images.
```

Figure 12: Prompts for SPATIOXOLVER (Part 3).

## J REPRESENTATIVE BENCHMARK SAMPLES ACROSS ALL SUB-CATEGORIES

We present representative benchmark samples for all 30 sub-categories in SPATIALAB, illustrating the diverse spatial reasoning challenges covered. These samples highlight the variety of visual phenomena, including 3D geometry, occlusion, orientation, relational positioning, size and scale, and navigation tasks. Together, they provide a concrete view of the types of reasoning each model must perform to succeed across the benchmark.

**3D Geometry: Gravity Effects**

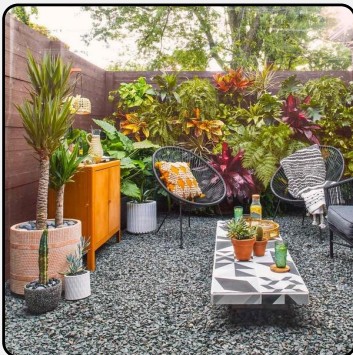

**Question: From which direction would the wind cause the liquid in the glass which is in the front to spill onto the ground first?**

**Options**
1. **From the left side**
2. **From the right side**
3. **From the front**
4. **From the back**

**Gold MCQ Answer: 1**

**Gold Open ended Answer: From the left side**

**MCQ Answers:**
- **GPT-5-mini: 4**
- **Gemini-2.5-Flash: 4**
- **InternVL3.5-72B: 1**
- **Kimi-VL-A3B-Thinking-2506: 1**

**Open-ended Answers:**
- **GPT-5-mini:** A gust blowing from the back of the patio (the top of the image, toward the viewer) would push the liquid forward over the near edge and make it spill first. Winds from the sides would tend to slosh it sideways rather than directly over the front rim.
- **Gemini-2.5-Flash:** The glass in the front is the one on the right side of the coffee table. Given its position, a wind blowing from the left (from the direction of the wooden cabinet and tall plant) would likely cause the liquid to spill first. This is because the glass is positioned with its opening facing somewhat towards the left, making it more susceptible to a gust from that direction.
- **InternVL3.5-72B:** The wind would need to come from the upper-right direction to cause the liquid in the glass on the front right of the table to spill first. This is due to the positioning of the glass towards the front edge of the table, closest to the right side of the image.
- **Kimi-VL-A3B-Thinking-2506:** Okay, let's tackle this question. The user is asking about the direction the wind would have to come from to spill the liquid in the front glass. Looking at the position of the glass: it's on the left front edge of the coffee table. The table seems to have a flat surface. If the wind blows from the left side (from the viewer's left, which would be the front of the image if the viewer.

Figure 13: Some examples from the benchmark (3D Geometry: Gravity Effects) and evaluation results.

## 3D Geometry: Shape Projection

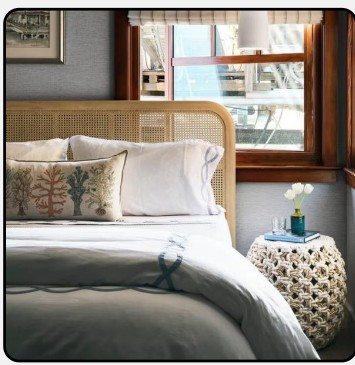

**Question: If the hanging light casts a shadow on the wall, which outline might appear?**

**Options**
1. **A rectangle narrowing at the bottom From the right side**
2. **A vertical cylinder**
3. **A circle with fuzzy edges**
4. **A triangle pointing downward**

**Gold MCQ Answer: 2**

**Gold Open ended Answer: A vertical cylinder**

**MCQ Answers:**
- **GPT-5-mini: 4**
- **Gemini-2.5-Flash: 1**
- **InternVL3.5-72B: 1**
- **Kimi-VL-A3B-Thinking-2506: 1**

**Open-ended Answers:**
- **GPT-5-mini:** You would see the pendant's dome-shaped profile: a rounded semicircle or oval shadow with a thin vertical line or narrow neck above where the cord hangs. The shadow would fall on the wall to the right of the window, roughly over the nightstand.
- **Gemini-2.5-Flash:** The hanging light is a simple, cylindrical pendant. If it casts a shadow on the wall, the outline would likely be a circular or elliptical shape, depending on the angle of the light source relative to the wall. Given its position near the window, the shadow might also be influenced by ambient light or other objects in the room.
- **InternVL3.5-72B:** The hanging light fixture is likely to cast a shadow resembling its own shape onto the wall. Given its cylindrical form, the shadow would appear as a soft-edged, elongated circle or oval, depending on the angle of the light source relative to the wall.
- **Kimi-VL-A3B-Thinking-2506:** Okay, let's tackle this question. The user is asking which outline a hanging light would cast on the wall if it casts a shadow. The wall near the window is light-colored, so the shadow would be visible. Since the shade is cylindrical, the shadow might be a cylinder. But maybe the outline is of the lampshade's perimeter. Alternatively, if the light is hanging from a chain, the shadow could be more elongated. But based on typical light fixtures, a cylindrical shade would cast a circular or oval shadow. The answer should be the outline of the hanging light's shade.

Figure 14: Some examples from the benchmark (3D Geometry: Shape Projection) and evaluation results.

## 3D Geometry: Spatial Containment

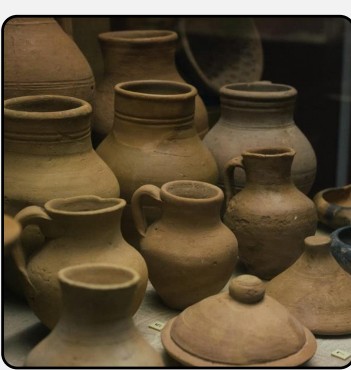

**Question: Look at the tall pot in the background with the wide mouth (top left). could the small jug from the center of the photo fit completely inside it?**

**Options**
1. **Yes, easily with room to spare**
2. **No, the small jug hand will stuck**
3. **No, the small jug is too tall.**
4. **Yes, but it would be a perfect fit with no room left**

**Gold MCQ Answer: 2**

**Gold Open ended Answer: No, the small jug hand will stuck**

**MCQ Answers:**
- **GPT-5-mini: 1**
- **Gemini-2.5-Flash: 1**
- **InternVL3.5-72B: 1**
- **Kimi-VL-A3B-Thinking-2506: 1**

**Open-ended Answers:**
- **GPT-5-mini:** No — the tall pot's mouth appears too narrow compared with the small jug's widest part and its handle. The jug wouldn't pass through the opening to sit completely inside.
- **Gemini-2.5-Flash:** No, the small jug from the center of the photo could not fit completely inside the tall pot in the background. While the tall pot has a wide mouth, the jug's overall height and width appear to be too large to be fully contained within the pot's interior. The jug's handle would also likely prevent it from fitting snugly.
- **InternVL3.5-72B:** The tall pot in the background with the wide mouth appears large enough to potentially accommodate the small jug from the center of the photo. Considering the proportions and the wide opening of the tall pot, it is likely that the small jug could fit completely inside it.
- **Kimi-VL-A3B-Thinking-2506:** Okay, let's see. The question is about whether the small jug in the center of the photo could fit completely inside the tall pot in the background with the wide mouth (top left). However, without explicit scale references, this conclusion relies on typical proportional relationships between similar pottery forms. Based on typical proportions, the small jug's height is likely shorter than the tall pot's, and its width seems narrow enough to fit.

Figure 15: Some examples from the benchmark (3D Geometry: Spatial Containment) and evaluation results.

**3D Geometry: Volume Comparison**

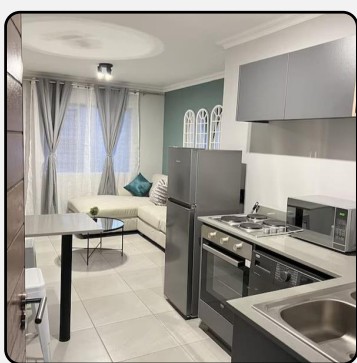

**Question: Compare the internal cooking volume of the microwave with the internal cooking volume of the larger oven below the stove.**

**Options**
1. **They both have the same volume A vertical cylinder**
2. **The microwave has slightly more volume**
3. **The oven below the stove has much more volume**
4. **The oven has more width, but the microwave has more height, so they are equal**

**Gold MCQ Answer: 3**

**Gold Open ended Answer: The oven below the stove has much more volume**

**MCQ Answers:**
- **GPT-5-mini: 3**
- **Gemini-2.5-Flash: 3**
- **InternVL3.5-72B: 3**
- **Kimi-VL-A3B-Thinking-2506: 3**

**Open-ended Answers:**
- **GPT-5-mini:** The microwave's cooking cavity is much smaller than the larger oven below the cooktop — it's only a small fraction of the oven's internal volume. The countertop microwave is sized for single plates or small dishes, whereas the built-in oven can hold full baking trays and multiple racks for larger meals.
- **Gemini-2.5-Flash:** The internal cooking volume of the microwave appears to be significantly smaller than that of the larger oven below the stove. The microwave is a compact countertop model, suitable for reheating or cooking smaller items..
- **InternVL3.5-72B:** The internal cooking volume of the microwave on the right is likely smaller than that of the larger oven located below the stove. This is because microwaves typically have more compact dimensions to fit on countertops, while ovens have a greater internal capacity to accommodate larger dishes and multiple items. The size and design visible in the image suggest the oven has a deeper and wider cooking area compared to the microwave.
- **Kimi-VL-A3B-Thinking-2506:** Okay, let's tackle this question. The user wants me to compare the internal cooking volumes of the microwave and the larger oven below the stove in the provided image. So, the internal cooking volume of the oven below the stove is larger than that of the microwave. Based on their relative sizes in the image, the internal cooking volume of the oven below the stove is substantially larger than that of the microwave

Figure 16: Some examples from the benchmark (3D Geometry: Volume Comparison) and evaluation results.

## 3D Geometry: Shape Projection

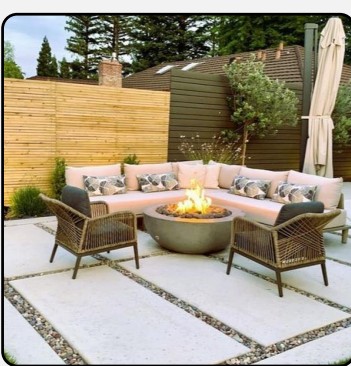

**Question: If you look at the seating arrangement from above, what shape does it form?**

**Options**
1. **A perfect square**
2. **A round shape**
3. **A rectangle**
4. **An irregular shape**

**Gold MCQ Answer: 4**

**Gold Open ended Answer: An irregular shape**

**MCQ Answers:**
- **GPT-5-mini:** 4
- **Gemini-2.5-Flash:** 4
- **InternVL3.5-72B:** 4
- **Kimi-VL-A3B-Thinking-2506:** 4

**Open-ended Answers:**
- **GPT-5-mini:** Viewed from above, the seating is arranged in a U-shape (a horseshoe) around the central fire pit. The L-shaped sectional plus the two facing chairs create that U configuration.
- **Gemini-2.5-Flash:** The seating arrangement, when viewed from above, primarily forms an L-shape with the large sectional sofa. This L-shape is complemented by two individual chairs placed opposite the open ends of the sectional. Together, the furniture creates a U-shaped configuration around the central fire pit, designed for communal gathering.
- **InternVL3.5-72B:** From above, the seating arrangement forms a U-shape around the central fire pit. The sectional sofa curves around the pit, leaving an open side opposite the tall fence, creating a semi-enclosed space.
- **Kimi-VL-A3B-Thinking-2506:** Okay, let's tackle this question. The user wants to know the shape formed by the seating arrangement when viewed from above. When viewed from above, the configuration clearly forms a right-angled L-shape, combining two parallel sections at a 90-degree angle. The symmetry of the layout and the placement of furniture around the fire pit reinforce this geometric form. The seating arrangement forms a right-angled L-shape when viewed from above.

Figure 17: Some examples from the benchmark (3D Geometry: Shape Projection) and evaluation results.

**Spatial Navigation: Accessibility Constraints**

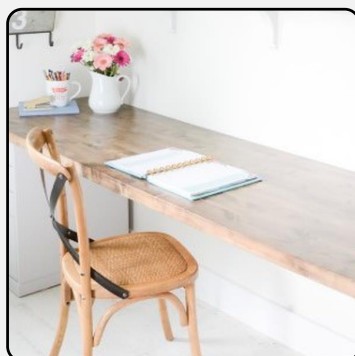

**Question:** The notebook at the leftmost corner of the table is inaccessible because of which object?

**Options**
1. Chair
2. Vase
3. Table
4. Mug with pens and pencil in it

**Gold MCQ Answer: 4**

**Gold Open ended Answer: Mug with pens and pencil in it**

**MCQ Answers:**
- **GPT-5-mini:** 1
- **Gemini-2.5-Flash:** 4
- **InternVL3.5-72B:** 4
- **Kimi-VL-A3B-Thinking-2506:** 1

**Open-ended Answers:**
- **GPT-5-mini:** The leftmost corner notebook is blocked by a wooden chair. The chair sits directly in front of the desk, preventing easy access to the notebook.
- **Gemini-2.5-Flash:** The notebook at the leftmost corner of the table is inaccessible because of the chair. The chair is positioned directly in front of the notebook, blocking direct access to it. To reach the notebook, one would first need to move the chair out of the way.
- **InternVL3.5-72B:** The notebook at the leftmost corner of the table is inaccessible because of the chair, which is positioned in front of it. The chair's backrest and seat block direct access to the notebook.
- **Kimi-VL-A3B-Thinking-2506:** Alright, let's tackle this spatial reasoning question. So, the question is asking why the notebook at the leftmost corner of the table is inaccessible. The chair is positioned near the leftmost corner of the table, and given its placement, it's reasonable to conclude that the chair is blocking access to the notebook. The chair's presence in that space would make it difficult to reach the notebook from the left side without moving the chair first. The chair is the object that's making the notebook at the leftmost corner inaccessible.

Figure 18: Some examples from the benchmark (Spatial Navigation: Accessibility Constraints) and evaluation results.

**Spatial Navigation: Obstacle Avoidance**

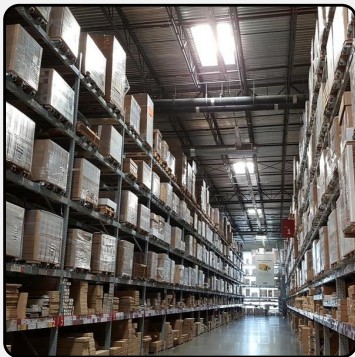

**Question: There is a clear direct path to the end of the hallway without colliding with any object.**

**Options**
1. **TRUE**
2. **FALSE**
3. **Maybe**
4. **Unable to determine**

**Gold MCQ Answer: 1**

**Gold Open ended Answer: TRUE**

**MCQ Answers:**
- **GPT-5-mini:** 1
- **Gemini-2.5-Flash:** 1
- **InternVL3.5-72B:** 1
- **Kimi-VL-A3B-Thinking-2506:** 1

**Open-ended Answers:**
- **GPT-5-mini:** Yes. The central aisle is unobstructed from the foreground to the far end, with shelving stacked along both sides and no items blocking the floor. You can walk straight down the hallway without colliding with any objects.
- **Gemini-2.5-Flash:** Yes, there is a clear direct path to the end of the hallway without colliding with any object. The image shows a wide, unobstructed aisle between two rows of tall shelving units filled with boxes. The floor is clear, indicating a safe passage for movement.
- **InternVL3.5-72B:** The image shows a large warehouse with high shelving units full of boxes on both sides of a central aisle. The aisle appears to be clear and unobstructed, leading directly to the end of the hallway. There are no objects or obstacles in the pathway, ensuring a direct and clear route to the end of the aisle.
- **Kimi-VL-A3B-Thinking-2506:** Okay, let's see. The user says there's a clear direct path to the end of the hallway without colliding with any object. A clear direct path exists because the shelving units are positioned symmetrically, leaving a central aisle free of objects. There are no visible hazards like hanging items, loose debris, or protruding elements in the visible portion of the hallway, allowing safe, uninterrupted movement to the end. The path is clear because the shelving units are symmetrically arranged, leaving the central walkway obstacle-free, with no visible hazards in the visible hallway.

Figure 19: Some examples from the benchmark (Spatial Navigation: Obstacle Avoidance) and evaluation results.

**Spatial Navigation: Pathway Existence**

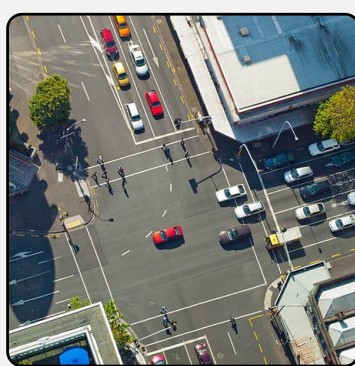

**Question: Which walkways are the safest to cross according to the picture?**

**Options**
1. **Top and bottom**
2. **Top and left**
3. **Left and right**
4. **Bottom and right**

**Gold MCQ Answer: 1**

**Gold Open ended Answer: Top and bottom**

**MCQ Answers:**
- **GPT-5-mini: 2**
- **Gemini-2.5-Flash: 3**
- **InternVL3.5-72B: 1**
- **Kimi-VL-A3B-Thinking-2506: 1**

**Open-ended Answers:**
- **GPT-5-mini:** The marked white crosswalks at each corner of the intersection (the painted lines across the road) are the safest to use. They are signalized and where pedestrians are waiting, unlike cutting across mid-block or diagonally through the intersection.
- **Gemini-2.5-Flash:** The walkways marked with white lines, commonly known as crosswalks, are the safest to cross. These designated areas indicate where pedestrians have the right-of-way or are expected to cross the street. In the image, several people are visible using these marked crosswalks, suggesting they are the intended and therefore safest routes for pedestrians.
- **InternVL3.5-72B:** The crosswalks aligned with the pedestrian signals at the corners are the safest places to cross, as they are equipped with traffic signals ensuring vehicles are stopped. The designated pedestrian areas with painted lines across the streets are clearly marked for safety. Additionally, sidewalks provide a safe space for walking along the roads.
- **Kimi-VL-A3B-Thinking-2506:** Okay, let's tackle this question. The user wants to know which walkways are the safest to cross based on the image provided. The safest walkways to cross, based on the image, are the marked crosswalks equipped with pedestrian signals. Specifically, the crosswalk at the bottom-left corner of the intersection is safest because it features a visible pedestrian signal indicating walk, reducing ambiguity with traffic flow.

Figure 20: Some examples from the benchmark (Spatial Navigation: Pathway Existence) and evaluation results.

## Spatial Navigation: Viewpoint Visibility

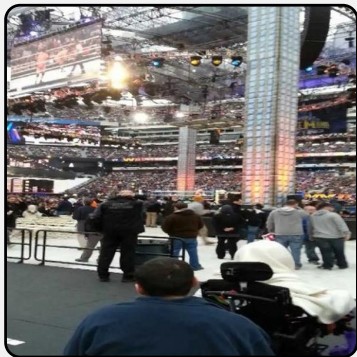

**Question:** The human standing the closest (blue jacket) has clear view of the stadium, how?

**Options**
1. Seeing directly
2. Watching on the big screen on the left side
3. There is no stadium
4. Unable to determine

**Gold MCQ Answer: 2**

**Gold Open ended Answer: Watching on the big screen on the left side**

**MCQ Answers:**
- **GPT-5-mini:** 1
- **Gemini-2.5-Flash:** 2
- **InternVL3.5-72B:** 2
- **Kimi-VL-A3B-Thinking-2506:** 1

**Open-ended Answers:**
- **GPT-5-mini:** Because he's standing on the raised walkway right behind the front-row barrier, his eye level is higher than the people immediately in front of him. That elevation (and the clear space by the railing) gives him an unobstructed line of sight over the crowd into the stadium.
- **Gemini-2.5-Flash:** The person in the blue jacket is seated in a wheelchair, which places them at a lower vantage point than the standing crowd. However, they have a clear view of the stadium because they are positioned at the very front of a section, likely an accessible seating area. This front-row placement ensures an unobstructed line of sight over the heads of any standing individuals further back.
- **InternVL3.5-72B:** The person in the blue jacket is standing near the front of the arena, which allows them a clear, unobstructed view of the stadium. Their position is close to the barrier separating the audience from the performance area, providing a direct line of sight towards the stage and the large screens overhead. This proximity minimizes visual obstructions from other spectators or structures.
- **Kimi-VL-A3B-Thinking-2506:** Okay, let's tackle this question. The user is asking how the person in the blue jacket (who is standing the closest) has a clear view of the stadium. The person's elevated seating, minimized obstructions from nearby spectators, and potential alignment with stadium features (e.g., screens or lighting) enable their clear view despite being relatively close to the ground.

Figure 21: Some examples from the benchmark (Spatial Navigation: Viewpoint Visibility) and evaluation results.

**Spatial Navigation: Spatial Sequence**

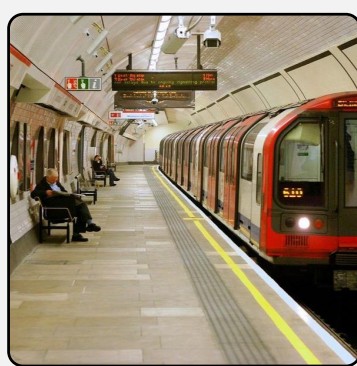

**Question: Which path is the easiest for the human sitting in the bench(reading paper) to follow to get on the train?**

**Options**
1. **Go straight to the first red colored door**
2. **To make a turn and go to the second floor**
3. **Go through the door at the end of the train**
4. **No viable option**

**Gold MCQ Answer: 1**

**Gold Open ended Answer: Go straight to the first red colored door**

**MCQ Answers:**
- **GPT-5-mini:** 1
- **Gemini-2.5-Flash:** 1
- **InternVL3.5-72B:** 1
- **Kimi-VL-A3B-Thinking-2506:** 1

**Open-ended Answers:**
- **GPT-5-mini:** Because he's standing on the raised walkway right behind the front-row barrier, his eye level is higher than the people immediately in front of him. That elevation (and the clear space by the railing) gives him an unobstructed line of sight over the crowd into the stadium.
- **Gemini-2.5-Flash:** The simplest way for the person to board the train is to rise from the bench, turn right, and walk a short distance along the platform. They would then cross the yellow safety line and step through one of the train doors. This route is straightforward and involves minimal effort.
- **InternVL3.5-72B:** The person sitting on the bench can follow a straightforward path to board the train by moving forward along the platform, parallel to the train tracks, while keeping to the right of the yellow safety line. This path will lead them directly to the nearest open train door. They should avoid crossing the yellow line until they are ready to enter the train to ensure safety.
- **Kimi-VL-A3B-Thinking-2506:** Okay, let's see. The question is about determining the easiest path for the person sitting on the bench to get on the train. First, I need to look at the image provided. The person is sitting on the bench which is alongside the platform walls. The train is on the opposite side. To get to the train, the person would..

Figure 22: Some examples from the benchmark (Spatial Navigation: Spatial Sequence) and evaluation results.

**Size and Scale: Scale Consistency**

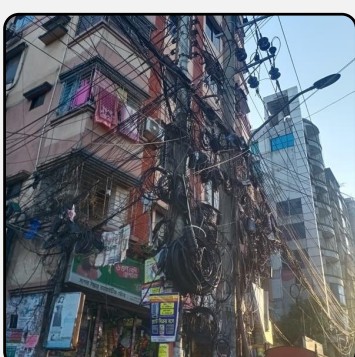

**Question: With the electric pole's capacity can all this wires be redesigned to carry properly in aligned structured design?**

**Options**
1. **The size of wires are too big**
2. **The wires need to be move into another pole**
3. **Wires can be set there following a layout**
4. **No, the bulk size of the wires are too much here for one pole**

**Gold MCQ Answer: 4**

**Gold Open ended Answer: No, the bulk size of the wires are too much here for one pole**

**MCQ Answers:**
- **GPT-5-mini: 3**
- **Gemini-2.5-Flash: 4**
- **InternVL3.5-72B: 4**
- **Kimi-VL-A3B-Thinking-2506: 3**

**Open-ended Answers:**
- **GPT-5-mini:** Not safely in its current state — the pole is clearly overloaded and the wires are tangled, so simply re-bundling them on the same pole would exceed mechanical, clearance and safety limits. A proper redesign is possible but requires removing redundant lines, separating circuits (high-voltage vs. telecom) onto organized crossarms, trays or underground ducts, and likely installing stronger or additional poles and anchors.
  **Gemini-2.5-Flash:** While it's technically possible to redesign the wiring for better alignment, the sheer volume and entanglement of wires in the image suggest that the current electric pole's capacity is severely overtaxed. Simply reorganizing the existing wires on the same pole would not address the underlying capacity issues or the safety hazards presented by such a dense and disorganized network.
- **InternVL3.5-72B:** Redesigning the extensive web of wires around the electric pole into a properly aligned and structured design is feasible but would likely require significant effort and resources. Proper planning and coordination with multiple service providers would be essential. Implementing such changes would enhance safety and aesthetics, but it involves logistical challenges and coordination to minimize service disruptions.
- **Kimi-VL-A3B-Thinking-2506:** Okay, let's tackle this question. The user is asking if all the wires on the electric pole can be redesigned to be properly aligned in a structured way. A redesigned system is feasible but dependent on evaluating regulatory, technical, and safety constraints. Prioritizing organized routing for essential cables while removing redundant ones could create.

Figure 23: Some examples from the benchmark (Size and Scale: Scale Consistency) and evaluation results.

---

**Size and Scale: Shadow-Size Projection**

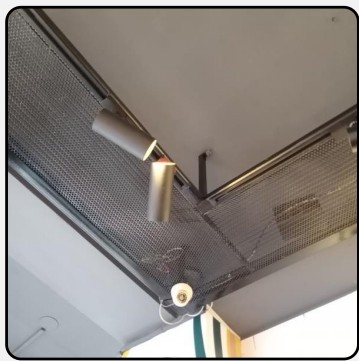

**Question: If one light is rotated so that they are perpendicular to each other, how will the shadow look over the wall?**

**Options**
1. It will cover almost all of the wall
2. The shadow will only take the size of the light frame as it's actual size
3. There won't be any shadow
4. Shadow will smaller than the actual light body

**Gold MCQ Answer: 1**

**Gold Open ended Answer: It will cover almost all of the wall**

**MCQ Answers:**
- **GPT-5-mini: 4**
- **Gemini-2.5-Flash: 1**
- **InternVL3.5-72B: 1**
- **Kimi-VL-A3B-Thinking-2506: 1**

**Open-ended Answers:**
- **GPT-5-mini:** If one lamp is rotated 90° so the two beams are perpendicular, you'll get two elongated oval patches of light on the wall at right angles to each other. Where the ovals overlap it will be brightest; outside them the wall will be darker, producing an L-shaped darker region and two perpendicular shadow lobes cast by the fixture.
- **Gemini-2.5-Flash:** If one light is rotated to be perpendicular to the other, the shadow cast on the wall would change significantly. The current shadows are likely elongated and somewhat parallel due to the lights' current orientation. With one light rotated, the shadows would intersect at a more acute angle, creating a more complex and possibly darker overlapping shadow where the light beams cross.
- **InternVL3.5-72B:** If one light is rotated so that the two lights are perpendicular to each other, the shadows on the wall will intersect at a right angle, creating intersecting patterns. The light beams will cast shadows in perpendicular directions, resulting in a cross-like effect where the shadows overlap. The intensity and shape of the shadows will depend on the orientation and angle of the lights relative to the wall.
- **Kimi-VL-A3B-Thinking-2506:** Okay, let's try to figure out this question. The user is asking about rotating a light so that they're parpassword to each other and how the shadow on the wall would look. If they're further away, the shadow would be smaller.

Figure 24: Some examples from the benchmark (Size and Scale: Shadow-Size Projection) and evaluation results.

**Size and Scale: Perspective Distortion**

**Question: If similar shaped pipes are stacked together separately will it take up less space?**

**Options**
1. **Yes**
2. **No**
3. **Take more**
4. **Similar**

**Gold MCQ Answer: 1**

**Gold Open ended Answer: Yes**

**MCQ Answers:**
- **GPT-5-mini:** 1
- **Gemini-2.5-Flash:** 1
- **InternVL3.5-72B:** 1
- **Kimi-VL-A3B-Thinking-2506:** 1

**Open-ended Answers:**
- **GPT-5-mini:** Yes - when like-shaped pipes are grouped you can pack them in their most efficient pattern ,which reduces wasted voids and the total footprint. The exact savings depend on orientation and dimensions; in some cases clever mixing or nesting of different shapes can beat separate stacks, but as a general rule same-shape grouping is more space-efficient.
- **Gemini-2.5-Flash:** Yes, if similarly shaped pipes are stacked together separately, it would likely take up less space. This is because organizing them by shape allows for more efficient packing and reduces wasted space between different forms. For example, square pipes can be stacked flush against each other, and circular pipes can be nested or arranged in a hexagonal pattern for optimal density.
- **InternVL3.5-72B:** Yes, stacking similar shaped pipes together separately will generally take up less space. Organizing them by shape allows for more efficient packing, reducing gaps and empty spaces between different sizes and types. This method optimizes the use of available storage area and improves accessibility.
- **Kimi-VL-A3B-Thinking-2506:** Okay, let's see. The question is asking if stacking similar shaped pipes separately would take up less space. Stacking similar-shaped pipes separately can potentially optimize space usage. When pipes of the same diameter and shape (e.g., all round or rectangular) are grouped together, they can be arranged more tightly, minimizing gaps between

Figure 25: Some examples from the benchmark (Size and Scale: Perspective Distortion) and evaluation results.

### Size and Scale: Relative Size Comparison

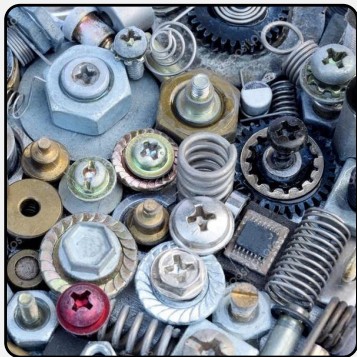

**Question: The ash-colored, thicker coiled spring will be surpassed by which coiled springs if all are kept in the standing position?**

**Options**
1. **Right bottom corner**
2. **Right top corner**
3. **Left top corner**
4. **Bottom mid spring**

**Gold MCQ Answer: 1**

**Gold Open ended Answer: Right bottom corner**

**MCQ Answers:**
- **GPT-5-mini:** 1
- **Gemini-2.5-Flash:** 4
- **InternVL3.5-72B:** 1
- **Kimi-VL-A3B-Thinking-2506:** 2

**Open-ended Answers:**
- **GPT-5-mini:** The ash-colored thicker spring in the center will be shorter than the thin, taller silver spring just behind it (top-center) and the long, dark-gray spring at the right edge when all are stood upright. Those two coils extend above the ash-colored spring.
- **Gemini-2.5-Flash:.** The ash-colored, thicker coiled spring, located near the center-right of the image, would be surpassed by the longer, thinner coiled spring positioned in the bottom right corner. Additionally, the two smaller, conical springs in the upper left quadrant would also be taller if fully extended and standing. These springs vary in height and coil density, indicating different potential applications.
- **InternVL3.5-72B:** The ash-colored, thicker coiled spring in the image will be surpassed in height by the two taller springs on its right. One of these taller springs is also ash-colored, while the other has a slightly darker hue. Both of these springs have more coils and are visibly longer when standing upright.
- **Kimi-VL-A3B-Thinking-2506:** Okay, let's try to figure out this question. The image shows a bunch of different coiled springs. The question is asking which coiled springs would be surpassed by the ash-colored, thicker one if all are kept in the standing position The thinner, tightly coiled springs (smaller in diameter or fewer coils) would be surpassed in height by the ash-colored, thicker spring when all are kept upright.

Figure 26: Some examples from the benchmark (Size and Scale: Relative Size Comparison) and evaluation results.

**Size and Scale: Relative Size Comparison**

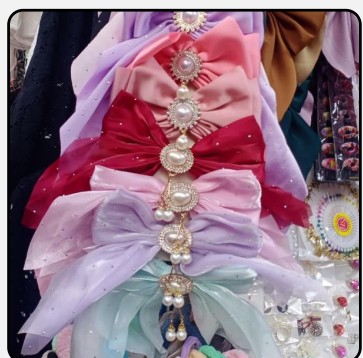

**Question: If the red bow is taken out of it's place and kept far away in the behind how will it look and what will be it's capacity comparing with it's other similar clips at front?**

**Options**
1. Large, same
2. Same, less
3. Small, same
4. Small, large

**Gold MCQ Answer: 3**

**Gold Open ended Answer: Small, same**

**MCQ Answers:**
- **GPT-5-mini:** 3
- **Gemini-2.5-Flash:** 2
- **InternVL3.5-72B:** 2
- **Kimi-VL-A3B-Thinking-2506:** 3

**Open-ended Answers:**
- **GPT-5-mini:** If you move the red bow to the back it will appear smaller and more obscured by the bows in front because of perspective and overlap, and its red color will look less vivid from that angle. Physically nothing about the bow changes — its size, shape and clasp strength remain the same as the other similar clips in front, so its holding capacity is unchanged.
- **Gemini-2.5-Flash:** If the red bow were taken out and placed far behind, it would appear smaller and less distinct due to perspective. Its capacity, referring to its visual prominence or ability to hold hair, would remain the same as the other similar clips. However, its perceived capacity might diminish as it would no longer be in the foreground, drawing less attention.
- **InternVL3.5-72B:** If the red bow is moved to the back, it will be partially obscured by the other bows in the foreground, making it less prominent. Its size and capacity to hold hair will remain the same as the others, but visually, it won't stand out as much as it does currently when placed in the front.
- **Kimi-VL-A3B-Thinking-2506:** Okay, let's tackle this question. The user is asking about removing the red bow and placing it behind, then comparing its appearance and capacity to the other similar clips in front. Removing the red bow would reduce its central visual impact, making the remaining front-facing bows appear denser but less balanced. Its capacity (physical or visual space) in the front area would decrease, potentially limiting the number of similar bows that can be displayed

Figure 27: Some examples from the benchmark (Size and Scale: Relative Size Comparison) and evaluation results.

## Depth & Occlusion : Complete Occlusion Inference

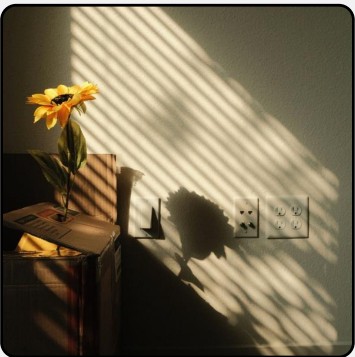

**Question: What is creating the striped shadow on the wall?**

**Options**
1. **The sunflower**
2. **Blinds**
3. **Window**
4. **Not applicable**

**Gold MCQ Answer: 2**

**Gold Open ended Answer: Blinds**

**MCQ Answers:**
- **GPT-5-mini:** **2**
- **Gemini-2.5-Flash:** **2**
- **InternVL3.5-72B:** **2**
- **Kimi-VL-A3B-Thinking-2506:** **2**

**Open-ended Answers:**
- **GPT-5-mini:** Sunlight streaming through window blinds (slatted shades) is casting the diagonal striped shadow on the wall. The slats break the light into parallel bands, producing the striped pattern.
- **Gemini-2.5-Flash:** The striped shadow on the wall is created by sunlight passing through a window with horizontal blinds. The blinds block portions of the light, casting distinct parallel lines onto the wall. This effect is common when direct sunlight enters a room through slatted window coverings.
- **InternVL3.5-72B:** The striped shadow on the wall is created by blinds or window slats. The sunlight is passing through the gaps between the horzontal blinds, casting parallel lines of light and shadow. This pattern is often seen when light is filtered through window coverings with narrow openings. The alignment and spacing of the lines suggest that the blinds are partially open, allowing the sun to project the shadow pattern onto the wall.
- **Kimi-VL-A3B-Thinking-2506:** The striped shadow on the wall is created by sunlight passing through a set of window blinds or shutters with narrow, evenly spaced slats. The parallel lines of the shadow result from the angular interaction between the light source (likely the sun) and the rigid, horizontal slats of the blinds, casting consistent geometric patterns onto the wall. The uniformity and sharpness of the shadows suggest the light is direct and possibly filtered through the gaps between the slats.

Figure 28: Some examples from the benchmark (Depth and Occlusion: Complete Occlusion Inference) and evaluation results.

## Depth & Occlusion : Layering Order

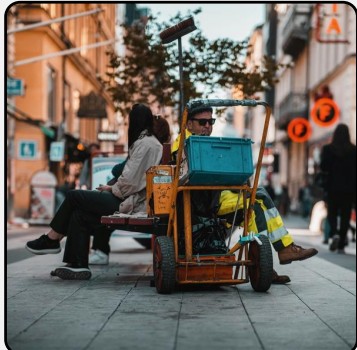

**Question: What is the frontmost object in the picture?**

**Options**
1. Trolley
2. Box
3. Man
4. Woman

**Gold MCQ Answer: 1**

**Gold Open ended Answer: Trolley**

**MCQ Answers:**
- **GPT-5-mini:** 1
- **Gemini-2.5-Flash:** 1
- **InternVL3.5-72B:** 1
- **Kimi-VL-A3B-Thinking-2506:** 1

**Open-ended Answers:**
- **GPT-5-mini:** The frontmost object is a small orange street-cleaning cart with a blue plastic bin mounted on it. A broom is attached to the cart and it sits on the pavement in the foreground.
- **Gemini-2.5-Flash:** The frontmost object in the picture is the yellow cart with a blue crate on top, which a man in a yellow and black suit is sitting on. Its wheels are closest to the viewer, indicating its position in the foreground. The people sitting on the bench behind the cart are further back in the scene.
- **InternVL3.5-72B:** The frontmost object in the image is a cleaning cart with wheels. It has a blue container attached to it and a broom standing upright. There's also a noticeable yellow brush or shovel handle next to the container. The cart appears to be positioned on a sidewalk in an urban environment.
- **Kimi-VL-A3B-Thinking-2506:** The perspective is from the ground level, looking along the sidewalk. The cart is positioned in the foreground, closest to the viewer. The frontmost object is a high-visibility work cart.

Figure 29: Some examples from the benchmark (Depth and Occlusion: Layering Order) and evaluation results.

**Depth & Occlusion : Partial Occlusion**

**Question: What is the color of the ball behind the white colored square shaped device?**

**Options**
1. **Cyan**
2. **Yellow**
3. **Red**
4. **Not applicable**

**Gold MCQ Answer: 1**

**Gold Open ended Answer: Cyan**

**MCQ Answers:**
- **GPT-5-mini:** 1
- **Gemini-2.5-Flash:** 2
- **InternVL3.5-72B:** 1
- **Kimi-VL-A3B-Thinking-2506:** 1

**Open-ended Answers:**
- **GPT-5-mini:** The ball directly behind the white, square-shaped device is blue (a cyan/teal colored ball). It sits slightly to the left and behind the device.
- **Gemini-2.5-Flash:** The ball directly behind the white-colored square-shaped device (GameBoy) is teal. There is also a yellow-green ball further to the right of the GameBoy.
- **InternVL3.5-72B:** The ball behind the white, square shaped device is blue.
- **Kimi-VL-A3B-Thinking-2506:** The color of the ball behind the white square-shaped device is blue.

Figure 30: Some examples from the benchmark (Depth and Occlusion: Partial Occlusion) and evaluation results.

**Depth & Occlusion : Reflective Surfaces**

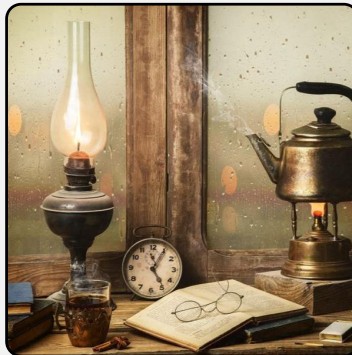

**Question: Which object is the shiniest among all the objects in the image?**

**Options**
1. **Kettle**
2. **Tea glass**
3. **Table clock**
4. **Lamp chimney**

**Gold MCQ Answer: 4**

**Gold Open ended Answer: Lamp chimney**

**MCQ Answers:**
- **GPT-5-mini:** 1
- **Gemini-2.5-Flash:** 4
- **InternVL3.5-72B:** 1
- **Kimi-VL-A3B-Thinking-2506:** 1

**Open-ended Answers:**
- **GPT-5-mini:** The frontmost object is a small orange street-cleaning cart with a blue plastic bin mounted on it. A broom is attached to the cart and it sits on the pavement in the foreground.
- **Gemini-2.5-Flash:** The frontmost object in the picture is the yellow cart with a blue crate on top, which a man in a yellow and black suit is sitting on. Its wheels are closest to the viewer, indicating its position in the foreground. The people sitting on the bench behind the cart are further back in the scene.
- **InternVL3.5-72B:** The frontmost object in the image is a cleaning cart with wheels. It has a blue container attached to it and a broom standing upright. There's also a noticeable yellow brush or shovel handle next to the container. The cart appears to be positioned on a sidewalk in an urban environment.
- **Kimi-VL-A3B-Thinking-2506:** The perspective is from the ground level, looking along the sidewalk. The cart is positioned in the foreground, closest to the viewer. The frontmost object is a high-visibility work cart.

Figure 31: Some examples from the benchmark (Depth and Occlusion: Reflective Surfaces) and evaluation results.

**Depth & Occlusion : Transparency Effects**

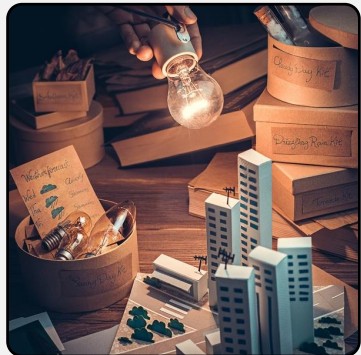

**Question: How many objects has filaments inside them in the image?**

**Options**
1. 7
2. 6
3. 5
4. 4

**Gold MCQ Answer: 4**

**Gold Open ended Answer: 4**

**MCQ Answers:**
- **GPT-5-mini: 3**
- **Gemini-2.5-Flash: 3**
- **InternVL3.5-72B: 4**
- **Kimi-VL-A3B-Thinking-2506: 4**

**Open-ended Answers:**
- **GPT-5-mini:** Three. The glowing bulb being held clearly shows a filament, and two more filament bulbs are visible inside the round box on the left.
- **Gemini-2.5-Flash:** There are three objects with filaments inside them. One is the illuminated light bulb being held, and the other two are unlit light bulbs inside the "Sunny Day Kit" box. All three are incandescent light bulbs, which contain a filament that glows when heated.
- **InternVL3.5-72B:** In the image, there are four objects with visible filaments. These include the lit light bulb being held, and three unlit bulbs inside the container labeled "Sunny Day Kit."
- **Kimi-VL-A3B-Thinking-2506:** There are 3 objects with filaments in the image - one held in hand and two in the "Sunny Day Kit" container.

Figure 32: Some examples from the benchmark (Depth and Occlusion: Transparency Effects) and evaluation results.

## Orientation : Cardinal Direction

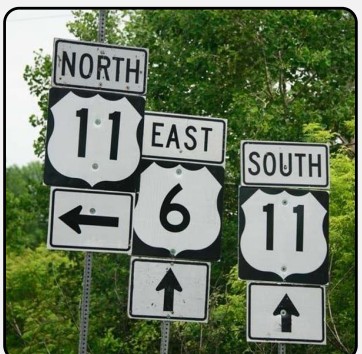

**Question: If you go straight following route 6, which direction are you traveling?**

**Options**
1. North
2. South
3. East
4. West

**Gold MCQ Answer: 3**

**Gold Open ended Answer: East**

**MCQ Answers:**
- **GPT-5-mini: 3**
- **Gemini-2.5-Flash: 3**
- **InternVL3.5-72B: 3**
- **Kimi-VL-A3B-Thinking-2506: 1**

**Open-ended Answers:**
- **GPT-5-mini:** You would be traveling east. The Route 6 shield is labeled "EAST" and the straight-ahead arrow beneath it shows that going straight follows eastbound Route 6.
- **Gemini-2.5-Flash:** If you go straight following route 6, you will be traveling East. The sign for Route 6 is directly below the "EAST" sign, and there is an arrow pointing straight up beneath the Route 6 shield, indicating that going straight leads to Route 6 East
- **InternVL3.5-72B:** If you go straight following Route 6, you are traveling east. This is indicated by the sign with the number "6" and the arrow pointing forward.
- **Kimi-VL-A3B-Thinking-2506:** Standard highway signs usually have the path of the route. So if it's route 11 East, the arrow would indicate the direction. But in the image, the left arrow is under the NORTH 11 EAST sign. Maybe that's indicating that route 11 East goes left? But the sign says East. Alternatively, maybe the arrows are indicating the direction of the route numbers. Like, route 11 North is going straight, route 11 East is turning left, etc. But then the route 6 sign has an upward arrow. So if you take route 6 straight, which direction is that? The arrow on the route 6 shield is pointing straight up. But the Question: is about direction—like North, South, East, West. Wait, the signs here are for routes, not cardinal directions. So maybe the answer is that going straight on route 6 would be in the direction indicated by the upward arrow. But the upward arrow might be part of the route 6 sign.

Figure 33: Some examples from the benchmark (Orientation: Cardinal Direction) and evaluation results.

### Orientation : Facing Direction

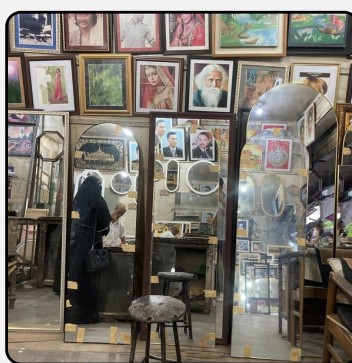

**Question: Suppose, Rabindranath Tagore's portrait is hung on the left wall and it faces directly into the room, what is the direction of the portrait with reference to the wall?**

**Options**
1. **Up**
2. **Down**
3. **Away**
4. **Toward**

**Gold MCQ Answer: 3**

**Gold Open ended Answer: Away**

**MCQ Answers:**
- **GPT-5-mini: 3**
- **Gemini-2.5-Flash: 3**
- **InternVL3.5-72B: 4**
- **Kimi-VL-A3B-Thinking-2506: 4**

**Open-ended Answers:**
- **GPT-5-mini:** If the portrait is on the left wall and faces directly into the room, it is oriented perpendicular to that wall and pointing inward. In other words, it faces away from the wall toward the center of the room.
- **Gemini-2.5-Flash:** The portrait of Rabindranath Tagore is located on the wall to the left, as seen in the image. If it faces directly into the room, its direction with reference to the wall would be perpendicular to the wall. This means the portrait is facing straight out from the wall into the space of the room.
- **InternVL3.5-72B:** If Rabindranath Tagore's portrait is hung on the left wall and faces directly into the room, it would be positioned perpendicular to the wall. This means the portrait is oriented so that viewers inside the room can face it directly when looking toward the wall on which it is hung.
- **Kimi-VL-A3B-Thinking-2506:** The portrait is hung on the left wall and faces directly into the room, meaning it is oriented inward toward the interior space.

Figure 34: Some examples from the benchmark (Orientation: Facing Direction) and evaluation results.

**Orientation : Object Rotation**

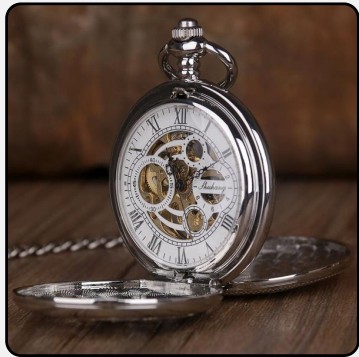

**Question: What is the approximate angle of the cases of the pocket watch?**

**Options**
1. 90
2. 45
3. 180
4. 120

**Gold MCQ Answer: 3**

**Gold Open ended Answer: 180**

**MCQ Answers:**
- **GPT-5-mini: 3**
- **Gemini-2.5-Flash: 3**
- **InternVL3.5-72B: 1**
- **Kimi-VL-A3B-Thinking-2506: 1**

**Open-ended Answers:**
- **GPT-5-mini:** About 180°. The two covers are opened in opposite directions and lie nearly flat, forming an almost straight line.
- **Gemini-2.5-Flash:** The cases of the pocket watch are open, revealing the intricate mechanism inside. The front cover is open to approximately a 120-degree angle relative to the watch face. The back case is also open, but its angle is less clear, appearing to be around 90 degrees or slightly more.
- **InternVL3.5-72B:** The pocket watch in the image appears to be partially open, with the angle between its two cases estimated to be around 60 degrees. The case opens hinge-wise, and the visible gap suggests that it is made to display the internal mechanical components of the watch. This angle provides a good view of the watch face and some of its intricate gears..
- **Kimi-VL-A3B-Thinking-2506:** When the watch is open, the cases form an angle. If the covers are both open to 180 degrees relative to each other, that would be a straight line. But in this image, the back cover isn't fully open. It looks like it's at around 90 degrees from the front cover. So the angle between the cases would be approximately 90 degrees.

Figure 35: Some examples from the benchmark (Orientation: Object Rotation) and evaluation results.

**Orientation : Stacking Orientation**

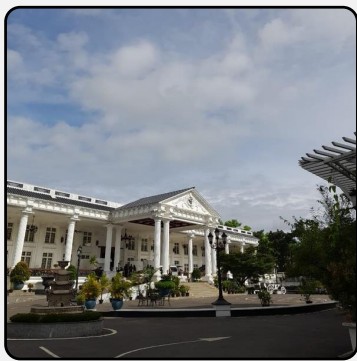

**Question: What is the relationship between the road and the other objects in the image?**

**Options**
1. **Behind the building and plants**
2. **In front of the plants and leads towards the building**
3. **The road is beside the building, with no clear direction**
4. **The road is stacked above the building and plants**

**Gold MCQ Answer: 2**

**Gold Open ended Answer: In front of the plants and leads towards the building**

**MCQ Answers:**
- **GPT-5-mini:** 2
- **Gemini-2.5-Flash:** 2
- **InternVL3.5-72B:** 2
- **Kimi-VL-A3B-Thinking-2506:** 2

**Open-ended Answers:**
- **GPT-5-mini:** The road runs in front of the large white building, curving around a central planted island and seating area. It separates and encircles the entrance and decorative planters, forming a driveway/roundabout in the courtyard.
- **Gemini-2.5-Flash:** The road in the image serves as an access path, leading towards the grand white building. It curves around a central island feature, which contains a decorative fountain and some potted plants. This suggests the road is a driveway or entrance loop designed to facilitate vehicle access to the building.
- **InternVL3.5-72B:** The road in the image forms a semi-circular driveway in front of a large white building with columns. It encircles a landscaped garden area with plants, a stone sculpture, and a fountain at the center, leading up to the building's entrance. The road is bordered by a curb that separates it from the garden and pedestrian space. Lamp posts are situated alongside the road, enhancing the building's stately appearance.
- **Kimi-VL-A3B-Thinking-2506:** The road curves around the front of the grand white building, acting as a transitional pathway that guides movement and attention toward the entrance while dividing space between the main structure and other elements like the pergola. It establishes scale and perspective while integrating with the surrounding landscape.

Figure 36: Some examples from the benchmark (Orientation: Stacking Orientation) and evaluation results.

**Orientation : Tool Handedness**

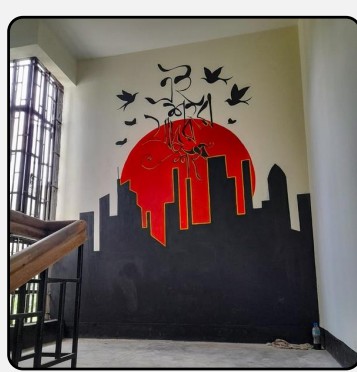

**Question: Considering the likely process of creating the mural, which tool's handedness would be most evident in the smooth, curved lines of the bengali script?**

**Options**
1. **Paint roller**
2. **Paint brush**
3. **Tape measure**
4. **Stencil**

**Gold MCQ Answer: 2**

**Gold Open ended Answer: Paint brush**

**MCQ Answers:**
- **GPT-5-mini: 2**
- **Gemini-2.5-Flash: 2**
- **InternVL3.5-72B: 2**
- **Kimi-VL-A3B-Thinking-2506: 2**

**Open-ended Answers:**
- **GPT-5-mini:** Those flowing, tapered curves are most characteristic of a paintbrush (or broad paint marker) rather than a spray can. The handedness of the brush user — most likely right-handed — would be most evident, since right-handed strokes tend to pull and taper in predictable left-to-right sweeps visible in the Bengali script.
- **Gemini-2.5-Flash:** The smooth, curved lines of the Bengali script suggest the use of a brush, likely a fine-tipped one, to achieve such fluidity. The artist's handedness would be most evident in the consistent direction and pressure applied to the brushstrokes.
- **InternVL3.5-72B:** In creating the mural, the smooth, curved lines of the Bengali script would most likely show the artist's handedness, particularly if the lines consistently curve more naturally in one direction. If the curves are more fluid and precise on one side, it could suggest that the artist is either right-handed or left-handed, depending on the direction of the dominant curves.
- **Kimi-VL-A3B-Thinking-2506:** If the artist is left-handed, they might use the brush in their left hand, leading to strokes that curve more naturally. Alternatively, the tool itself (like a specific brush tip) might be designed for left or right handed use.

Figure 37: Some examples from the benchmark (Orientation: Tool Handedness) and evaluation results.

**Relative Positioning : Alignment Patterns**

**Question:** Which of the objects is the closest to the centerline of the image?

**Options**
1. Rice
2. Nut
3. Avocado
4. Berries

**Gold MCQ Answer:** 4

**Gold Open ended Answer:** Berries

**MCQ Answers:**
- **GPT-5-mini:** 4
- **Gemini-2.5-Flash:** 4
- **InternVL3.5-72B:** 4
- **Kimi-VL-A3B-Thinking-2506:** 3

**Open-ended Answers:**
- **GPT-5-mini:** The red bowl of berries (raspberries and blueberries) is closest to the image's centerline. It sits almost directly in the middle of the composition, slightly left of the exact center..
- **Gemini-2.5-Flash:** The red bowl containing raspberries and blueberries is the closest object to the centerline of the image. Its position is almost perfectly centered horizontally, and it is located in the upper-middle section of the image vertically. While other objects like the avocado halves are also quite central, the red bowl's overall placement makes it the most aligned with the image's central axis.
- **InternVL3.5-72B:** In the image, the avocado halves are the closest to the centerline. One avocado is placed near the top center, and the other is slightly below it to the right, straddling the vertical centerline of the image.
- **Kimi-VL-A3B-Thinking-2506:** The bowl of mixed berries is closest to the centerline.

Figure 38: Some examples from the benchmark (Relative Positioning: Alignment Patterns) and evaluation results.

**Relative Positioning : Betweenness Relationships**

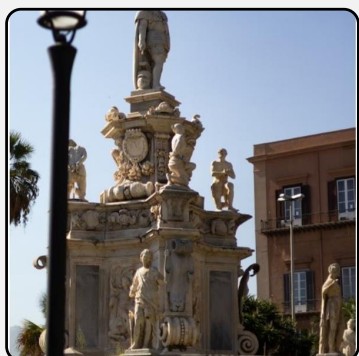

**Question: What is the object between the building and the street lamp?**

**Options**
1. **A monument of various statues**
2. **Statue**
3. **Tree**
4. **Humans**

**Gold MCQ Answer: 1**

**Gold Open ended Answer: A monument of various statues**

**MCQ Answers:**
- **GPT-5-mini:** 1
- **Gemini-2.5-Flash:** 1
- **InternVL3.5-72B:** 1
- **Kimi-VL-A3B-Thinking-2506:** 1

**Open-ended Answers:**
- **GPT-5-mini:** Between the building and the street lamp stands a stone statue of a person. It appears to be one of the sculpted figures that are part of the fountain in the foreground.
- **Gemini-2.5-Flash:** The object located between the building and the street lamp is a large, ornate monument. This monument features multiple levels adorned with various statues and intricate carvings. At its very top, there is a prominent statue of a person.
- **InternVL3.5-72B:** The object between the building and the street lamp is a detailed stone monument or fountain. It features several statues, including a prominent figure standing on top and additional figures around its structure. The monument is ornate, with decorative elements and possibly bas-reliefs on its sides, and it appears to be situated in an open, possibly public area.
- **Kimi-VL-A3B-Thinking-2506:** The object between the building and the street lamp is a detailed stone monument or fountain. The monument is ornate, with decorative elements and possibly bas-reliefs on its sides, and it appears to be situated in an open, possibly public area.

Figure 39: Some examples from the benchmark (Relative Positioning: Betweenness Relationships) and evaluation results.

## Relative Positioning : Corner/Angle Positioning

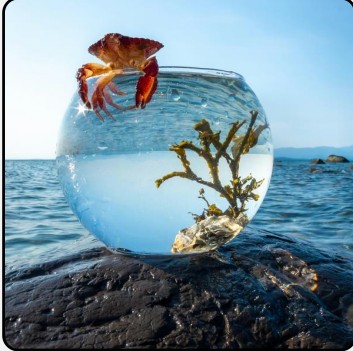

**Question: What is in the top left corner of the plant?**

**Options**
1. A lobster
2. A crab
3. An octopus
4. Nothing

**Gold MCQ Answer: 2**

**Gold Open ended Answer: A crab**

**MCQ Answers:**
- **GPT-5-mini: 2**
- **Gemini-2.5-Flash: 2**
- **InternVL3.5-72B: 2**
- **Kimi-VL-A3B-Thinking-2506: B**

**Open-ended Answers:**
- **GPT-5-mini:** A crab is in the top left corner of the plant, perched on the rim of the fishbowl. It's hanging over the water with its claws visible.
- **Gemini-2.5-Flash:** The plant in the image is a type of seaweed or coral, and it is contained within a glass bowl. In the top left corner of this plant, there is a branch of the plant extending upwards and slightly to the left. This branch is a dark, greenish-brown color.
- **InternVL3.5-72B:** In the top left corner of the plant, there is a small crab that is perched on edge of the bowl, outside the water. It is gripping the rim with its legs, and its body is partially inside the bowl. The background shows the open water, providing contrast to the scene.
- **Kimi-VL-A3B-Thinking-2506:** The answer is the crab

Figure 40: Some examples from the benchmark (Relative Positioning: Corner/Angle Positioning) and evaluation results.

## Relative Positioning : Directional Relations

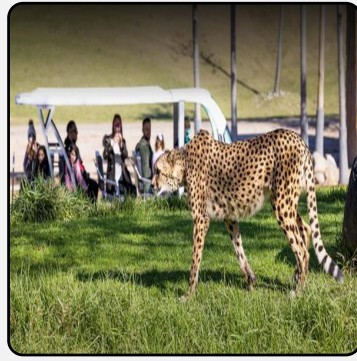

**Question: What is located directly behind the cheetah?**

**Options**
1. **Person**
2. **Tree**
3. **Saffari vehicle**
4. **Cheetah**

**Gold MCQ Answer: 3**

**Gold Open ended Answer: Saffari vehicle**

**MCQ Answers:**
- **GPT-5-mini:** 3
- **Gemini-2.5-Flash:** 3
- **InternVL3.5-72B:** 3
- **Kimi-VL-A3B-Thinking-2506:** 1

**Open-ended Answers:**
- **GPT-5-mini:** Directly behind the cheetah is a white open-sided tour vehicle or tram. Several people are seated inside the tram, watching and taking photos.
- **Gemini-2.5-Flash:** Directly behind the cheetah, there is a white open-sided vehicle, likely a safari or tour cart. Inside this vehicle, several people are seated, observing the cheetah. Some of these individuals are holding up cameras or phones, seemingly taking pictures or videos.
- **InternVL3.5-72B:** Directly behind the cheetah, there is a group of people sitting and standing around, appearing to watch or interact with the cheetah. They are positioned near a small covered vehicle, which looks like a safari cart or shuttle. The area is a grassy patch, and there are trees and a hill in the further background.
- **Kimi-VL-A3B-Thinking-2506:** People seated on benches under a canopy are located directly behind the cheetah.

Figure 41: Some examples from the benchmark (Relative Positioning: Directional Relations) and evaluation results.

**Relative Positioning : Proximity Gradients**

**Question: Which book is closer to the book "american marriage" - "atonement" or "cloud atlas"?**

**Options**
1. **Cloud Atlas**
2. **Atonement**
3. **Crossed**
4. **nan**

**Gold MCQ Answer: 2**

**Gold Open ended Answer: Atonement**

**MCQ Answers:**
- **GPT-5-mini:** 2
- **Gemini-2.5-Flash:** 2
- **InternVL3.5-72B:** 2
- **Kimi-VL-A3B-Thinking-2506:** 2

**Open-ended Answers:**
- **GPT-5-mini:** "Atonement" is closer to "american marriage" than "Cloud Atlas." "Atonement" sits just to the right with only one small book between them, while "Cloud Atlas" is further to the right on the shelf.
- **Gemini-2.5-Flash:** "Atonement" is closer to "An American Marriage" than "Cloud Atlas." "Atonement" is directly adjacent to "An American Marriage" on its right side. "Cloud Atlas" is further to the right, with "Americanah" separating it from "Atonement."
- **InternVL3.5-72B:** The book "atonement" is closer to "american marriage" than "cloud atlas." In the image, "atonement" is directly next to "american marriage," while "cloud atlas" is located further to the left..
- **Kimi-VL-A3B-Thinking-2506:** Cloud Atlas is closer to "American Marriage" than "Atonement."

Figure 42: Some examples from the benchmark (Relative Positioning: Proximity Gradients) and evaluation results.

# K  STATISTICAL ROBUSTNESS AND DATASET STABILITY

To ensure that the benchmark results are not artifacts of random initialization or dataset noise, we evaluate both model robustness across runs and dataset stability across repeated trials. Such analyses are essential for small to medium scale benchmarks, where minor fluctuations can lead to misleading fine-grained conclusions (Drummond, 2009; Drummond & Japkowicz, 2010).

Let $N$ denote the number of items *questions* in the benchmark and $R = 3$ the number of independent runs of the same model. Each run uses a distinct random seed while preserving all other hyperparameters.

## K.1  MODEL ROBUSTNESS

For checking model evaluation robustness, we perform several statistical studies, as described below.

### K.1.1  MULTIPLE RUN AVERAGES AND DEVIATIONS

Let $a_{i,r} \in 0, 1$ denote the correctness of item $i \in 1, \ldots, N$ in run $r \in 1, 2, 3$. The per-item average accuracy is:

$$\bar{a} * i = \frac{1}{R} \sum *r = 1^R a_{i,r} \tag{1}$$

and its corresponding standard deviation is:

$$\sigma_i = \sqrt{\frac{1}{R-1} \sum_{r=1}^{R} (a_{i,r} - \bar{a}_i)^2} \tag{2}$$

The overall mean accuracy per run is:

$$A_r = \frac{1}{N} \sum_{i=1}^{N} a_{i,r} \tag{3}$$

and its standard deviation across runs is:

$$\sigma_A = \sqrt{\frac{1}{R-1} \sum_{r=1}^{R} (A_r - \bar{A})^2} \tag{4}$$

where $\bar{A} = \frac{1}{R} \sum_{r=1}^{R} A_r$ is the overall mean accuracy across all runs.

### K.1.2  INTRA-CLASS CORRELATION (ICC)

To quantify agreement between runs, we compute the Intra-Class Correlation coefficient (Shrout & Fleiss, 1979), using a two-way mixed-effects model:

$$\text{ICC}(3, k) = \frac{\sigma_{\text{between}}^2}{\sigma_{\text{between}}^2 + \frac{\sigma_{\text{within}}^2}{k}} \tag{5}$$

where $\sigma_{\text{between}}^2$ is the variance between items and $\sigma_{\text{within}}^2$ is the residual variance across runs. ICC values above 0.75 indicate good reliability, while values above 0.9 imply excellent consistency (Koo & Li, 2016).

## K.2  DATASET STABILITY AND INTERNAL CONSISTENCY

We also analyze the stability of the dataset itself, whether question difficulty remains consistent across runs.

### K.2.1 RESAMPLING STUDY

To further assess **robustness**, we perform a resampling analysis. From the total of $C = 30$ subcategories, we randomly sample subsets of size $S \in 20, 25$ for each sub-category from each of the $R = 3$ runs, across both models and evaluation types (MCQ and open-ended).

For each run $r$ and subset size $S$, the accuracy is computed as:

$$A_{r,S} = \frac{1}{S} \sum_{i=1}^{S} a_{i,r,S}, \tag{6}$$

where $a_{i,r,S}$ denotes the accuracy for subcategory $i$ in run $r$ under subset size $S$.

The mean accuracy across runs for each subset size is:

$$\bar{A} * S = \frac{1}{R} \sum *r = 1^R A_{r,S}, \tag{7}$$

and the corresponding standard deviation:

$$\sigma_S = \sqrt{\frac{1}{R-1} \sum_{r=1}^{R} (A_{r,S} - \bar{A}_S)^2}. \tag{8}$$

As a complementary check, the Wilcoxon signed-rank test assesses median differences between paired samples. Let $d_r = A_{r,20} - A_{r,25}$ be the difference in accuracies for run $r$, excluding ties ($d_r \neq 0$). The test statistic is computed as:

$$W = \min(W^+, W^-), \tag{9}$$

where $W^+$ and $W^-$ are the sums of signed ranks of positive and negative differences, respectively. The standardized $z$-score is given by:

$$z = \frac{W - \frac{n(n+1)}{4}}{\sqrt{\frac{n(n+1)(2n+1)}{24}}}, \tag{10}$$

where $n$ is the number of nonzero pairs.

If the resulting $p$-value satisfies $p > 0.05$, we again fail to reject $H_0$, confirming that the accuracies from 20- and 25-sample subsets are statistically indistinguishable.

### K.2.2 ITEM-LEVEL CONSISTENCY

For each item $i$, its difficulty can be defined as $d_i = 1 - \bar{a} * i$. If item difficulty is consistent, the correlation between runs should be high:

$$\rho * r_1, r_2 = \frac{\text{Cov}(a_{\cdot,r_1}, a_{\cdot,r_2})}{\sigma_{r_1} \sigma_{r_2}} \tag{11}$$

A high mean pairwise correlation ($\bar{\rho} > 0.8$) suggests stable item difficulty and reliable dataset behavior.

### K.2.3 CRONBACH'S ALPHA

To measure overall internal consistency, we compute Cronbach's alpha (Cronbach, 1951):

$$\alpha = \frac{R}{R-1} \left( 1 - \frac{\sum_{r=1}^{R} s_r^2}{s_T^2} \right) \tag{12}$$

where $s_r^2$ is the variance of run $r$ and $s_T^2$ is the variance of the total scores aggregated across runs. Values of $\alpha \geq 0.9$ indicate excellent internal consistency (Tavakol & Dennick, 2011).

### K.3 Empirical Results

We perform the analysis on two models: one large proprietary model, **Gemini-2.5-Flash**, and one small open-source model, **Qwen-2.5-vl-7b-Instruct**, thereby covering both large- and small-scale models as well as proprietary and open-source paradigms. We also conduct all tests in both multiple-choice (MCQ) and open-ended evaluation formats (treated as binary: correct or incorrect). For statistical analysis, we used the following Python packages and functions: `scipy.stats`[2] (Virtanen et al., 2020) for `wilcoxon` test; `pingouin`[3] (Vallat, 2018) for `intraclass correlation` and `cronbach alpha`; and `numpy`[4]'s (Harris et al., 2020) `numpy.corrcoef` for correlations.

#### K.3.1 Model Evaluation Robustness

**Multiple Run Averages and Deviations**

We analyzed the stability of model accuracy across $R = 3$ independent runs. The standard deviation of accuracy ($\sigma_A$) across runs is negligible ($< 0.6\%$), indicating that the reported scores are precise estimates unaffected by random seeding noise.

Table 26: Model Accuracy and Variance across 3 Independent Runs.

| Model | Format | Mean Accuracy ($A$) | Std Dev ($\sigma_A$) |
|---|---|---|---|
| Qwen-2.5-vl-7b | MCQ | 0.4079 | 0.0021 |
| Gemini-2.5-Flash | MCQ | 0.5193 | 0.0054 |
| Qwen-2.5-vl-7b | Open | 0.1936 | 0.0047 |
| Gemini-2.5-Flash | Open | 0.3124 | 0.0042 |

The consistently low standard deviations across all models and evaluation formats indicate that performance estimates are highly stable and reproducible, with minimal sensitivity to stochastic run-to-run variation. This stability suggests that observed accuracy differences meaningfully reflect underlying model capability rather than noise introduced by random initialization or decoding. Notably, although open-ended evaluation exhibits slightly higher variance than MCQ, the absolute magnitude remains negligible, confirming that free-form generation is also robustly assessed under the current protocol. These findings imply that the dataset size and scoring methodology are sufficient to yield convergent accuracy estimates without requiring extensive repeated runs. Consequently, single-run evaluations are likely to be reliable for large-scale comparisons, while multi-run analyses primarily serve to validate robustness. Overall, this strengthens the empirical credibility of the reported results and supports confident cross-model and cross-format performance interpretation.

**Intra-Class Correlation (ICC)**

We computed ICC(3,k) to quantify the reliability of the scoring mechanism.

Table 27: Intra-Class Correlation (ICC) Reliability Scores.

| Model | Format | ICC(3,k) | Interpretation |
|---|---|---|---|
| Qwen-2.5-vl-7b | MCQ | 0.988 | Excellent |
| Gemini-2.5-Flash | MCQ | 0.990 | Excellent |
| Qwen-2.5-vl-7b | Open | 0.983 | Excellent |
| Gemini-2.5-Flash | Open | 0.981 | Excellent |

The consistently high ICC(3,k) scores demonstrate that the evaluation exhibits near-perfect reliability across independent runs, regardless of model size or response format. This indicates that item-level difficulty ordering is preserved, and that performance fluctuations do not alter relative rankings among samples. Such strong agreement suggests that the scoring pipeline is internally consistent and not sensitive to stochastic variation in model outputs. Importantly, this rules out random guessing as a dominant factor, even in open-ended generation, where variability is typically higher. From

---

[2]https://scipy.org/
[3]https://pingouin-stats.org/build/html/index.html
[4]https://numpy.org/

an evaluation standpoint, these results imply that the benchmark provides a dependable signal for assessing model capability. Consequently, longitudinal comparisons and fine-grained ablations can be conducted with high confidence in measurement stability.

Overall, the combined evidence from multiple-run variance analysis and ICC reliability demonstrates that the evaluation is highly robust to stochastic effects. Both absolute performance (low standard deviation) and relative performance (near-perfect ICC) remain stable across independent runs, confirming that results are not driven by random seeding or decoding noise. This robustness ensures that observed performance gaps across models and formats reflect genuine capability differences. As a result, the evaluation framework provides a reliable and reproducible basis for comparative analysis and benchmarking.

### K.3.2 DATASET STABILITY AND INTERNAL CONSISTENCY

**Resampling Study** To ensure statistical reliability, this resampling procedure is repeated over **1,000 independent trials**, each time drawing new random subsets of size $S = 20$ and $S = 25$ from the available subcategories. Repeating the test at this scale stabilizes the distribution of accuracy differences and $p$-values, ensuring that conclusions are not driven by a particular random draw. The consistent failure to reject $H_0$ across all trials therefore provides strong evidence that the observed robustness is systematic rather than incidental.

Table 28: Resampling Study (1,000 Trials): Wilcoxon Signed-Rank Test ($S = 20$ vs. $S = 25$).

| Model | Format | Mean $p$-value | % Trials $p > 0.05$ | Conclusion |
|---|---|---|---|---|
| Qwen-2.5-vl-7b | MCQ | 0.290 | 100% | No Sig. Diff. |
| Gemini-2.5-Flash | MCQ | 0.285 | 100% | No Sig. Diff. |
| Qwen-2.5-vl-7b | Open | 0.294 | 100% | No Sig. Diff. |
| Gemini-2.5-Flash | Open | 0.294 | 100% | No Sig. Diff. |

The resampling results provide strong empirical evidence that accuracy estimates have already converged at relatively small subcategory sizes. The consistently high mean (p)-values and the absence of statistically significant differences across all 1,000 trials indicate that performance metrics are insensitive to moderate reductions in sample size. This stability holds uniformly across models and evaluation formats, suggesting that neither model scale nor output structure introduces additional sampling uncertainty. Importantly, these findings imply that the variance in accuracy is dominated by model behavior rather than data sparsity at the subcategory level. As a result, the current subcategory size ($\approx 50$) can be considered statistically redundant, offering a substantial safety margin beyond the minimum required for reliable estimation. This supports the validity of the benchmark design and suggests that future extensions could reallocate samples across categories without compromising statistical robustness.

**Item-Level Consistency and Cronbach's Alpha** We measured the mean pairwise Pearson correlation ($\bar{\rho}$) between runs and Cronbach's Alpha to assess internal consistency.

Table 29: Item-Level Consistency Metrics.

| Model | Format | Mean Pairwise $\bar{\rho}$ | Cronbach's $\alpha$ |
|---|---|---|---|
| Qwen-2.5-vl-7b | MCQ | 0.966 | 0.988 |
| Gemini-2.5-Flash | MCQ | 0.972 | 0.990 |
| Qwen-2.5-vl-7b | Open | 0.950 | 0.983 |
| Gemini-2.5-Flash | Open | 0.946 | 0.981 |

The extremely high mean pairwise correlations ($> 0.94$) demonstrate strong item-level stability, indicating that the relative difficulty of individual questions is preserved across independent runs. This consistency shows that model responses are systematically structured rather than subject to random variation, even under stochastic decoding. The similarly high Cronbach's $\alpha$ values ($> 0.9$) further confirm that the dataset exhibits excellent internal coherence, with items contributing reliably to the overall accuracy measure. Notably, the slight reductions observed in open-ended formats are expected given their greater expressive flexibility, yet remain well within the range of excellent reliability. Together, these results indicate that the benchmark functions as a unified and internally

consistent measurement instrument. Consequently, model performance differences can be confidently attributed to true capability variation rather than instability at the item level.

### K.3.3 OVERALL DISCUSSION

Our expanded statistical testing shows our evlauiton and datset is robust **ICC and Cronbach's** $\alpha$ **scores consistently exceed 0.98**, and the **standard deviation of accuracy across runs is negligible**. Furthermore, the **resampling study** confirms that even smaller subsets produce statistically indistinguishable results from the full set, proving that our subcategory sizes ($\sim 50$) are sufficient for robust conclusions. These results demonstrate that the granular analysis in the paper reflects genuine model capability differences, not statistical noise.

Our expanded statistical testing shows that the evaluation protocol and dataset are highly robust, with **ICC and Cronbach's** $\alpha$ **scores consistently exceeding 0.98**, indicating near-perfect reliability and internal consistency. The **standard deviation of accuracy across runs remains negligible** ($< 0.6\%$), confirming that reported model performance is stable and not influenced by stochastic effects such as random seeding or decoding variability. In addition, the **resampling study** demonstrates that even substantially smaller subsets ($N = 20$ and $N = 25$) yield statistically indistinguishable accuracy estimates compared to the full set, establishing that the chosen subcategory size ($\sim 25$) is well beyond the threshold required for convergence. Collectively, these results show that both absolute scores and relative performance rankings are stable across runs and sampling regimes. This statistical robustness ensures that fine-grained comparisons across models, formats, and subcategories are meaningful. Consequently, the granular analyses reported in the paper reflect genuine differences in model capability rather than artifacts of sampling variance or evaluation noise.

