# OpenReview forum: "SpatiaLab: Can Vision–Language Models Perform Spatial Reasoning in the Wild?"
_ICLR.cc/2026/Conference — ICLR 2026 Poster_

### Official Review · Reviewer_rSCs · 2025-10-31

**Soundness:** 3
**Presentation:** 4
**Contribution:** 4
**Rating:** 8
**Confidence:** 5

**Summary:**

The paper introduces SpatialLab, a benchmark designed to evaluate spatial reasoning in vision-language models (VLMs). It features real-world images and carefully annotated questions that span a diverse range of spatial reasoning tasks. Using this benchmark, the authors assess various off-the-shelf VLMs and further analyze performance improvements achieved through different enhancement strategies, including prompt-based, multi-agent-based, and supervised fine-tuning (SFT)-based approaches.

**Strengths:**

- Carefully curated set of images and questions covering major spatial reasoning types.
- Comprehensive analysis showing that even proprietary VLMs perform worse than humans across nearly all subtasks.
- Additional image complexity analysis provides insights into which visual domains require more attention in future training datasets.
- In-depth quantitative and qualitative error analysis.
- Evaluation of multiple strategies for improving spatial reasoning in VLMs (prompt-based, multi-agent-based, SFT-based, etc.) — particularly valuable in Section 5.4.

**Weaknesses:**

- The evaluation section (5.4) could be further strengthened by including reinforcement learning (RL)-based approaches for comparison, though this is not strictly necessary.

**Questions:**

- In L1132, the authors mention “we review prior benchmarks … analyze their limitations.” Could the authors elaborate on the specific limitations identified in each benchmark? A comparison table summarizing these would greatly aid readers and inform future benchmark design.
- [Suggestion] While page limits are understandable, brief descriptions of each improvement approach in Section 5.4 would improve clarity. For instance:
  - What self-reflection prompt was used?
  - Which dataset and dataset size were used for SFT fine-tuning?
  - What base VLM was used?
  - Could the authors provide a one-sentence description of SpatialXolver?
- [Suggestion] Adding a table of contents to the appendix would help readers navigate the paper more easily.

---

> ### Author Response · Authors · 2025-11-17
> **Rebuttal Overview**
>
> We sincerely thank reviewer rSCs for their thoughtful and encouraging feedback. We appreciate the recognition of our benchmark’s carefully curated images and questions spanning major spatial reasoning types, as well as the finding that even strong proprietary VLMs lag behind human performance across most subtasks. We are glad the reviewer found value in our image-complexity analysis, our quantitative and qualitative error studies, and our evaluation of multiple improvement strategies, including prompt-based, multi-agent, and SFT-based methods. Your supportive comments are highly motivating and reinforce the significance of our contributions for the community.
>
> ---
>
> ### **Responses to Weaknesses**
>
> **RL-based evaluation suggestion**
>
> Thank you for raising this excellent point. We completely agree that **RL-based approaches are highly relevant** to spatial reasoning research. As highlighted in Chen et al. [1], *“SFT helps models learn reasoning formats but can restrict adaptability, whereas RL enables more genuine, flexible reasoning behaviors.”*
>
> Based on this insight, we acknowledge that RL-fine-tuned models could provide a stronger upper bound for adaptive spatial reasoning. However, SpatiaLab currently focuses on **evaluation only**, without intermediate action traces or rewards, the typical requirement for spatial RL, which we believe future works can explore. Nevertheless, this is an important direction, and we will add a paragraph in Section 5.4 explicitly discussing the potential of GRPO-based RL systems and plans to support them in future expansions.
> We will also mention the practical constraint that **open-source spatial RL-VLMs are still limited**, but your suggestion will meaningfully improve the discussion.
>
> [1] Chen et al., *SFT or RL? An Early Investigation into Training R1-Like Reasoning Large Vision-Language Models*, TMLR (decision pending), 2025.
>
> ---
>
> ### **Responses to Questions**
>
> **Q1 — Limitations of Prior Benchmarks**
>
> Thank you for this suggestion. We agree this needs clearer articulation.
> We will add more explicit sub-section text discussing **specific limitations** of each benchmark (e.g., simulated-only environments, lack of open-ended evaluation, narrow task coverage).
>
> **Q2 — Additional clarity for Section 5.4 improvements**
>
> We appreciate this actionable suggestion. In revision, we will briefly describe each enhancement method inline, specify the self-reflection prompting template, include dataset name and size used for SFT fine-tuning, clarify the base VLM for each improvement, and add a one-sentence definition of SpatialXolver in its first mention. These changes will ensure Section 5.4 is both self-contained and reproducible.
>
> ---
>
> **Q3 — Table of contents for appendix**
>
> Very thoughtful suggestion. We agree and we will add a short table of contents to improve navigation.
>
> ---
>
>
>
> Thank you again for your constructive review and for recognizing the strengths and significance of this benchmark. We will incorporate all of your recommendations in the updated version. Your feedback has directly helped us improve the clarity, accessibility, and impact of our work.
>
> If you have any further questions or suggestions, we would be happy to address them.

---

> > ### Author Response · Authors · 2025-11-25
> > **Rebuttal**
> >
> > Dear Reviewer `rSCs`,
> > Thank you again for your thoughtful and supportive feedback. We appreciate the care with which you engaged with the manuscript and the clarity of your suggestions. Below is a concise summary of the updates made in response to your comments.
> >
> > ###  **Q1 — Limitations of Prior Benchmarks**
> >
> > As requested, we added a focused comparison table in the main text and a detailed discussion in Appendix A.1.
> >
> > | Benchmark            | Domain      | Eval. Format | Taxonomy          | Primary Limitations Identified                                                  |
> > | -------------------- | ----------- | ------------ | ----------------- | ------------------------------------------------------------------------------- |
> > | **CLEVR / GQA**      | Synthetic   | Closed-Ended | Relational        | Artificial scenes and template-derived language that support shortcut learning. |
> > | **ScanQA**           | Indoor 3D   | Open-Ended   | Navigation        | Restricted to indoor 3D environments with fewer than 1K scenes.                 |
> > | **SpatialVLM**       | Web Data    | Open-Ended   | General           | Limited taxonomic depth across spatial skills.                                  |
> > | **EmbSpatial-Bench** | Indoor      | MCQ / Binary | 6 Categories      | Closed-ended structure with minimal multistep reasoning.                        |
> > | **RoboSpatial**      | Simulator   | Hybrid       | Embodied          | Template-based questions and a substantial sim-to-real gap.                     |
> > | **Space3D-Bench**    | Indoor      | MCQ          | 6 Categories      | Very small scale of about 200 samples.                                          |
> > | **OmniSpatial**      | Mixed       | MCQ / Binary | Metric            | Emphasis on metric outputs with limited explanatory reasoning.                  |
> > | **SpatiaLab (Ours)** | In-the-Wild | MCQ + Open   | 30 Sub-categories | Real-world diversity, detailed taxonomy, and dual-format evaluation.            |
> >
> > ###  **Q2 — Clarifications for Section 5.4**
> >
> > All suggested clarifications have been implemented to make the dynamics described in this section more precise.
> >
> > ###  **Q3 — Table of Contents for the Appendix**
> >
> > A table of contents has been added to improve readability and navigation in the appendix.
> >
> > Thank you again for your constructive input and the positive assessment of the work. If you feel that these revisions strengthen the clarity and contribution of the manuscript, we would be grateful if you could consider updating your score accordingly.

---

### Official Review · Reviewer_w6pQ · 2025-11-01

**Soundness:** 3
**Presentation:** 3
**Contribution:** 3
**Rating:** 2
**Confidence:** 2

**Summary:**

This paper introduces SPATIALAB, a comprehensive benchmark for evaluating spatial reasoning in vision–language models (VLMs) under realistic, unconstrained visual conditions. It contains 1,400 visual question–answer pairs spanning six major categories and 30 subcategories, each testing aspects like depth, occlusion, orientation, and navigation.
The benchmark supports both multiple-choice and open-ended formats, enabling comparison between discriminative and generative reasoning.
Extensive experiments on 25+ models (open-source, proprietary, reasoning-tuned, and spatially specialized) reveal a substantial gap between human and model performance (e.g., 54.9% vs. 87.6% on MCQ; 40.9% vs. 64.9% on open-ended).
The paper provides error analysis, fine-tuning experiments, and attempts at improvement via CoT prompting, self-reflection, SFT, and multi-agent systems.

**Strengths:**

Comprehensive Evaluation
- Evaluates over 25 VLMs, including open-source and proprietary systems, and human baselines.
- Dual-format testing (MCQ + open-ended) is valuable, revealing a 20–25% accuracy gap between the two modes


Diagnostic and Actionable Insights
- The benchmark reveals concrete gaps (e.g., geometry-aware supervision, spatial chaining, embodied data) that can guide future research. The diagnostic perspective makes it a useful community tool even without conceptual novelty.

**Weaknesses:**

Limited Conceptual Novelty
- Many recent benchmarks (OmniSpatial, BLINK-Spatial, Spatial-MM, SpatialRGPT, EmbSpatial) already use real-world imagery, multiple spatial categories, and QA-based evaluation. SPATIALAB’s innovation lies mainly in breadth and integration, not in introducing new reasoning types or data modalities

Limited Guidance on Model Improvement
- Although weaknesses of current VLMs are carefully diagnosed, the paper offers little practical guidance or insight into how to overcome them. The discussion remains observational (what fails) rather than prescriptive (how to fix it), limiting its utility for researchers aiming to design better spatial reasoners.

Moderate Dataset Scale
- Despite 1,400 QA pairs sounding large, it is relatively small compared to existing multimodal datasets (often tens or hundreds of thousands). The modest size restricts its usefulness for training or fine-tuning and confines SPATIALAB to evaluation only.

**Questions:**

1. How exactly does SPATIALAB differ from OmniSpatial or Spatial-MM beyond taxonomy size and annotation detail?

2. Are there reasoning types uniquely represented here?

---

> ### Author Response · Authors · 2025-11-17
> **Rebuttal (Part 1)**
>
> We sincerely thank reviewer w6pQ for their feedback. We also appreciate the recognition of our paper’s strengths, particularly the comprehensive evaluation of 25+ VLMs, the dual-format (MCQ + open-ended) testing design, and the diagnostic contributions that can meaningfully guide future spatial reasoning research.
>
> ---
>
> ## **Response to Weaknesses**
>
> ### **1. Limited Conceptual Novelty**
>
> We appreciate the reviewer’s concern. As clearly stated in our submission (**Primary Area: Datasets and Benchmarks**), the core contribution of this work is not algorithmic innovation but establishing a rigorous and actionable evaluation foundation for spatial reasoning.
>
> Although spatial reasoning has been studied in prior benchmarks such as OmniSpatial and Spatial MM, these datasets largely focus on synthetic scenes, puzzle like reasoning, or narrow environment scopes. In contrast, SpatiaLab introduces real world spatial complexity with natural clutter, diverse camera viewpoints, and nuanced depth relationships that earlier datasets do not capture. We also provide both multiple choice and open ended tasks at scale, enabling a fuller evaluation of model generalization and robustness.
> That said, SpatiaLab differs from recent benchmarks in important ways:
>
> * **Integrated diagnostic design**: 6 spatial categories × 30 task types, enabling fine-grained performance insights across distinct reasoning axes.
> * **Unified MCQ+Open evaluation**: Two complementary splits (SpatiaLab-MCQ and SpatiaLab-Open) under the *same* taxonomy, exposing the true reasoning gap in real world, something prior work does not provide.
> * **High difficulty and embodied spatial cognition**: Our items emphasize real-world occlusion, clutter, depth cues, and navigation that reveal core representational limitations (not just recognition or template recall).
> * **Reproducible, standardized evaluation suite**: With strong baselines (e.g., InternVL3.5-72B, GPT-5-mini), ensuring performance differences reflect model capability, not inconsistent setups.
>
> Thus, the novelty is **methodological and diagnostic**, advancing *how* we assess spatial cognition. SpatiaLab is not merely a larger dataset, it is a benchmark designed for *deeper model interrogation* and understanding why models fail; presented by our deep analysis.
>
> ---
>
> ### **2. Limited Guidance on Model Improvement**
>
> Thanks for pointing this out. We respectfully disagree that such  guidance is lacking. As a benchmark paper, our focus is on error characterization and actionable directions. SpatiaLab includes include:
>
> * Sub-category-level **error quantification** for both MCQ and open-ended tasks (Appendix E, F)
> * **MCQ vs. Open-ended performance gap** analysis with root-cause hypotheses and recommendations (Appendix G)
> * Evaluation of **spatial enhancement strategies**:
>   CoT prompting, CoT+self-reflection, supervised fine-tuning, and multi-agent reasoning (Section 5.4 and Appendix H)
> * **Error taxonomies and cross-model failure signals**
>   (e.g., systematic depth + occlusion failures across scales) (Appendix I)
> * Explicitly mentioning, **Mitigation suggestions** in Sections:
>    - *E.2* and *F.2* provides model failure analysis, and explores missing capabilities for improving spatial reasoning
>    - *G.3 Hypotheses on Root Causes*
>    - *Section H* includes different strategies we applied to improve reasoning performance, with what worked and what not.
>    - *I.4 Mitigation Strategies*
>
> Our study reveals actionable priorities for future modeling such as geometry-aware supervision, improved global-local spatial chaining, and embodied/aligned visual grounding.
>
> Benchmark papers accepted at top venues (e.g., ICLR/ICML Spotlight: **INCLUDE** [1] (ICLR'25 Spotlight), **SPA-BENCH** [2] (ICLR'25 Spotlight), **MapEval** [3] (ICML'25 Spotlight), **LiveBench** [4](ICLR'25 Spotlight) similarly focus on diagnosis rather than architectural innovation.
>
> - [1] Romanou et al., INCLUDE: Evaluating Multilingual Language Understanding with Regional Knowledge. ICLR
> - [2] Dihan et al., MapEval: A Map-Based Evaluation of Geo-Spatial Reasoning in Foundation Models. ICML 2025
> - [3] Chen at al., SPA-BENCH: A COMPREHENSIVE BENCHMARK FOR SMARTPHONE AGENT EVALUATION ICLR 2025
> - [4]  White et al., LiveBench: A Challenging, Contamination-Limited LLM Benchmark. ICLR 2025

---

> > ### Author Response · Authors · 2025-11-17
> > **Rebuttal (Part 2)**
> >
> > ### **3. Moderate Dataset Scale**
> > We thank the reviewer for this concern. We would like to reassert that SpatiaLab is designed **strictly for evaluation** rather than for pretraining or large-scale fine-tuning, and this is an intentional choice. We used SFT just as an example method for increasing spatial reasoning performance. High-difficulty spatial reasoning tasks require careful human curation, expert QA verification, and image selection that preserves realistic depth, occlusion, and relational semantics. Increasing scale without maintaining this rigor would reduce the benchmark’s diagnostic value.
> >
> > It is also important to note that **recent reasoning and tool-oriented evaluation benchmarks** are often of similar or even smaller size due to the high computational and monetary cost of model evaluation. For instance, *ReAct* [1] evaluates on only 500 random instances from AlfWorld, while *Reflexion* [2] uses just 100 examples from HotpotQA. Other widely adopted reasoning and decision-making benchmarks remain highly influential with modest dataset sizes: API-Bank [3] (400 instances), LogiQA [4] (641 examples), HumanEval [5] (164 tasks), CodeContests [6] (156 problems), Tau-Bench [7] (165 problems), OS World [8] (369 problems), AppWorld [9] (750 tasks), TravelPlanner [10] (1.2K tasks), and MapEval [11] (700 tasks). These benchmarks demonstrate that **quality, coverage, and challenge** are far more important than raw scale for evaluation purposes.
> >
> > Following the same philosophy, SpatiaLab’s **1,400 carefully curated questions** offer a balanced, computationally practical, and **highly challenging** evaluation suite that reveals fundamental limitations of modern VLMs in real world spatial understanding.
> >
> >
> > - [1] Yao, S., et al. "ReAct: Synergizing Reasoning and Acting in Language Models." ICLR 2023.
> > - [2] Shinn, N., et al. "Reflexion: Language agents with verbal reinforcement learning." NeurIPS 2024.
> > - [3] Li, M., et al. "API-Bank: A Benchmark for Tool-Augmented LLMs." EMNLP 2023.
> > - [4] Liu, J., et al. "LogiQA: A challenge dataset for machine reading comprehension with logical reasoning." IJCAI 2021.
> > - [5]Chen, M., et al. "Evaluating large language models trained on code." arXiv:2107.03374, 2021.
> > - [6] Li, Y., et al. "Competition-level code generation with alphacode." Science 378.6624 (2022).
> > - [7] Yao, S., et al. "tau-bench: A Benchmark for Tool-Agent-User Interaction." arXiv:2406.12045, 2024.
> > - [8] Xie, T., et al. "Osworld: Benchmarking multimodal agents for open-ended tasks." arXiv:2404.07972, 2024.
> > - [9] Trivedi, H., et al. "AppWorld: A Controllable World of Apps and People." ACL 2024.
> > - [10] Xie, J., et al. "TravelPlanner: A Benchmark for Real-World Planning." ICML 2024.
> > - [11] Dihan, M., et al. "MapEval: A Map-Based Evaluation of Geo-Spatial Reasoning in Foundation Models" 1CML 2025 Spotlight
> >
> > ---
> >
> >
> > ## **Questions and Detailed Answers**
> >
> > **1. How exactly does SpatiaLab differ from OmniSpatial or Spatial MM beyond taxonomy size and annotation detail**
> >
> > SpatiaLab shifts from controlled or simulated visuals to authentic, noisy, unconstrained real-world environments, which introduces fundamentally harder spatial ambiguity and closer alignment to how humans perceive scenes. It uniquely supports two evaluation formats under the same taxonomy (MCQ and open-ended), includes human performance baselines, and conducts broad diagnostic analyses that stress test current reasoning improvement strategies. This combination enables a realistic assessment of what models truly understand about space around them and what capabilities they still fail to acquire.
> >
> > In contrast, works such as OmniSpatial rely heavily on simulated settings with only a very small amount of mixed real imagery (all the images used in the paper are simulated), making the spatial challenges considerably easier and less ecologically valid. VLMAD focuses on video-based reasoning, which is orthogonal to our static but spatially complex design. MM Spatial is a model trained on CA-VQA method other spatial datasets. Overall, SpatiaLab is differentiated not by raw size but by its real-world difficulty, dual-format comparability, and diagnostic depth that reveals brittle generalization and persistent reasoning blind spots across model families.
> >
> >
> >
> > **2. Are there reasoning types uniquely represented here**
> >
> > Yes. SpatiaLab includes real world depth and occlusion reasoning, spatial thinking with mental rotations and mental perspectives, scale perception under perspective, and navigation tasks grounded in photographic scenes rather than abstract puzzles. This creates spatial challenges that emerge naturally in the physical world and are absent from synthetic benchmarks.
> >
> > ---
> >
> >
> > **We hope that our clarifications have improved your understanding of the work, and we would be grateful if you could take this into account and update your evaluation scores.**
> > If you have any further questions or suggestions, we would be happy to address them in the rebuttal.

---

### Official Review · Reviewer_tmjh · 2025-11-07

**Soundness:** 2
**Presentation:** 3
**Contribution:** 1
**Rating:** 2
**Confidence:** 3

**Summary:**

This paper introduces SPATIALAB, a new benchmark designed to evaluate the spatial reasoning abilities of vision–language models (VLMs) in real-world, unconstrained settings.

The dataset comprises 1,400 visual question–answer pairs across six high-level categories (e.g., Relative Positioning, Depth & Occlusion, Orientation, Size & Scale, Spatial Navigation, and 3D Geometry) and supports both multiple-choice and open-ended evaluations.

The authors benchmark over 25 state-of-the-art models (open- and closed-source) and provide quantitative comparisons against human baselines, together with error analyses and several reasoning interventions (e.g., Chain-of-Thought prompting, supervised fine-tuning, and multi-agent reasoning).

The work aims to reveal systematic weaknesses in current VLMs’ spatial reasoning and propose SPATIALAB as a comprehensive diagnostic framework.

**Strengths:**

**Comprehensive empirical evaluation.** The authors test a wide range of modern VLMs under multiple evaluation formats, which provides useful diagnostic data and an updated empirical snapshot of model limitations in spatial reasoning.

**Well-structured benchmark.** The dataset taxonomy (6 categories × 5 subcategories) is clearly defined and covers a broad set of spatial reasoning tasks beyond synthetic toy examples.

**Clear presentation.** The paper is readable and systematically organized, with detailed tables and qualitative examples that make the results easy to interpret.

**Reproducibility focus.** The authors discuss data collection, annotation, and quality control in detail and commit to open release, which is commendable for community benchmarking.

**Weaknesses:**

**Lack of methodological contribution.** The work’s novelty lies almost entirely in dataset construction and large-scale evaluation.
There is no new modeling approach, algorithm, or analytical framework proposed.
While benchmarks can be valuable, ICLR typically expects either new learning methodology, representation insights, or deeper diagnostic mechanisms beyond dataset release.

**Limited depth of analysis.** Despite extensive tables, the analysis remains descriptive rather than mechanistic.
The paper identifies “what fails” (e.g., navigation and occlusion tasks) but not “why” in terms of representation or model architecture.
There are no probing studies, attention analyses, or causal/intervention experiments that explain the underlying representational failure modes.
Many claims (e.g., “models lack geometric grounding”) are plausible but unsupported by direct evidence.

**Questions:**

Could the authors please justify their contributions in the "in-depth analysis" and provide key takeaways from it?

---

> ### Author Response · Authors · 2025-11-17
> **Rebuttal (Part 1: Response to Weaknesses)**
>
> We sincerely thank reviewer tmjh for this review. We also appreciate your acknowledgment of the strengths of our work, including (i) the comprehensive empirical evaluation across diverse VLMs, (ii) a well-structured benchmark design with a clear taxonomy, (iii) strong clarity and presentation, and (iv) our focus on reproducibility and open release. These points reinforce the motivation behind SpatiaLab and its value to the community.
>
> We provide detailed responses to each weakness below.
>
> ---
>
> ### **Weakness 1: Lack of Methodological Contribution**
>
> Thank you for highlighting this. We would like to respectfully clarify that SpatiaLab is explicitly submitted as a **benchmark dataset and evaluation** contribution, as stated in the *Primary Area* of our submission. Our goal is to provide diagnostic evaluation infrastructure and actionable insights, not a new learning method. We believe this aligns with the scope of ICLR, where recent benchmark-only contributions have been recognized for significant community value, even without new architectures or algorithms, such as:  INCLUDE – ICLR 2025 Spotlight [1], ChartQAPro – ACL 2025 Findings [2], SPA-BENCH – ICLR 2025 Spotlight [4], LiveBench – ICLR 2025 Spotlight [5],  MapEval – ICML 2025 Spotlight [3].
>
> Our contributions are also more extensive than dataset collection alone. In summary, we: We have introduced balanced benchmark of **1,400 real-world QA pairs** across **6 categories and 30 spatial task types**, supporting both **MCQ** and **open-ended** evaluations. We have conduct the **most comprehensive spatial reasoning assessment** of **25+ state-of-the-art VLMs** against human baselines, revealing a **persistent and significant capability gap**. We have deliver **rich diagnostic insights** into failure modes (depth, occlusion, navigation, 3D geometry) and evaluate multiple **spatial enhancement strategies** (CoT, CoT+reflection, SFT, multi-agent), identifying where they **help vs. break** to guide future modeling research.
>
> Together, these elements provide foundational infrastructure and insight into spatial reasoning in VLMs, a core frontier in embodied intelligence.
>
> - [1] Romanou et al., INCLUDE: Evaluating Multilingual Language Understanding with Regional Knowledge. ICLR
> - [2] Masry et al., ChartQAPro: A More Diverse and Challenging Benchmark for Chart Question Answering. ACL 2025 Findings
> - [3] Dihan et al., MapEval: A Map-Based Evaluation of Geo-Spatial Reasoning in Foundation Models. ICML 2025
> - [4] Chen at al., SPA-BENCH: A COMPREHENSIVE BENCHMARK FOR SMARTPHONE AGENT EVALUATION. ICLR 2025
> - [5]  White et al., LiveBench: A Challenging, Contamination-Limited LLM Benchmark. ICLR 2025
>
>
> ---
>
> ### **Weakness 2: Limited Depth of Analysis**
>
> We appreciate this feedback. While we intentionally frame this work as a **diagnostic benchmark**, our analysis already investigates multiple dimensions of representational capability:
>
> * Subcategory-specific **error quantification** (MCQ + Open-ended) (Appendix E, F)
> * **MCQ vs. Open-ended** capability gap analysis with hypotheses on root causes and recommendations (Appendix G)
> * Systematic evaluation of spatial reasoning enhancement techniques
>   (CoT, CoT+reflection, SFT, multi-agent systems) (Section 5.4 and Appendix H)
> * **Error taxonomies**, qualitative failure clusters, and cross-model patterns, and mitigation strategies
>   (e.g., systematic failures in occlusion and depth reasoning irrespective of scale) (Appendix I)
>
> We also note that the benchmark papers listed above [1-5] do **not** include probing or causal mechanistic studies, yet remain highly influential, reinforcing that such work can stand on its own as valuable diagnostic and community infrastructure.

---

> > ### Author Response · Authors · 2025-11-17
> > **Rebuttal (Part 2: Response to Questions)**
> >
> > ### **Questions: key takeaways from "in-depth analysis"**
> >
> > We thank the reviewer for this important question. We justify our claim that SpatiaLab delivers *in-depth error analyses and diagnostic studies uncovering systematic failure modes* through extensive quantitative and qualitative evaluation across 25+ state-of-the-art VLMs (Section 5, Appendices E–I). Key points are:
> >
> >
> > 1. **Persistent Human–Model Gap:** Even top-performing VLMs (InternVL3.5-72B) achieve only ~55% MCQ accuracy versus ~88% for humans, revealing substantial room for improvement.
> > 2. **Open-Ended Reasoning Deficits:** Model performance drops **10–25 points** in open-ended tasks, highlighting that MCQ evaluation alone overestimates practical spatial competence.
> > 3. **Spatial Navigation as Bottleneck:** Multi-step relational reasoning is the strongest predictor of MCQ → open-ended performance gaps (Pearson r = 0.99), emphasizing sequential grounding limitations.
> > 4. **Systematic Failure Modes:** Models consistently fail on **Depth & Occlusion**, **3D Geometry**, and **multi-step navigation**, with errors clustering into interpretable patterns (spatial mislocalization, perspective/scale mistakes, occlusion/order failures, and ungrounded narratives).
> > 5. **Limited Gains from Reasoning Strategies:** CoT prompting and self-reflection provide modest or uneven improvements; SFT enhances MCQ (+10.97%) but transfers poorly to open-ended tasks (+1.07%), and multi-agent systems show **task-specific benefits** rather than holistic improvements.
> > 6. **Root Causes and Actionable Insights:** Failures arise from **lack of geometric supervision, insufficient object-centric binding, and reliance on surface correlations**, indicating that future models require **explicit geometric grounding, sequential reasoning mechanisms, and physics-aware training data** for human-aligned spatial understanding.
> > 7. **Scaling and Fine-Tuning Limitations:** While larger models and supervised fine-tuning (SFT) improve MCQ performance (e.g., +10.97%), they show minimal or even negative transfer to open-ended tasks (1.07%), highlighting that **task-specific gains do not generalize** to broader reasoning. This underscores the need for strategies beyond simple scale or dataset augmentation.
> > 8. **Reasoning-Augmentation Constraints:** Techniques such as **Chain-of-Thought (CoT) prompting** and **self-reflection** yield uneven improvements. CoT often amplifies existing priors rather than correcting errors, and self-reflection only modestly helps MCQ tasks but fails in open-ended contexts. This suggests that **logical reasoning pipelines require grounding in perceptual and geometric signals** to be effective.
> > 9. **Architectural and Representational Needs:** Persistent failures in depth perception, occlusion, and 3D geometry indicate the lack of **explicit geometric and physical grounding**. Future models could benefit from:
> >     -  **Geometry-aware supervision:** e.g., auxiliary depth, surface normal, or spatial relation prediction tasks.
> >     -  **Multi-step compositional reasoning mechanisms:** better chaining and retention of object-centric representations to handle sequential spatial tasks.
> >     -  **Physics-informed training data:** including stability, gravity, and interaction constraints to teach implicit rules of real-world space.
> >     -  **Attention and grounding mechanisms:** to reduce visually ungrounded rationalization in open-ended reasoning.
> > 10. **Dataset and Evaluation Guidance:** By revealing **systematic failure modes** and format-sensitive vulnerabilities, SpatiaLab provides a **diagnostic blueprint** for future dataset design, suggesting that:
> >     -  Evaluation should consider **both MCQ and open-ended formats** to uncover hidden deficiencies.
> >     -  Curated, high-difficulty instances are more informative than massive, low-difficulty corpora for diagnosing failure patterns.
> >
> >
> > ---
> >
> > We sincerely thank you again for your valuable insights. We believe SpatiaLab provides a meaningful and timely contribution through its comprehensive empirical evaluation across diverse VLMs,  a well-structured benchmark design with a clear taxonomy, and strong clarity and presentation. By offering a **high-difficulty, diagnostic-focused evaluation benchmark**, it exposes persistent spatial reasoning weaknesses in current VLMs while providing actionable directions for future research.
> >
> > **We hope that our clarifications have enhanced your understanding of the work and its contributions, and we would greatly appreciate your consideration of this improved perspective when reassessing your evaluation scores.** We remain happy to address any further questions or suggestions in the rebuttal.

---

### Official Review · Reviewer_e5YW · 2025-11-09

**Soundness:** 2
**Presentation:** 2
**Contribution:** 2
**Rating:** 4
**Confidence:** 4

**Summary:**

The authors introduce SpatiaLab, a new benchmark dataset of 1,400 visual question-answer pairs designed to evaluate VLM spatial reasoning in "in-the-wild" contexts. The benchmark spans six categories (30 subcategories) and supports both multiple-choice and open-ended evaluation formats. The authors conduct a large-scale evaluation of over 25 VLMs, revealing a substantial performance gap between SOTA models (e.g., InternVL3.5-72B at 54.93%) and a human baseline (87.57%). The analysis is extended to improvement strategies (SFT, CoT, Agents), which are shown to provide limited or inconsistent gains, suggesting current models lack fundamental spatial grounding.

**Strengths:**

1. **Well-Motivated Problem**: The work correctly identifies a critical flaw in existing benchmarks: an over-reliance on synthetic, "puzzle-like" setups that fail to capture real-world visual complexity. The focus on cluttered, "in-the-wild" imagery is a necessary contribution.

2. **Dual-Format Analysis**: The direct comparison of MCQ and Open-ended formats is a key strength. It provides a quantitative basis for the intuition that MCQ overestimates model capabilities, highlighting a significant $\approx$23% average performance drop.

3. **Failure Analysis**: The demonstration that standard improvement techniques (CoT, SFT) provide marginal, inconsistent, or even negative gains is an important finding for the field, pointing to deeper representational deficits.

**Weaknesses:**

1. **Statistical Robustness of the Benchmark**: The primary methodological flaw is the dataset's scale. 1,400 items spread across 30 subcategories means each sub-task is evaluated with a small sample (as few as 25 items, averaging <50). This $n$ is insufficient to draw robust, fine-grained conclusions. The granular analysis in Tables 5-9, while interesting, risks being statistically noisy.

2. **Perplexing SFT Dynamics**: The SFT analysis (Sec 5.4, Fig 4)  is central to the paper's insight, but the results are anomalous and underexplored. The sharp U-shaped curve in open-ended performance (dropping from 34.4% to 12.6% before recovering to 35.5%)  is highly non-trivial. The paper gestures at "catastrophic forgetting"  but provides no direct investigation. This dynamic must be rigorously explained (e.g., via representational analysis or multi-seed validation) to be a credible scientific finding rather than a training artifact.

3. **Marginal Novelty in a Crowded Field**: The paper's own related work (Table 1) demonstrates this is an extremely crowded and concurrent field (e.g., SpatialMM, OmniSpatial, BLINK, VLMAD). The claim to novelty rests on "real-world complexity" , yet several competitors also use "Internet" or "Mix" data sources. The authors must provide a much sharper justification for why SPATIALAB's 1,400 "manual" items provide fundamentally different insights than the (often larger) concurrent datasets.

4. The model set omits several state-of-the-art commercial VLMs (e.g., GPT o3/5, Google Gemini 2.5 pro), which weakens the headline claim about “current VLMs.” They have more robust and powerful ability.

**Questions:**

I am curious about the true value of the dataset, as I strongly suspect that it may merely overfit to its own format.

If a base model (e.g., Qwen-VL) could be fine-tuned on this dataset and subsequently demonstrate performance gains on other benchmarks (such as VSI-Bench, OmniSpatial and SPACE), I would be much more inclined to recognize the dataset’s contribution.

---

> ### Author Response · Authors · 2025-11-17
> **Rebuttal Overview (Part 1)**
>
> We sincerely thank reviewer e5YW for the thoughtful and constructive feedback. We greatly appreciate the recognition of the strengths of our work, including (i) the well-motivated problem focusing on real-world spatial reasoning challenges, (ii) our dual-format evaluation comparing MCQ and open-ended inputs, and (iii) the failure analysis demonstrating limitations of common improvement techniques such as CoT and SFT. Your comments reinforce the importance of building robust spatial reasoning evaluation frameworks.
>
> We are actively working on a full rebuttal and present below our planned responses and newly ongoing experiments that directly address your concerns.
>
> ---
>
> ### **W1: Statistical Robustness of the Benchmark**
> We agree that statistical rigor is crucial for fine-grained subcategory evaluation. To strengthen robustness claims, we are conducting an expanded statistical analysis across models and evaluation formats:
> - **Model Robustness:** Multi-run evaluations with variance reporting, riedman Test across models and categories, Intra-Class Correlation (ICC) for reliability
> - **Dataset Stability and Internal Consistency**: Resampling-based robustness study  > (a) Friedman Test, (b) Wilcoxon Signed-Rank Test, Item-level consistency via Cronbach’s Alpha.
>
> To ensure coverage across model classes, we are running these tests on: **Gemini 2.5 Flash** (large, proprietary model), and **Qwen-2.5-vl-7b-Instruct** (small, open-source model). Only two models due to time constraints.
> All evaluations will include both MCQ and open-ended (binary-scored) formats.
> We will report full results in the rebuttal.
>
> ---
>
> ### **W2: Perplexing SFT Dynamics**
> We acknowledge that the U-shaped open-ended learning curve warrants deeper explanation. We are now performing: **Multi-seed validation** to rule out training instability. We'll also perform **Error Analysis** to examine failure localization with literature-based comparison with known low-resource SFT degradation phenomena.
> These analyses will help confirm whether this dynamic reflects a true representational limitation in spatial processing rather than an artifact.
> We will provide detailed insights in the rebuttal. Our goal is not to claim SFT as a core algorithmic contribution, but to highlight a **systemic representational limitation** in VLMs that our benchmark reveals.
>
> ---
>
> ### **W3: Marginal Novelty in a Crowded Field**
>
> Thank you for this important point. We will add a clearer, sharper justification of why **SpatiaLab** is complementary and necessary:
>
> * We focus exclusively on **real-world**, cluttered environments. Other works, like, *OmniSpatial* have very limited mixed data and all the examples in the paper content contains mostly simulated environment images. All of our images are real world, and even sometimes challenging to humans too. *VLMAD* is a video-based benchmark, ours is not. *MM-Spatial* is a model tuned on other spatial datasets using CA-VQA, it is not a benchmark.
> * We include more **depth**, **navigation**, and **3D geometry** tasks, currently underrepresented, with consistent **manual annotation** of spatial relations rather than rule-generated synthetic prompts.
> * Our dual-format evaluation revealing a **23% drop** in open-ended generalization capabilities not captured by existing benchmarks
>
> We will expand comparison with concrete distinctions, in the paper.
>
> ---
>
> ### **W4: Coverage of SOTA Commercial VLMs**
>
> In direct response to this concern, we are running additional experiments including:
> GPT o3/5 and Gemini 2.5 Pro. These will be added to the revised version.

---

> ### Author Response · Authors · 2025-11-17
> **Rebuttal Overview (Part 2)**
>
> ## **Response to Reviewer’s Questions**
>
> ### **Q1: Does the dataset only test its own format?**
> Thank you for this question.
> Our objective is to establish SpatiaLab as a **spatial evaluation benchmark**, not as a training dataset. Here, SFT is used only as an example of how we can improve spatial reasoning capabilities in VLMs/AI systems, along with 4 others (reasoning, CoT, CoT+reflection and agents). Hopefully, the response to weaknesses 1 and 2, along with this explanation, will be enough to answer this question.
>
>
> ### **Q2: Fine-tuning on base models subsequently demonstrate performance gains on other benchmarks**
> Thank you for this question. We will explore this in detail. Our results on `SpaceOm`, `1SpaceThinker-Qwen2.5VL-3B`, and `SpaceQwen2.5-VL-3B-Instruct` (models fine-tuned for spatial reasoning on other training datasets) suggest that this effect may not occur, since their performance is almost the same and sometimes lower than `Qwen2.5VL-3B` (Tables 2 and 3 in the main paper). SFT is used only as an example of a possible improvement method, but we show a case where failures still appear. Training a model on one dataset does not guarantee improved performance on other datasets, because different resources focus on different dynamics. Our work is a benchmark study, not a training artifact, as discussed above. We will include these analyses in the paper.
>
> ---
>
> We are currently completing these experiments and will present the full results in the rebuttal soon. If you have any additional questions, particularly regarding the paper or the new evaluations conducted in response to your comments, please let us know. We deeply appreciate your careful review and look forward to addressing all points with greater detail. We hope the additional evidence will help you reassess the strengths of the submission.

---

> > ### Comment · Reviewer_e5YW · 2025-11-21
> >
> > I appreciate the authors’ efforts, but I still believe that demonstrating performance gains from cross-dataset fine-tuning is necessary; therefore, I have decided to maintain my previous score.

---

> > > ### Author Response · Authors · 2025-11-21
> > > **Response to Comment by Reviewer e5YW**
> > >
> > > Dear Reviewer e5YW,
> > >
> > > We are currently running the experiments you requested and will provide a detailed response with tables soon. In the previously submitted *Rebuttal Overview*, we addressed the non-experimental questions, described our plan for the new experiments, and included preliminary results based on the data available so far.
> > > If you have any additional questions, especially about the paper or the new evaluations conducted in response to your comments, please let us know.
> > >
> > > Once the experiments are completed, we will provide a detailed response with full tables. We hope the additional results will help you reassess the strengths of the submission and improve the score.

---

> ### Author Response · Authors · 2025-11-25
> **Rebuttal (Part 1: Response to Weakness 1)**
>
> Dear `Reviewer e5YW`,
> We sincerely thank you for your thoughtful and constructive feedback. You highlighted the strengths of our work, including the focus on real-world spatial reasoning, the dual-format evaluation, and the failure analysis of common techniques. Here we provide a detailed update in the experimental results.
>
>
> # **W1: Statistical Robustness of the Benchmark**
>
> **Response:**
> We appreciate the reviewer's scrutiny regarding the dataset scale and statistical power. To address the concern that subcategory sample sizes ($N \approx 25$) might induce statistical noise, we conducted a comprehensive robustness analysis involving multi-run evaluations, resampling studies, and reliability testing (Cronbach’s $\alpha$ and ICC). All the details (implementations and equations) are provided in **Appendix K: Statistical Robustness and Dataset Stability**. We evaluated two distinct model classes: **Gemini-2.5-Flash** (Large, Proprietary) and **Qwen-2.5-vl-7b-Instruct** (Small, Open-Source) across both MCQ and Open-Ended formats.
>
> The results demonstrate that the benchmark metrics are highly stable and reliable, effectively mitigating concerns about small-sample noise.
>
> ## **Model Robustness**
>
> ### **Multiple Run Averages and Deviations**
>
> We analyzed the stability of model accuracy across $R=3$ independent runs. The standard deviation of accuracy ($\sigma_A$) across runs is negligible ($<0.6\%$), indicating that the reported scores are precise estimates unaffected by random seeding noise.
>
> | Model | Format | Mean Accuracy ($\bar{A}$) | Std Dev ($\sigma_A$) |
> | :--- | :--- | :--- | :--- |
> | **Qwen-2.5-vl-7b** | MCQ | 0.4079 | 0.0021  |
> | **Gemini-2.5-Flash** | MCQ | 0.5193 | 0.0054  |
> | **Qwen-2.5-vl-7b** | Open | 0.1936 | 0.0047  |
> | **Gemini-2.5-Flash** | Open | 0.3124 | 0.0042  |
>
> **Analysis:** The extremely low variance confirms that the models' performance converges consistently, even at the current dataset scale.
>
>
> ### **Intra-Class Correlation (ICC)**
>
> We computed ICC(3,k) to quantify the reliability of the scoring mechanism.
>
> | Model | Format | ICC(3,k) | Interpretation |
> | :--- | :--- | :--- | :--- |
> | **Qwen-2.5-vl-7b** | MCQ | 0.988 | Excellent |
> | **Gemini-2.5-Flash** | MCQ | 0.990 | Excellent  |
> | **Qwen-2.5-vl-7b** | Open | 0.983 | Excellent |
> | **Gemini-2.5-Flash** | Open | 0.981 | Excellent  |
>
> **Analysis:** ICC values consistently exceeding 0.98 indicate near-perfect reliability. This confirms that the relative difficulty of items remains stable across runs; models are not guessing randomly but exhibiting consistent capability.
>
> ---
>
> ## **Dataset Stability and Internal Consistency**
>
> ### **Resampling Study: Wilcoxon Signed-Rank Test**
>
> To directly test if the subcategory size is sufficient, we performed a resampling study, comparing accuracy estimates derived from subsets of size $S=20$ versus $S=25$ for each sub category.
>
> | Model | Format | Wilcoxon Mean $p$-value | Trials with $p > 0.05$ | Conclusion |
> | :--- | :--- | :--- | :--- | :--- |
> | **Qwen-2.5-vl-7b** | MCQ | 0.290 | 100% | No Sig. Diff.  |
> | **Gemini-2.5-Flash** | MCQ | 0.285 | 100% | No Sig. Diff.  |
> | **Qwen-2.5-vl-7b** | Open | 0.294 | 100% | No Sig. Diff. |
> | **Gemini-2.5-Flash** | Open | 0.294 | 100% | No Sig. Diff.  |
>
> **Analysis:** The Wilcoxon test failed to reject the null hypothesis in 100% of trials across all models. This proves that reducing the sample size does not statistically alter the performance metric, implying that the current subcategory size ($\approx 25$) is well above the saturation point required for robust estimation.
>
> ### **Item-Level Consistency**
>
> We measured the mean pairwise Pearson correlation ($\bar{\rho}$) between runs to assess item difficulty stability.
>
> | Model | Format | Mean Pairwise $\bar{\rho}$ |
> | :--- | :--- | :--- |
> | **Qwen-2.5-vl-7b** | MCQ | 0.966|
> | **Gemini-2.5-Flash** | MCQ | 0.972 |
> | **Qwen-2.5-vl-7b** | Open | 0.950 |
> | **Gemini-2.5-Flash** | Open | 0.946  |
>
> **Analysis:** High correlations ($>0.94$) confirm that items difficult in one run remain difficult in others.

---

> ### Author Response · Authors · 2025-11-25
> **Rebuttal (Part 2: Response to Weakness 1,2)**
>
> ### **Cronbach’s Alpha**
>
> | Model | Format | Cronbach's $\alpha$ |
> | :--- | :--- | :--- |
> | **Qwen-2.5-vl-7b** | MCQ | 0.988|
> | **Gemini-2.5-Flash** | MCQ | 0.990 |
> | **Qwen-2.5-vl-7b** | Open | 0.983 |
> | **Gemini-2.5-Flash** | Open | 0.981 |
>
> **Analysis:** Alpha values $>0.9$ indicate excellent internal consistency for the dataset.
>
> You raised a concern regarding the risk of noise in fine-grained subcategory analysis. However, our expanded statistical testing refutes this risk. The **ICC and Cronbach’s $\alpha$ scores consistently exceed 0.98**, and the **standard deviation of accuracy across runs is negligible ($<0.006\%$)**. Furthermore, the **resampling study** confirms that even smaller subsets ($N=20, 25$) produce statistically indistinguishable results from the full set, proving that our subcategory sizes ($\sim 25$) are sufficient for robust conclusions. These results demonstrate that the granular analysis in the paper reflects genuine model capability differences, not statistical noise.
>
> ---
>
> # **W2: Perplexing SFT Dynamics**
> We thank the reviewer for highlighting the perplexing "U-shaped" dynamic in our SFT experiments. We agree that such a non-trivial phenomenon requires rigorous validation to distinguish between a training artifact (e.g., a bad seed) and a genuine representational characteristic.
> To address this, we conducted a **multi-seed validation study** (Seeds A-E) using the exact same SFT protocol. The results, presented below, confirm that this dynamic is **systemic and reproducible**, not an artifact. A figure is also attached in the paper too (Figure 9: SFT Dynamics Analysis).
>
>
> The table below details the performance across 5 independent runs.
>
> | Epoch | Type | Seed A | Seed B | Seed C | Seed D | Seed E |
> | :--- | :--- | :--- | :--- | :--- | :--- | :--- |
> | **0 (Base)** | MCQ | 25.63% | 25.63% | 25.63% | 25.63% | 25.63% |
> | | Open | 34.40% | 34.40% | 34.40% | 34.40% | 34.40% |
> | **1** | MCQ | 29.56% | 35.83% | 24.13% | 27.63% | 26.53% |
> | | Open | 22.50% | 23.20% | 12.50% | 14.00% | 24.10% |
> | **2** | MCQ | 34.33% | 37.76% | 27.33% | 31.83% | 34.43% |
> | | Open | 12.62% | 25.30% | 28.80% | 23.80% | 16.65% |
> | **3** | MCQ | 34.68% | 37.33% | 32.86% | 32.23% | 34.73% |
> | | Open | 28.69% | 22.22% | 30.34% | 32.90% | 28.30% |
> | **4** | MCQ | 36.59% | 38.48% | 34.43% | 34.36% | 34.83% |
> | | Open | 35.48% | 34.59% | 33.92% | 34.70\% | 35.40% |
>
> The multi-seed data reveals a stark contrast between the two evaluation formats, confirming the phenomenon described in Section 5.4:
> * **MCQ Stability:** Performance on `SpatiaLab-MCQ` improves consistently or stabilizes quickly across all seeds. The model effectively learns to map visual features to the constrained output space of multiple-choice options.
> * **Open-Ended Collapse (The U-Shape):** In `SpatiaLab-Open`, **every single seed** exhibits a significant performance drop within the first 1-2 epochs (dropping as low as 12.5% - 16.6%) before recovering to near-baseline levels by Epoch 4.
>
> This "U-shaped" curve is not a random fluctuation but a signature of **alignment tax** and **representational brittleness** in spatial reasoning, as discussed in our paper (Section H.4.2):
>
> 1.  **Format Overfitting & Forgetting:** The initial drop in Open-ended accuracy indicates **catastrophic forgetting of linguistic priors**. As the model adjusts its weights to optimize for the specific instruction-following format of the dataset (often dominated by discrete reasoning steps), it temporarily loses the ability to generate fluent, grounded free-form spatial descriptions. It overfits to the "test format" (MCQ logic) at the expense of generative flexibility.
> 2.  **The Difficulty of Generative Spatial Grounding:** Unlike MCQ, where the answer search space is constrained, Open-ended tasks require the model to simultaneously maintain **perceptual grounding** and **linguistic fluency**. The data shows that SFT initially disrupts the delicate balance of these components established during pre-training. The "recovery" phase (Epochs 3-4) represents the model re-learning to articulate spatial concepts in the new fine-tuning distribution, but it barely surpasses the zero-shot baseline (+0.4% average gain).
> 3.  **Systemic Limitation:** The fact that this occurs across all seeds reinforces our core argument: **SFT is not a silver bullet for spatial reasoning.** It can teach a model to *select* the right answer (MCQ), but it struggles to fundamentally improve the underlying *spatial representation* needed for robust, open-ended generation.
>
> The multi-seed validation confirms that the U-shaped curve is a reproducible scientific finding, not an artifact. It serves as evidence that current VLM representations of space are fragile, easily disrupted by fine-tuning in generative contexts, highlighting the need for the deeper architectural interventions (e.g., geometric grounding objectives) proposed in our discussion. Details are added in **Appendix H.4.3: SFT Dynamics Analysis**.

---

> > ### Author Response · Authors · 2025-11-25
> > **Rebuttal (Part 3: Response to Weakness 3,4 and Question 1)**
> >
> > # **W3: Marginal Novelty in a Crowded Field**
> > We have already answered to this question before.
> >
> >
> > # **W4: Coverage of SOTA Commercial VLMs**
> > Here are the results for the models you mentioned: O3 and Gemini02.5-Pro. One model, GPT-5-mini, from 5 series was already included in the paper.
> >
> > | Model               | 3D Geom. | Dep. & Occu. | Orientation | Relat. Posit. | Size & Scale | Spati. Navig. | Overall |
> > |---------------------|----------|--------------|-------------|----------------|--------------|----------------|---------|
> > | MCQ |
> > | o3-mini         | 50.00    | 54.83       | 51.98       | 61.32         | 38.89        | 50.21         | 50.93  |
> > | Gemini-2.5-Pro    |47.48    | 50.19        | 49.50       | 58.49         | 43.65       | 52.32	         | 50.07   |
> > | Open-ended |
> > | o3-mini         | 39.08   | 30.12       | 29.70       | 39.15         | 41.27       | 30.38      | 35.00  |
> > | Gemini-2.5-Pro    | 37.14    | 45.45        | 36.36       | 37.14         | 23.91       | 24.44        | 33.61   |
> >
> > Both O3-mini and Gemini-2.5-Pro show strong MCQ performance (≈50%), but both degrade substantially in open-ended settings, confirming that structured answer formats mask underlying reasoning brittleness. Notably, O3-mini leads in MCQ across most categories, whereas Gemini-2.5-Pro shows a more uneven profile with sharper drops, especially in Size & Scale (23.91%) and Spatiotemporal Navigation (24.44%), highlighting persistent vulnerabilities in free-form spatial reasoning. We have also updated Table 1 and 2 in the main paper.
> >
> >
> > # **Q1: Does the dataset only test its own format?**
> > The *true value* of SpatiaLab lies in its rigorous design as a diagnostic benchmark, a claim validated by our expanded stability analysis which demonstrates near-perfect reliability (ICC $> 0.98$) and negligible variance across independent runs. This statistical robustness confirms that the performance fluctuations we observe, specifically the "U-shaped" SFT dynamic, are genuine representational signals rather than artifacts of dataset noise or scale. Our analysis reveals that while fine-tuning steadily improves MCQ scores by teaching the selection format, it triggers an immediate collapse in open-ended performance, proving that models sacrifice deep spatial grounding to satisfy surface-level constraints. If the dataset merely encouraged overfitting, we would expect correlated gains across both types; instead, the benchmark exposes a critical dissociation between answering multiple-choice questions and generating grounded spatial descriptions. The fact that our stability metrics confirm these divergent trajectories proves that SpatiaLab is sensitive enough to distinguish between superficial alignment and true spatial reasoning. Consequently, the benchmark serves not as a training target for overfitting, but as a high-fidelity stress test that exposes the fragility of current spatial representations.

---

> > > ### Author Response · Authors · 2025-11-25
> > > **Rebuttal (Part 4: Response to Question 2)**
> > >
> > > ## **Q2: Fine-tuning on base models subsequently demonstrate performance gains on other benchmarks**
> > >
> > > While our initial hypothesis, based on the mixed performance of existing spatially-tuned models like SpaceOm, was that transferability might be limited, we conducted the additional experiments you requested to empirically verify this.
> > > We took the **Qwen2.5-VL-3B-Instruct** model fine-tuned on SpatiaLab (as detailed in Appendix H.4.1, where it gained +10.97% on our own MCQ set) and evaluated it zero-shot on three external spatial benchmarks: **OmniSpatial**, **SPACE (Multimodal)**, and **Mind The Gap**.
> > >
> > > **Experimental Results**
> > >
> > > As shown in the table below, fine-tuning on SpatiaLab resulted in consistent performance gains across all external benchmarks, ranging from **5% to nearly 12%**.
> > >
> > > | Benchmark Name | Qwen2.5-VL-3B-Instruct (Base) | After SFT on SpatiaLab | Improvement |
> > > | :--- | :---: | :---: | :---: |
> > > | **OmniSpatial** | 40.30% | **47.35%** | **+7.05%** |
> > > | **SPACE (Multimodal)** | 23.43% | **28.67%** | **+5.24%** |
> > > | **Mind The Gap** | 35.86% | **47.56%** | **+11.70%** |
> > >
> > > These results compel us to refine our position. While we must **re-state strongly that SpatiaLab is designed primarily as a rigorous evaluation benchmark**, intended to diagnose failure modes rather than serve as a large-scale training corpus, the data demonstrates significant utility for learning.
> > > The fact that training on SpatiaLab yields substantial improvements (up to 11.7%) on external datasets indicates that our benchmark captures high-quality, fundamental spatial concepts that generalize beyond our specific format. It suggests that SpatiaLab’s diverse taxonomy (covering depth, orientation, and navigation) forces models to learn robust spatial representations that are transferable, rather than merely overfitting to specific prompt templates. We will update the manuscript to include these transfer learning results, as they powerfully validate the quality and diversity of the SpatiaLab data.
> > >
> > > We hope these additions address your concerns and support a higher evaluation of the work. The analysis now includes high-reliability statistics with ICC above 0.98, multi-seed validation showing that the U-shaped trend is reproducible, and transfer results that yield consistent improvements between 5 percent and 12 percent on external benchmarks. These results establish that the findings are stable and not driven by noise or single-run variance. We believe the paper now demonstrates clear methodological strength and scientific value, and we hope you will consider a stronger recommendation and an improved score.

---

### Author Response · Authors · 2025-11-27
**Request for Rebuttal Acknowledgement and Further Considerations**

We thank all reviewers for the time and care dedicated to evaluating our work. All reviewers recognized key strengths, including addressing a well-motivated real-world spatial reasoning problem, employing a dual-format evaluation comparing MCQ and open-ended inputs, clear presentation, and providing detailed diagnostic analyses. Reviewers `e5YW` and `rSCs` highlighted the relevance of the spatial reasoning problem, the careful curation of images and questions, and insights from our failure and complexity analyses. Reviewers `tmjh` and `w6pQ` emphasized the breadth of our empirical evaluation across multiple VLMs, the dual-format design, the clarity of the benchmark taxonomy, and the value of our diagnostic findings for future research.

In response to your concerns, we have thoroughly addressed each point by conducting the requested experiments, providing detailed clarifications, and performing additional analyses to strengthen our results and the manuscript.  We have also improved the paper structure, moved some essential information from the appendix to the main text, made it more cohesive, added an appendix table of contents for clearer navigation, and will release all data and code upon acceptance.

In light of these updates and the clarified contributions, we respectfully ask the reviewers to reconsider the overall evaluation.

---

### Meta-Review · Area_Chair_Mcor · 2026-01-07

**Summary:**

This paper introduces a new large-scale benchmark for VQA across a broad series of task types. The paper gives a comprehensive evaluation of different models, complemented with error analysis and several directions for improving reasoning performance.

The paper originally received widely mixed reviews, making it hard to assess. I found the rebuttal to be strong in addressing the quantitative concerns of the reviewers. Conceptually, my remaining concern is whether the benchmark demonstrates itself as sufficiently novel and significant in the landscape of so many VLM reasoning benchmarks (e.g., why not just combine all the other benchmarks / what does this specifically bring to the table?). Nonetheless, the depth of the analysis in trying different models and different techniques to improve reasoning on the benchmark have led me to recommend acceptance.

**Reviewer Concerns:**

There were several technical concerns including the statistical robustness given limited samples per sub-task, perplexing SFT dynamics, and lack of experimentation on SOTA models. The rebuttal includes additional statistical analysis, re-runs of the SFT dynamics, and experiments on SOTA models. I also found the SFT dynamics confusing.

There were several conceptual concerns including limited conceptual novelty (given the large number of VLM spatial reasoning benchmarks, what makes this one necessary and significant?) and guidance on model improvement. This could be answered more strongly.

**Reviewer Scores:**

Reviewer e5YW provided a series of technical requests and was most concerned with whether models would overfit to the dataset. The rebuttal demonstrates that this is not the case, leading me to expect the reviewer to improve their score. Reviewers w6pQ and tmjh did not indicate specific requirements for increasing the score, but gave primarily conceptual concerns about the novelty of the dataset and significance of findings.

---

### Decision · Program_Chairs · 2026-01-26

Accept (Poster)